JGP Journal of General Physiology

# Interactions between selectivity filter and pore helix control filter gating in the MthK channel

Wojciech Kopec[1], Andrew S. Thomson[2], Bert L. de Groot[1], and Brad S. Rothberg[2]

K[+] channel activity can be limited by C-type inactivation, which is likely initiated in part by dissociation of K[+] ions from the selectivity filter and modulated by the side chains that surround it. While crystallographic and computational studies have linked inactivation to a "collapsed" selectivity filter conformation in the KcsA channel, the structural basis for selectivity filter gating in other K[+] channels is less clear. Here, we combined electrophysiological recordings with molecular dynamics simulations, to study selectivity filter gating in the model potassium channel MthK and its V55E mutant (analogous to KcsA E71) in the pore-helix. We found that MthK V55E has a lower open probability than the WT channel, due to decreased stability of the open state, as well as a lower unitary conductance. Simulations account for both of these variables on the atomistic scale, showing that ion permeation in V55E is altered by two distinct orientations of the E55 side chain. In the "vertical" orientation, in which E55 forms a hydrogen bond with D64 (as in KcsA WT channels), the filter displays reduced conductance compared to MthK WT. In contrast, in the "horizontal" orientation, K[+] conductance is closer to that of MthK WT; although selectivity filter stability is lowered, resulting in more frequent inactivation. Surprisingly, inactivation in MthK WT and V55E is associated with a widening of the selectivity filter, unlike what is observed for KcsA and reminisces recent structures of inactivated channels, suggesting a conserved inactivation pathway across the potassium channel family.

## Introduction

Potassium (K[+]) channels are critically important in the control of many cellular functions, including action potential firing, neurotransmitter and hormone release, and muscle contraction (Hille, 2001). A great deal of effort has been focused on understanding the molecular basis of K[+] permeation and channel gating, and it is now clear that K[+] channels gate at multiple structural loci (Panyi and Deutsch, 2006). For example, in voltage-gated K[+] channels as well as inwardly rectifying K[+] channels, the pore-lining helices from each subunit can form a constriction (activation gate) at the cytosolic entrance to the pore (S6 gating; Yellen, 1998; Jensen et al., 2012; Yellen, 2002), whereas some these and other K[+] channels can also undergo conformational changes near the selectivity filter, at the extracellular end of the pore (SF gating), to gate ion permeation (Fig. 1, a and b; Cordero-Morales et al., 2006; Cordero-Morales et al., 2007). A combination of structural and functional studies has revealed that the mechanisms underlying each of these gating processes are unlikely to arise from a single blueprint, and that diversity among K[+] channels is likely to have yielded several complementary mechanisms to underlie gating that may be coupled to transmembrane voltage, ligands, or other stimuli (Xu and McDermott, 2019; Thomson and Rothberg, 2010; Pless et al., 2013; Schönherr and Heinemann, 1996). It was observed

previously that the prokaryotic K[+] channel MthK exhibits SF gating that shares many properties in common with C-type inactivation of Kv channels. Specifically, channel opening is greatly decreased by increasing depolarization and decreasing extracellular K[+] concentration ([K[+]]), and the gating process shows a selective sensitivity to cationic channel blockers, similar to that observed with the modulatory effect of quaternary ammonium derivatives on C-type inactivation in the *Shaker* K[+] channel (Thomson and Rothberg, 2010; Thomson et al., 2014; López-Barneo et al., 1993; Baukrowitz and Yellen, 1995, 1996a, 1996b). Interestingly, pore-helix glutamate that is a key determinant of inactivation in KcsA (E71) does not play a role in MthK gating; instead, this position contains a hydrophobic valine side chain (V55, Fig. 1, c and d), identical to *Shaker* (V438), as well as mammalian Kv1.2 (V366) and Kv1.3 (V440) channels. This implies that the molecular driving forces and structural changes underlying SF gating among these channels may be different from one another. To better understand the molecular basis for SF gating in MthK, we examined the functional consequences of substituting V55 in MthK with (protonated) glutamate (Fig. 1 e) by electrophysiology and analyzed potential underlying molecular effects by in silico electrophysiology molecular simulation. We observe that the V55E mutation yields a gating phenotype

[1]Computational Biomolecular Dynamics Group, Max Planck Institute for Multidisciplinary Sciences, Göttingen, Germany;   [2]Department of Medical Genetics and Molecular Biochemistry, Temple University Lewis Katz School of Medicine, Philadelphia, PA, USA.

Correspondence to Wojciech Kopec: wkopec@gwdg.de.

with markedly reduced open state stability, manifested as decreased channel open probability and decreased open times over a wide range of voltages. Further, the V55E mutation also decreases the unitary conductance of MthK channels. On the molecular level, simulations reveal two distinct orientations of the E55 side chain ("vertical" and "horizontal") that differentially affect the SF properties. The vertical orientation is reminiscent of E71 seen in structures of KcsA (Fig. 1 d), where it is hydrogen-bonded with an aspartate behind the SF (D80 in KcsA; D64 in MthK). In this orientation, MD simulations predict a low conductance of MthK V55E. In the second, horizontal orientation, that is with the E55 side chain oriented perpendicular to the membrane, K$^+$ conductance is closer to MthK WT, but longer simulations show that this orientation promotes SF entering into the non-conducting (inactivated) state. In MD simulations, SFs of both WT and V55E channels undergo gating transitions that share similarities with the C-type inactivated conformations observed in stably inactivated mutants of *Shaker* and Kv1.2 channels, and in WT Kv1.3 channels. In a full accord with electrophysiology, V55E SF shows, in simulations, a larger degree of instability than WT, which arises from the entry of additional water molecules behind the SF and K$^+$ ion unbinding from the top of the SF. Taken together, these data suggest a mechanism for SF gating (inactivation) in MthK channels and highlight the critical role for interactions between the SF and pore-helix in stabilizing the K$^+$ ion conduction pathway. Finally, we highlight several important differences between two popular molecular force fields (CHARMM36m and AMBER14sb) used in MD simulations of K$^+$ channels.

## Materials and methods

### Protein preparation and single-channel recording

MthK was expressed, purified, and reconstituted into proteoliposomes as described previously (Pau et al., 2010). Proteoliposomes were composed of *Escherichia coli* lipids (Avanti), rapidly frozen in liquid N$_2$, and stored at –80°C until use. Protein concentrations in proteoliposomes ranged from 5 to 25 µg protein per mg lipid. Mutations were introduced using the QuikChange Site-Directed Mutagenesis Kit (Agilent Technologies) and confirmed by DNA sequencing.

Recordings were obtained using planar lipid bilayers of POPE:POPG (3:1) in a horizontal bilayer chamber, essentially as described previously (Thomson et al., 2014). For these experiments, solution in the *cis* (top) chamber contained 200 mM KCl, 10 mM HEPES (pH 8.1), and 100 µM Cd$^{2+}$. Solution in the *trans* (bottom) chamber contained either 5 or 200 mM KCl and 10 mM HEPES (pH 7.0). This arrangement ensured that MthK channels incorporated into the bilayer with the cytosolic side facing the *trans* chamber would be silent, whereas channels with the cytosolic side facing the *cis* chamber would be maximally activated (by Cd$^{2+}$). Within each bilayer, multiple solution changes were performed using a gravity-fed perfusion system that enabled exchange of solution in the *trans* chamber. To ensure completeness of solution changes, the *trans* chamber was washed with a minimum of 10 ml (~10 chamber volumes) of solution prior to recording under a given set of conditions.

Single-channel currents were amplified using a Dagan PC-ONE patch clamp amplifier with low-pass filtering to give a final effective filtering of 1 kHz. This level of filtering corresponds to a dead time (minimum time required for an event to reach 50% threshold) of 0.179 ms. Currents were digitized at a rate of 50 kHz (20 µs per sample) and analyzed by measuring durations of channel openings and closings at each current level by 50% threshold analysis, using pClamp 9.2. These were used to calculate NPo as:

$$NPo = \sum_{i=1}^{n} iP_i, \qquad (1)$$

in which $N$ is the number of channels in the bilayer, $i$ is the open level, and $P_i$ is the probability of opening at that level. The mean single-channel open probability ($Po$) is obtained by dividing $NPo$ by $N$, determined by recording under conditions where the maximum level of channel opening can be observed. The voltage-dependence of $Po$ was described by the Boltzmann equation

$$Po = Pomax \big/ \{1 + \exp[z\delta(V - V_{1/2})/k_BT]\}, \qquad (2)$$

in which $Pomax$ is the maximal $Po$, $z\delta$ is the effective gating valence (in $e_0$), $V_{1/2}$ is the voltage at half-maximal $Po$, $k_B$ is Boltzmann's constant, and $T$ is temperature. Data points (e.g., $Po$ and mean interval durations) are presented as mean ± SEM of three to five observations for each data point, and collectively represent data from a total of 17 different bilayers.

Mean unitary current amplitudes were determined by constructing all-points histograms from digitized currents at each voltage. These could be described in part by a sum of two Gaussian distributions,

$$f(I) = A_O exp\left\{-[(I - I_O)/\omega_O]^2\right\} + A_C exp\left\{-[(I - I_c)/\omega_c]^2\right\}, \qquad (3)$$

in which $A_O$ and $A_C$ are the heights of the distributions of open and closed current levels, respectively, $I_O$ and $I_C$ are the means of the distributions, and $\omega_O$ and $\omega_C$ are the widths of the distributions. Thus, the unitary current amplitude is given by the difference $I_O$ minus $I_C$. As a check on this method of estimating mean current amplitudes, amplitudes were additionally determined for individual idealized openings in conjunction with 50% threshold analysis using pClamp9.2. In this approach, the open amplitude is calculated as the difference between the closed baseline level and mean current level following a 50% threshold crossing, excluding points during the transitions between the open and closed levels (i.e., 0.179 ms after the threshold-crossing for opening, and 0.179 ms prior to the threshold-crossing for closing for currents in this paper, which were low-pass filtered at 1 kHz, as described in the pClamp User Guide). If the measured duration of the opening is <0.358 ms (i.e., 2 × 0.179 ms), then the amplitude is recorded as the current level at the midpoint of the opening.

### MD simulations

We used the same setup as we used for MthK WT in our recent publication (Kopec et al., 2019), where we thoroughly

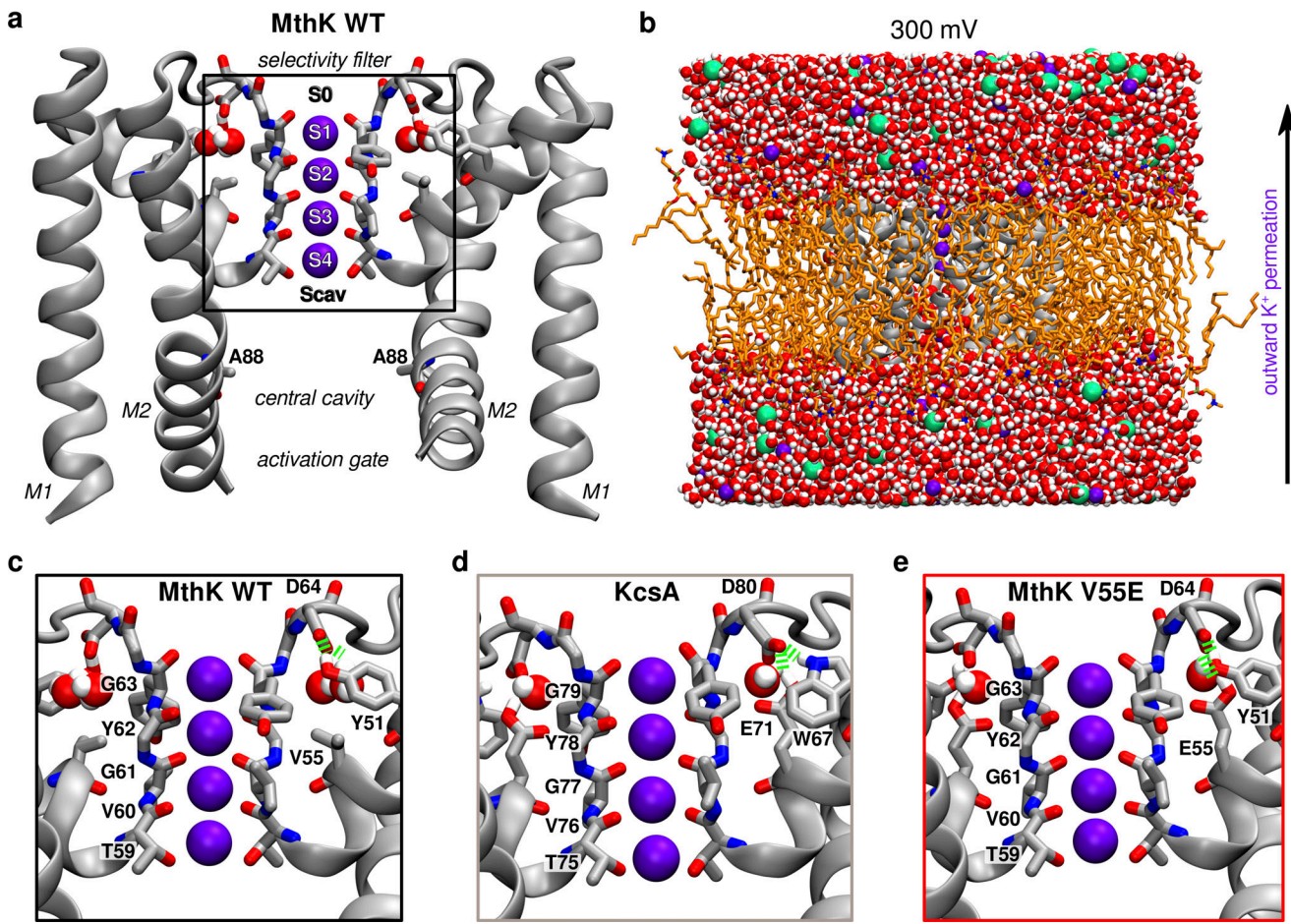

Figure 1. **Basic architecture of a K⁺ channel. (a)** Crystal structure of MthK in an open state (PDB ID 3LDC, only two diagonally opposite subunits of a tetramer are shown), with highlighted important elements—the selectivity filter in its conducting conformation consists of four main K⁺ binding sites S1–S4 and additional binding sites S0 and Scav. The central cavity, typically filled with water molecules, is located below the filter and above the activation gate. **(b)** All-atom system used in MD simulations. The channel is shown in grey, lipids (POPC) in orange, water as red-white spheres, K⁺ ions as purple spheres, and chloride ions as green spheres. Transmembrane voltage of 300 mV was used in simulations, resulting in an outward permeation of K⁺ ions through the channel as indicated by the arrow. **(c–e)** Closer view on an interaction network behind the selectivity filter. In WT MthK, D64 makes hydrogen bonds (green springs) with Y51 and two crystallographic water molecules (c). In KcsA (PDB ID 1K4C; d) a residue corresponding to Y51 is W67. Additionally protonated glutamate (E71, V55 in MthK) takes part in the same interaction network, with only one water molecule. The model of MthK V55E (e) is based on MthK WT, with V55 replaced with protonated E71 from KcsA and one water molecule.

characterized ion permeation and selectivity filter-activation gate coupling, using the CHARMM36m (referred to as CHARMM; MacKerell et al., 1998; Huang et al., 2017) and AMBER14sb (referred to as AMBER; Maier et al., 2015) force fields. We did not use any further modifications of the carbonyl oxygen–potassium ion interactions, as non-modified force fields seem to better reproduce solvation-free energies (Yu et al., 2010). The data for MthK WT in the current manuscript (black traces) are taken directly from our previous work (Kopec et al., 2019). For MD simulations of the V55E mutant, the valine residue in MthK WT was replaced by protonated glutamate from KcsA (PDB ID 1K4C) after aligning both channels with their (identical) selectivity filters. The resulting structure was further equilibrated for at least 100 ns, whereas all other simulation details and parameters were identical to those used before. After equilibration, we imposed the desired degree of activation gate opening by applying harmonic distance restraints between the

CA atoms of the pairs of terminal residues—P19 and F97. The equilibrium distances were adopted from our previous publication, and the resulting average distances are plotted in Fig. 4 b and Fig. 5 a. For additional simulation sets, where the orientation of the E55 side chain was restrained as well, additional restraints were used. In the AMBER force field, where the horizontal orientation is frequently visited, we used the following restrains: to keep the E55 side chain in the vertical orientation, we restrained the distance between the proton from the protonated side chain of E55 and one of the carboxylate oxygens of the D64 residue, to the distance of 0.167 nm (an average seen in non-restrained simulations) with a force constant of 500 kJ/mol/nm². To keep the E55 side chain in the horizontal orientation, we restrained the distance between the non-protonated oxygen from the E55 side chain and the amide proton of the G61 residue, which is nearby, to the distance of 0.200 nm with the force constant of 500 kJ/mol/nm². In the CHARMM

force field, where the vertical orientation is seen almost exclusively, we had to use much stronger restraints to generate MD simulations with the horizontal orientation. We first used some of the snapshots from the AMBER simulations, with the E55 side chain in the horizontal orientation, converted them to the CHARMM force field, and then used position restraints on the side-chain atoms, with a force constant of 20,000 kJ/mol/nm². Only such a strong force constant prevented the side chains from returning to the vertical orientation.

The systems were simulated at an external field applied along the z-axis to generate the membrane voltage of ∼300 mV. The voltage ($V$) was calculated with:

$$V = EL_z, \qquad (4)$$

where $E$ denotes the applied electric field and $L_z$ the length of the simulation box along the z-axis. All simulations were performed with GROMACS MD software, versions 5.1 and 2020 (Berendsen et al., 1995; Hess et al., 2008; Abraham et al., 2015; Lindahl et al., 2001; Pronk et al., 2013). All the simulation settings were identical to our previous work (Kopec et al., 2019). For each opening and force field, 10 (short) or 20 (long) simulations were performed, each lasting 1 µs (short simulations) or 5 µs (long simulations), which resulted in a total simulation time (for all systems) of ∼920 µs. Ionic currents were estimated by counting the number of K⁺ ions crossing the SF in each simulation, using a custom FORTRAN code (available as a Supplementary software associated with our previous publication, Kopec et al., 2019). Remaining quantities were calculated using GROMACS tools—gmx distance (distances), gmx angle (angles), and gmx select (number of water molecules). Most of the presented data (besides individual simulation traces and counts) are averages from at least 10 independent simulation replicas. Error bars represent 95% confidence intervals or SEM. For data analysis, Python 3 was used, together with numpy (Oliphant, 2007), pandas (McKinney, 2011), seaborn, matplotlib (Hunter, 2007), scipy (Jones et al, 2001), and bootstrapped modules. Molecular visualizations were rendered with VMD (Humphrey et al., 1996).

### Protonation free energy calculations
To assess the free energy of deprotonation of the E55 side chain in its horizontal orientation, we used non-equilibrium free energy calculations using GROMACS 2020 and the pmx package (Gapsys et al., 2015) to calculate the *pKa* shift ($\Delta pKa$) of the E55 side chain from the experimental value of 4 of glutamate in bulk water (Thurlkill et al., 2006), following a recent application (Bastys et al., 2018). In detail, one of the four protonated E55 was selected, and the proton on the carboxylate group was alchemically switched to a dummy atom. At the same time, an opposite process, i.e., switching a dummy atom to a proton on the carboxylate group, was performed for glutamate-containing tripeptide (G-E-G) in bulk, that served as a reference state (see Fig. S31 a). Both processes were carried out simultaneously in the same simulation box and coupled to the same λ variable, ensuring the electrical neutrality of the simulation box at all times. This way, the calculated ΔG (Fig. S31, d and e) is equal to ΔG_protein—ΔG_reference, with the G-E-G tripeptide as a

reference state, and can be directly plugged into the equation (Awoonor-Williams and Rowley, 2016)

$$\Delta pKa = \Delta G / (2.303RT), \qquad (5)$$

where $R$ and $T$ are the gas constant and simulation temperature, respectively. In the free-energy calculations, a lower temperature of 298K and salt concentration of 150 mM was used. The side chain of E55 was kept restrained with position restraints in the horizontal orientation as described above. The central atom of the G-E-G peptide was position restrained as well, to keep the tripeptide away from the membrane and the channel. For each force field, five independent equilibrations, each 20 ns long for both end states, were performed. From each equilibration, 100 equally spaced snapshots were selected (ignoring the first 5 ns) for fast, non-equilibrium transitions (500 transitions from state 0 [dummy] to 1 [proton] and 500 transitions from state 1 to 0). Each non-equilibrium transition lasted 10 ns, due to the relatively slow dynamics of the D64 side chain, which could flip toward the extracellular space when E55 was deprotonated (see Fig. S31, b and c). The final free energy difference was then calculated using pmx. Low error bars (see Fig. S31, d and e) across the five repeats indicate converged simulations.

### Online supplemental material
Fig. S1 shows differences in E55 dihedral energetics in two force fields. Figs. S2, S3, S4, S5, S6, S7, S8, and S9 show time evolution of the distance between the D64 CG and Y51 HH atoms in individual MD trajectories. Figs. S10, S11, S12, S13, S14, S15, S16, and S17 show time evolution of distances between G61 CA atoms in oppositely oriented monomers in individual MD trajectories. Figs. S18, S19, S20, S21, S22, S23, S24, and S25 show time evolution of distances between G63 CA atoms in oppositely oriented monomers in individual MD trajectories. Fig. S26 shows conformations of MthK and MthK V55E in MD simulations with the voltage of 150 mV. Fig. S27 shows average occupancies of ion binding sites in MD trajectories. Fig. S28 shows probabilities of molecular events in MD simulations of MthK WT with the AMBER force field. Fig. S29 shows D64 dynamics in MthK WT with different force fields in MD simulations. Fig. S30 shows the effect of the V55E mutation on D64 dynamics in MD simulations. Fig. S31 summarizes free energy calculations of D64 protonation. Fig. S32 compares selectivity filters of MthK and KcsA channels.

## Results
### The V55E mutation decreases stability of the open state
Mutation of E71 to alanine, in the pore-helix of KcsA, was observed to abrogate a key interaction between the side-chain carboxylates of E71 and D80; elimination of this interaction apparently results in stabilization of the open state and loss of inactivation in KcsA channels (Cordero-Morales et al., 2006). It was observed previously that MthK channels can also exhibit an inactivation-like phenomenon, in which the open state is stabilized by occupancy of a K⁺-selective site at the external end of the SF (Thomson and Rothberg, 2010; Thomson et al., 2014). To determine whether MthK inactivation and KcsA inactivation might arise through similar molecular interactions, we

examined functional consequences of substituting MthK V55 (at the position analogous to KcsA E71) with glutamate. We observe that with symmetrical 200 mM K$^+$ solutions, MthK WT channels, activated by 100 µM Cd$^{2+}$ at the cytosolic side of the channel, gate with high open probability (Popen) of >0.9 at voltage ranging from −150 to +140 mV (Fig. 2; Thomson et al., 2014; Suma et al., 2020; Smith et al., 2013). At strongly depolarized voltages (>140 mV), Popen decreases sharply with increasing depolarization, with a voltage-dependence of e-fold per 14 mV and a V$_{1/2}$ for this decrease at 190 ± 1.5 mV (7). This robust decrease in Popen is enhanced by decreasing external [K$^+$] (K$^+_{ext}$), such that a decrease from 200 to 5 mM K$^+_{ext}$ shifts the V$_{1/2}$ leftward from 190 to 95 ± 2.6 mV (Fig. 2 c; Thomson et al., 2014). These gating effects are described by a model in which voltage-dependent outward K$^+$ ion movement, followed by dissociation of a K$^+$ ion from the external end of the SF, leads to a conformational change within the SF that consists of an off-axis rotation of a carbonyl group (at V60) that normally lines the selectivity filter (Thomson and Rothberg, 2010; Thomson et al., 2014). This carbonyl rotation can halt conduction and thus gate the channel (Thomson et al., 2014). In contrast, with symmetrical 200 mM K$^+$ solutions, the MthK V55E mutant reaches a maximum Popen of 0.64 ± 0.02 at −100 mV, and Popen decreases with either depolarization or stronger hyperpolarization, reaching a value of 0.054 ± 0.02 at 200 mV and 0.24 ± 0.01 at −200 mV (Fig. 2 c). Over the entire 400 mV voltage range examined in these experiments (from −200 to +200 mV), MthK V55E channels exhibited lower Popen than MthK WT channels recorded under the same conditions. The single-channel currents in Fig. 2 a suggested that the basis for the decreased Popen in MthK V55E channels may be a decrease in mean open time compared to WT. Consistent with this observation, Fig. 2 d illustrates that the mean open time of V55E channels is much briefer than WT over voltages ranging from −200 to +200 mV. Mean open times of V55E channels were 10- to over 100-fold briefer than WT, with a maximal mean open time of 1.6 ± 0.03 ms at −100 mV and 0.41 ± 0.02 ms at 200 mV for V55E channels. In contrast, the mean open time of MthK WT channels reached a maximum of 330 ± 46 ms at +80 mV, and decreased to 23.6 ± 5.3 ms at 200 mV. Although the durations of openings were markedly different between V55E and WT channels, closed times of V55E channels were very similar to WT over a wide range of voltages up to 140 mV (Fig. 2 e). Interestingly, although V55E channels are characterized by lower Po than WT channels over a wide range of voltages, WT channels show an increase toward longer mean closed times than V55E at voltages >140 mV (at 200 mV, mean closed time of WT, 58.2 ± 17 ms; V55E, 9.11 ± 2.1 ms). This is consistent with the idea that destabilization of the open state in V55E channels, rather than increased stabilization of the closed state, is the critical determinant of the gating phenotype in V55E channels. Together, these observations indicate that the V55E mutation leads to decreased lifetime, and thus decreased stability of the open (conducting) state.

## The V55E mutation decreases unitary conductance

In single-channel recordings, one can observe that in addition to decreased open times, V55E channel openings frequently do not reach the same amplitude as WT openings (Fig. 2 a). Measuring the amplitudes of unitary currents for V55E channels is made complicated by the observation that V55E openings are very brief (Fig. 2 d and Fig. 3), and amplitudes of brief events are truncated by the effects of filtering, such that with the 1 kHz low-pass filtering for recordings in these studies (resulting in a dead-time of 0.179 ms; see Materials and methods), the amplitudes of openings with durations <0.716 ms (dead-time × 4) will be truncated (McManus et al., 1987). In addition, channel openings can occur in bursts containing brief closings, and very brief closings that fail to cross the threshold could also affect estimates of unitary current amplitude.

To gain insight toward whether the V55E mutation might reduce the unitary conductance of the channel, we quantified unitary current amplitudes over a range of voltages by constructing all-points histograms and determined the difference between mean open and closed (baseline) current levels based on the peaks of these histograms. Representative histograms of WT and V55E currents at 100 and 200 mV are shown in Fig. 3, a and b. For WT channels, all-point histograms appeared as bimodal distributions, corresponding to the closed and open current levels for these channels (Fig. 3 c). Each peak was fitted with a Gaussian distribution (blue dashed curves) to estimate the mean open and closed current levels, and unitary current was determined as the difference between the levels. So, for example, the WT open channel current amplitude at 200 mV from the representative all-points histogram in Fig. 3 a was 14.9 pA (15.7 − 0.78 pA), and the mean WT amplitude at 200 mV was 14.78 ± 0.17 pA (n = 3).

For V55E channels, unitary current measurements by the all-points histogram approach (Fig. 3 b) were complicated by the skewed distribution of points in the intermediate current range between open and closed. The shapes of the V55E all-points histograms can be attributed to the overrepresented time spent in transitions between opening and closing in V55E channels; because V55E openings were so brief, a large fraction of the digitized signal comprised transitions between open and closed levels. In addition, the low open probability of these data resulted in very small peaks for the open levels in most of the data sets. To better resolve both the maximum and minimum current levels, we plotted the all-point histograms on semilogarithmic coordinates (lower panels in Fig. 3, a and b). Using semi-logarithmic coordinates, one can observe the rapid fall-off at the ends of the histograms; these were well-described by the tails of the Gaussian functions (blue dashed curves) to enable estimates of the open and closed levels. Thus, for example, although the open probability for V55E channels at 200 mV was low (∼0.05), Fig. 3 b (lower right panel) shows that the maximum open current level at 200 mV could be resolved as 9.8 pA, yielding an estimate of open amplitude as 9.5 pA (9.8 minus 0.34 pA); the mean V55E amplitude at 200 mV for channels in this analysis was 8.98 ± 0.040 pA (n = 3), which was significantly lower than the WT current amplitude (14.8 ± 0.17 pA) measured in the same conditions (t test; P < 0.001).

As an additional check on our estimates of open current levels from WT and V55E channels, we measured the amplitudes

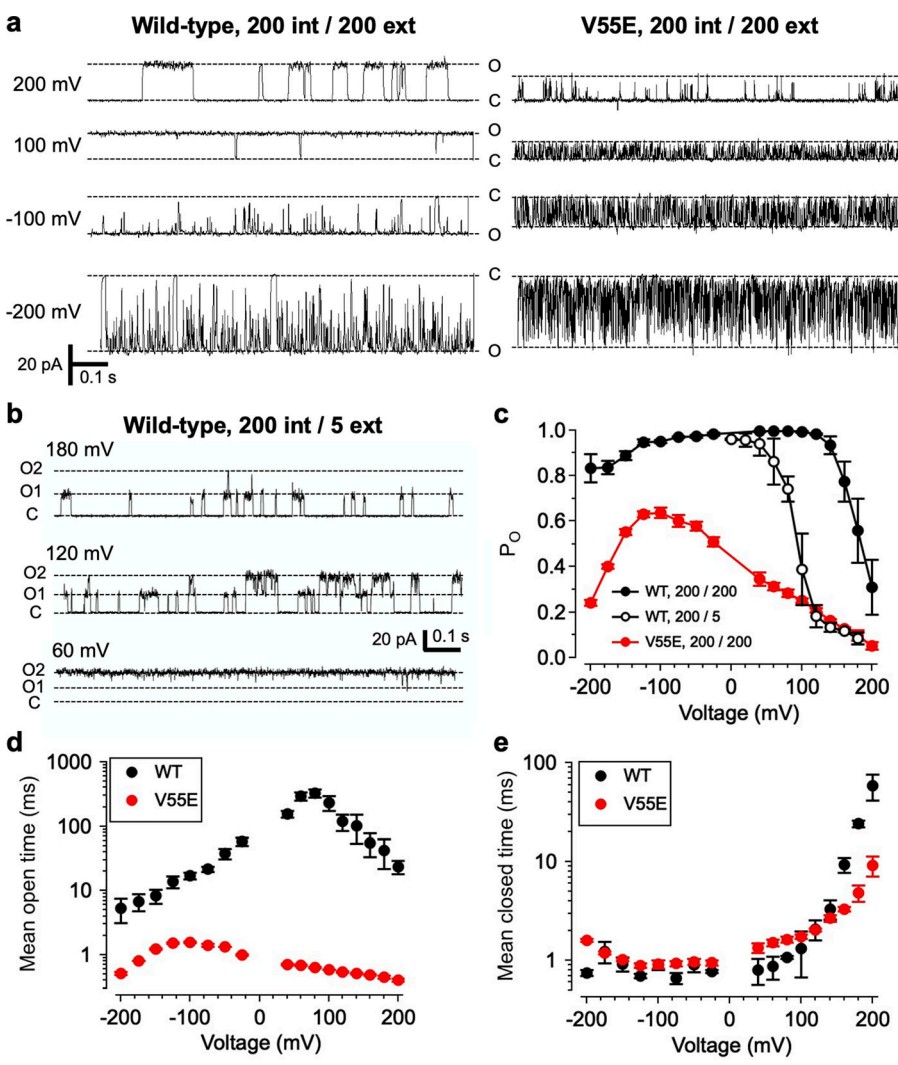

**Figure 2. Effects of V55E mutation on MthK channel gating. (a)** Representative single-channel current through MthK WT and V55E channels over a range of voltages. For these recordings, solutions at both sides of the membrane contained 200 mM KCl. "O" and "C" indicate open and closed channel current levels, respectively. Strong depolarization (200 mV) leads to decreased open probability (Po) for WT channels, and V55E channels consistently display briefer openings than WT at all tested voltages. **(b)** Current through MthK WT channels with 5 mM external K[+]. This bilayer contained two active channels, indicated by open levels "O1" and "O2." Lower external [K[+]] enhances inactivation at mildly depolarized voltages. **(c)** Po vs. voltage for WT channels with 200 mM (black circles, $n$ = 5 bilayers) or 5 mM external K[+] (open circles, $n$ = 3 bilayers), compared with V55E with 200 mM external K[+] ($n$ = 5 bilayers). **(d)** MthK WT channel open times were voltage dependent, with a maximal mean open time of 1,200 ± 120 ms at 60 mV; in contrast V55E were much briefer, with the mean open time not exceeding 6.7 ± 0.2 ms over a 400-mV range. **(e)** WT and V55E closed times were similar over a wide range of voltages, consistent with the idea that the major effect of the V55E mutation was to destabilize the open/conducting state relative to WT. [K[+]]$_{ext}$ = 200 mM (WT $n$ = 5; V55E $n$ = 5).

of individual openings in conjunction with measurement of dwell times (see Materials and methods). As noted above, amplitudes of brief openings with durations <0.716 ms are truncated due to the effects of low-pass filtering (McManus et al, 1987). Although in principle, openings with durations >0.716 ms should be long enough to reach the "full" open current level, openings very close to this threshold would yield amplitude estimates based on only a few digitized points and would thus be vulnerable to error due to the presence of noise (up to ±0.4 pA). We therefore used a threshold of 1.074 ms (dead-time × 6) for openings included in this analysis. Representative openings meeting the 1.074 ms criterion are shown in Fig. 3 d (long duration V55E openings are indicated by asterisk). Histograms constructed from WT and V55E unitary current measurements at 100 and 200 mV (Fig. 3, e and f) illustrate that long duration V55E openings have significantly lower amplitudes than WT amplitudes. For openings at 100 mV, the mean amplitude for WT was 9.6 ± 0.5 pA, while V55E was 5.8 ± 0.5 pA; at 200 mV, WT openings were 14.2 ± 0.9 pA, while V55E openings were 10.0 ± 1.3 pA (given as mean ± SD). These measurements were consistent with those obtained using all-point histograms, together supporting the idea that the V55E mutation underlies

altered channel function in terms of both reduced stability of the open conformation and ion conduction.

## Shorter in silico electrophysiology simulations—AMBER14sb force field

To probe the effects of the V55E mutation in the MthK channel on an atomistic scale, we used our previous computational protocol to monitor changes in outward current, caused by motions of the inner helices (the activation gate, Fig. 4 a; Kopec et al., 2019). Initially, we used the AMBER14sb force field (referred to as AMBER, see Materials and methods for details). When compared to the WT channel, simulated currents at 300 mV in MthK V55E display a similar overall behavior, with an initial increase in magnitude upon the activation gate opening (measured as an average distance between CA of A88 residues from oppositely oriented monomers of MthK; Fig. 4 b), that is between 1.4 and 1.6 nm, from the initial value of 2–3 pA. Around the opening of 1.6 nm, the current reaches its maximum of ~16 pA in WT and ~10 pA in V55E. Further activation gate opening (above the value of 1.6 nm) leads to a current decrease, back to 2–3 pA in both channels. In WT, we have previously traced this behavior to the variations in the width of the S4 binding site, formed exclusively

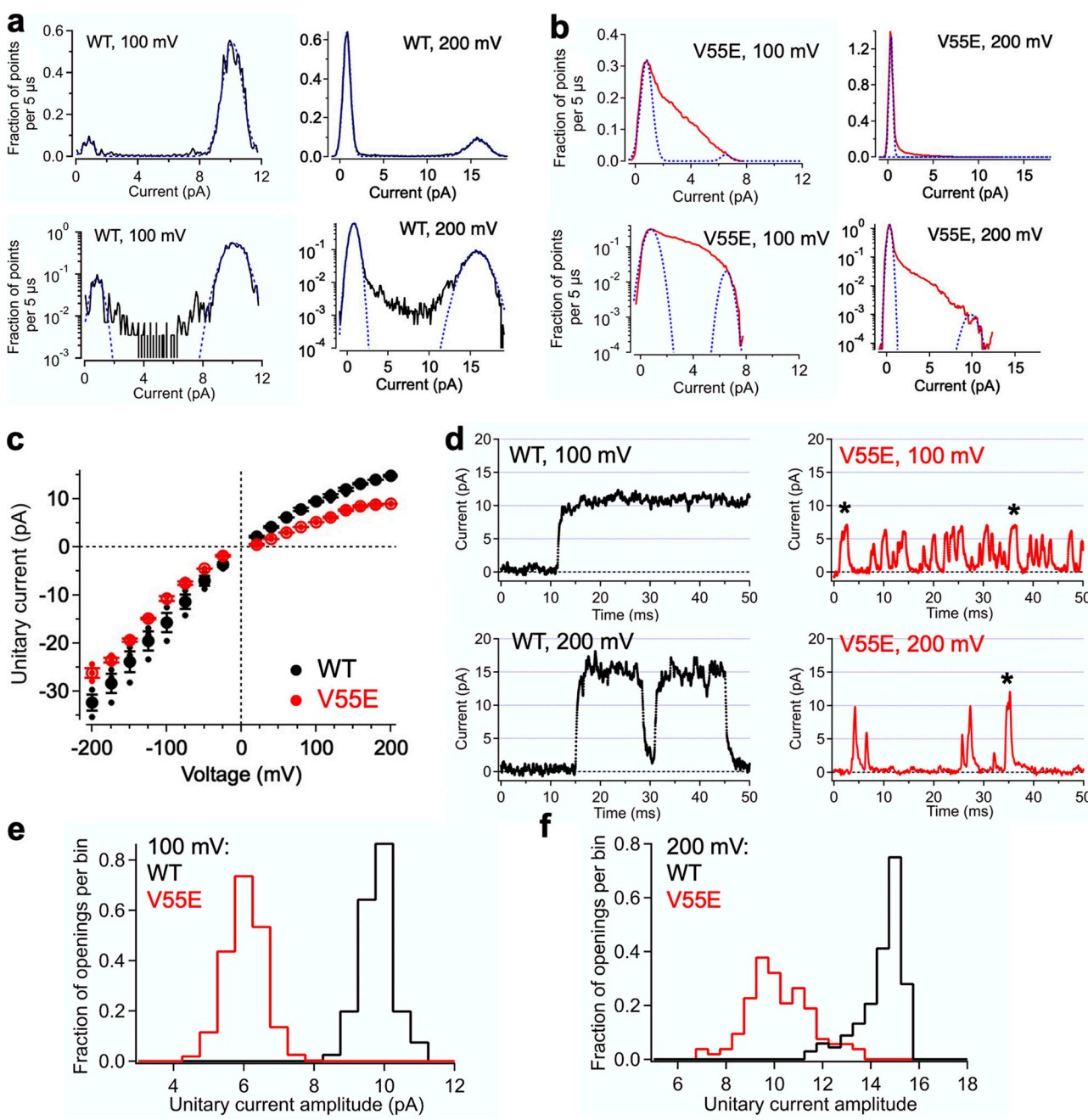

Figure 3.  **Effects of V55E mutation on MthK channel conductance. (a and b)** Representative all-points histograms from a WT and a V55E channel, respectively, constructed from selected data segments at 100 and 200 mV. Peaks correspond to open and closed current levels. These were fitted with two Gaussian components (blue dashed lines), and the difference between the means yields an estimate of the current amplitude. Upper panels show histograms on a linear scale; the same histograms are shown on a logarithmic scale on the corresponding lower panels. **(c)** Unitary current vs. voltage for WT and V55E channels, determined from all-points histograms at voltages ranging from −200 to +200 mV. Data points represent means ± SEM (smaller points represent data from individual experiments). **(d)** Representative single-channel openings from WT and V55E channels at 100 mV (top) and 200 mV (lower). V55E openings with durations >1.074 ms are indicated by an asterisk. **(e)** Histograms of amplitudes of single-channel openings with durations >1.074 ms at 100 mV, each normalized to an area of 1.0 (WT, 81 openings; V55E, 963 openings). Mean unitary currents for these openings were: WT 9.6 ± 0.5 pA; V55E 5.8 ± 0.5 pA (mean ± SD). **(f)** Histograms as in e from openings at 200 mV (WT, 136 openings; V55E, 106 openings). Mean unitary currents for these openings were: WT 14.2 ± 0.9 pA; V55E 10.0 ± 1.3 pA (mean ± SD).

by T59 (Fig. 4 a). Indeed, when T59 CA–CA distance (i.e., the width of the S4 ion binding site) is used as an opening coordinate, currents in V55E show a similar trend upon S4 opening as in WT (Fig. 4 c). In contrast however, the maximum outward current in V55E at 300 mV (~10 pA) is markedly lower than the WT

maximum current (~16 pA), suggesting that an additional energetic barrier for ion permeation (apart from the one imposed by T59) exists somewhere along the permeation pathway.

In simulations of MthK V55E, we observed that the E55 side chain adopts two distinct rotameric states—one that is seen in

crystal structures of KcsA, where the (protonated) side chain of E55 (E71 in KcsA) is hydrogen-bonded with the side chain of D64 (D80 in KcsA), behind the SF (vertical orientation of E55, Fig. 4 d). In the second orientation, this hydrogen bond is broken, and instead the side chain of E55 adopts a more in-plane orientation with respect to the membrane (horizontal orientation of E55). We suspected that this rotameric transition might have a large influence on the conducting properties of the SF. For a clearer picture, we performed separate simulations with the E55 side chain weakly restrained to one of the two orientations (simulation sets "E55 vertical" and "E55 horizontal"). These simulations show that the orientation of the E55 side chain has a major effect on the outward current, and on its dependence on both gate and S4 opening (Fig. 4, e and f). In E55 vertical simulations, the currents are markedly lower than in the non-restrained case, at low and moderate activation gate openings, and only at larger openings reach the level of 7–8 pA (Fig. 4 e, brown traces). Furthermore, the current does not decrease at very large openings. The reason for that is clear when the S4 width is used as the opening coordinate (Fig. 4 f, brown traces): in the E55 vertical simulations, the S4 site does not get wider than ∼0.88 nm, showing that the activation gate S4 coupling is restricted when the E55 side chain adopts the vertical orientation, in contrast to non-restrained simulations of V55E (and WT, see Fig. 4 c). In E55 horizontal simulations, the current reaches its maximum value at small activation gate openings of ∼1.43 nm, in contrast to 1.6–1.65 nm in the non-restrained case (and WT; Fig. 4 e, light orange traces). Interestingly, the S4 width in E55 horizontal is identical to MthK WT, at openings corresponding to the maximum current. In other words, the horizontal orientation of the E55 side chain affects the activation gate—S4 coupling: the S4 binding site is able to open wider at smaller activation gate openings. Note however, that the restraints used were sufficiently mild, that there is still a significant portion of vertically oriented E55 side chains in these simulations (Fig. 4 g). The maximum current in the E55 horizontal simulations reaches the value of MthK WT, further demonstrating that MthK WT and V55E can, in principle, conduct ions at a similar rate. Overall, currents in non-restrained MthK V55E are affected not only by the activation gate opening (as in WT) but additionally also by the rotameric state of the E55 side chain (Fig. 4 e, red traces). We next sought to determine why in these simulations are V55E currents reduced compared to MthK WT or E55 horizontal simulations, even though in all these cases the optimal width of the S4 site is reached (∼0.86 nm; Fig. 4, f and h). To gain additional insights, we calculated negative logarithms of potassium densities in the SF that might be treated as approximate, one-dimensional free energy profiles for ion permeation (Fig. 4 i). It is important to note that such profiles do not reflect the underlying multi-ion permeation process, including permeation barriers. They indicate the regions in the SF where ion binding is favored and disfavored. A comparison between profiles for MthK WT vs. MthK V55E (black curve vs. solid brown and light orange curves in Fig. 4 i) reveals that the barriers are reduced at the entry to the SF, for both orientations of the E55 side chain (i.e., between Scav and S4, and between S4 and S3). However, in E55 vertical

simulations, the barrier between S1 and S0 is increased compared to WT (both brown curves). Consequently, the width of the SF between S1 and S0 (i.e., the CA–CA distance of Y62) does not show any variation in E55 vertical simulations, being restricted to the value of ∼0.79 nm, in stark contrast to both MthK WT and E55 horizontal simulations (Fig. 4 j). Interestingly, this value lies between those observed in MthK WT and E55 horizontal simulations, suggesting that this particular value leads to a very strong binding of a $K^+$ ions to S2. Indeed in E55 vertical simulations, the occupancy of S2 by $K^+$ ions is > 0.95, compared to ∼0.82 in WT. In V55E horizontal simulations, at an opening that shows the maximum current, the free energy profile is quite flat with low barriers, consistent with the observed high current (Fig. 4 i, light orange curve). The profile at a larger opening (dashed curve) reveals the effect of the horizontal orientation of E55 on S4 in the AMBER force field: the S4 binding site is compromised (a free energy minimum is replaced by a maximum), which likely results in two other high barriers, namely, between the S3 and S2 as well as S2 and S1 ion binding sites, drastically decreasing the current at larger openings in E55 horizontal simulations.

### Shorter in silico electrophysiology—CHARMM36m force field

Subsequently, we simulated the same systems with the CHARMM36m force field (referred to as CHARMM, see Materials and methods for details), to see whether the previous observations are force field dependent. In our previous study of MthK WT, both force fields showed the same trend in the current dependence on the lower gate opening, as well as very similar maximum currents (Kopec et al., 2019). Overall, simulations of V55E done with the CHARMM force field show little currents (up to 1.5 pA, i.e., more than an order of magnitude smaller than in WT) at all levels of the lower gate opening (Fig. 5 a). The S4 does however become wider in V55E, similarly to WT (Fig. 5 b), suggesting that a large free energy barrier for ion permeation exists somewhere else in the SF. In contrast to AMBER simulations, where we observed two rotamers of the E55 side chain, in CHARMM the E55 side chain adopts almost exclusively the vertical orientation (Fig. 5 c, red traces). Initial tries of inducing the horizontal orientations were unsuccessful; we then used some of the horizontal orientations seen in AMBER for further simulations with the CHARMM force field and imposing strong position restraints (Fig. 5 c, light orange traces). For completeness, we also simulated a system with distance restraints present in the vertical orientation (Fig. 5 c, brown traces). This simulation set, E55 vertical, behaves almost identical as the unrestrained case, indicating that the application of restraints has negligible effects on the overall channel function. Similar to simulations done with the AMBER force field, when the E55 side chain adopts the horizontal orientation in the CHARMM force field, the channel is able to permeate $K^+$ ions as efficiently as WT (Fig. 5 d). In CHARMM, however, that happens at the same gate opening level as in WT, which somewhat unexpectedly corresponds to slightly larger openings of the S4 in E55 horizontal (Fig. 5, e and f). These observations suggest that the free energy barrier imposed by T59 is not always the dominating one in CHARMM. Indeed, in the WT channel, there is a

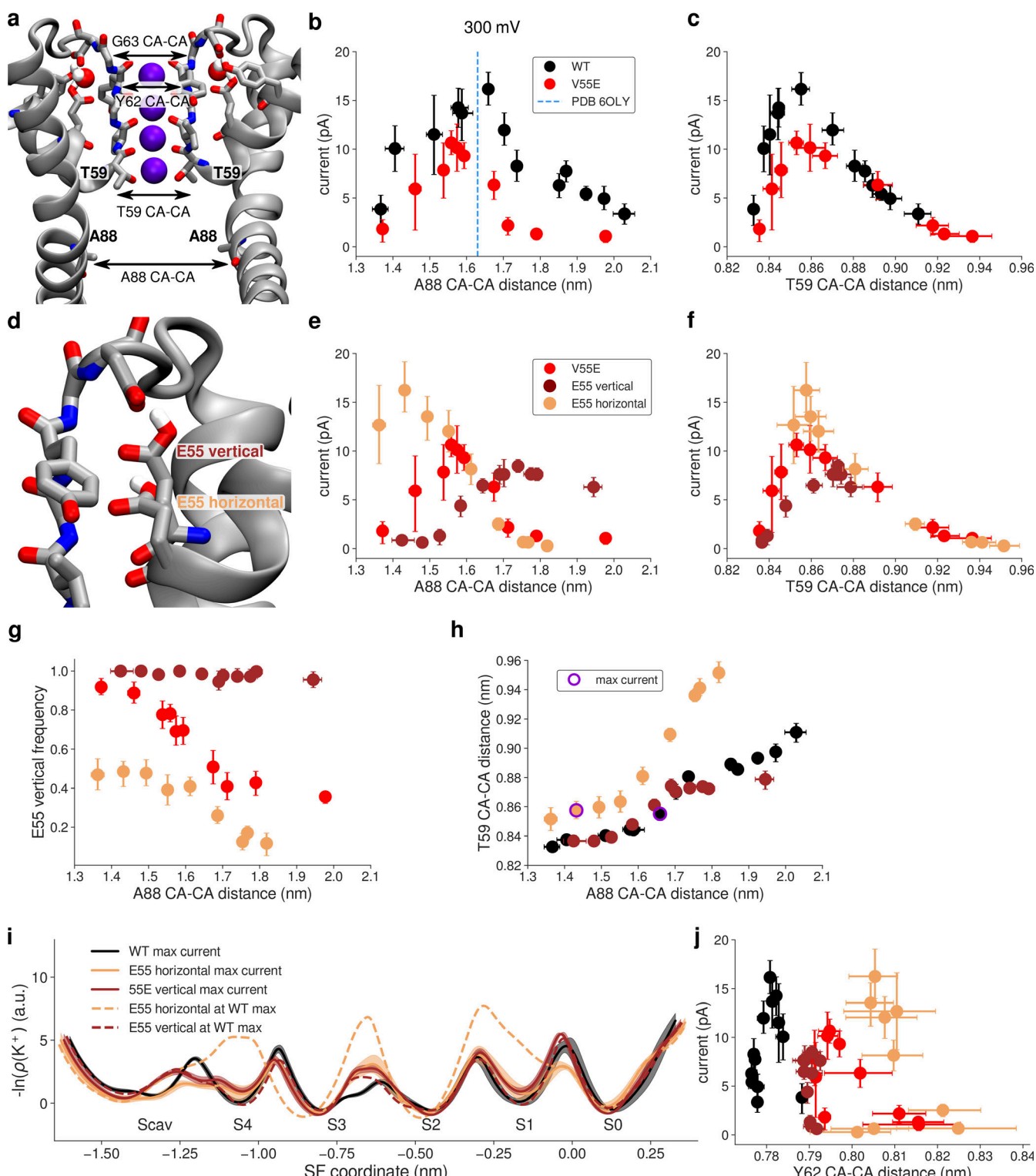

Figure 4. **Outward currents in MthK V55E in MD simulations with the AMBER force field. (a)** Starting structure of MthK V55E and the definition of distances used as opening coordinates (A88 CA–CA distance: activation gate; T59 CA–CA distance: S4 width). **(b)** Outward current through MthK WT (black traces) and MthK V55E (red traces) as a function of the activation gate opening. Blue vertical line marks the largest opening level seen experimentally. **(c)** Same as b but with the S4 width used as the opening coordinate. **(d)** Two rotameric states (orientations) of E55 observed in MD simulations. **(e)** Outward current in E55 vertical (brown traces) and E55 horizontal (light orange traces) simulation sets in comparison to current in V55E (red traces), as a function of the activation gate opening. **(f)** Same as e but with the S4 width used as the opening coordinate. **(g)** Frequency of the vertical rotameric state of E55 as a function of the activation gate opening. **(h)** Correlation between S4 width and the activation gate opening in E55 vertical and E55 horizontal in comparison to WT. Points for which the maximal current of ~16 pA was recorded are indicated by a violet ring. **(i)** Negative logarithmic densities ("free energy profiles") of K⁺ ions in the SF. Minima correspond to stable ion binding sites and maxima to free energy barriers between them. **(j)** Currents in all systems with the Y62 (forming S1 and S0 sites) CA–CA distance used as an opening coordinate. For all panels, error bars represent 95% confidence intervals.

Figure 5. **Outward currents in MthK V55E in MD simulations with the CHARMM force field. (a)** Outward current through MthK WT (black traces) and MthK V55E (red traces) as a function of the activation gate opening. Blue vertical line marks the largest opening level seen experimentally. **(b)** Same as a but with the S4 width used as the opening coordinate. **(c)** Frequency of the vertical rotameric state of E55 as a function of the activation gate opening. **(d)** Outward current in E55 vertical (brown traces) and E55 horizontal (light orange traces) in comparison to current V55E (red traces), as a function of the activation gate opening. **(e)** Same as d but with the S4 width used as the opening coordinate. **(f)** Correlation between S4 width and the activation gate opening in E55 vertical and E55 horizontal in comparison to WT. Points for which the maximal current of ~20 pA was recorded are indicated by a violet cross. **(g)** Negative logarithmic densities (free energy profiles) of K ions in the SF. Minima correspond to stable ion binding sites and maxima to free energy barriers between them. **(h)** Current in all systems with the G63 (forming S0 site) CA–CA distance used as an opening coordinate. For all panels, error bars represent 95% confidence intervals.

barrier of a similar height located between S2 and S1 binding sites (Fig. 5 g, black traces). This barrier becomes dominant in E55 horizontal simulations, even though the barrier between S4 and S3 got lowered further, explaining the same maximal current in both WT and E55 horizontal simulations.

The remaining question is, again, why is ion conduction in unrestrained V55E and E55 vertical simulations so low (max. 1.5 pA in the CHARMM force field)? The free energy profiles (Fig. 5 g, brown traces) show several very high barriers, as well as severely affected Scav and S0 ion binding sites. The lack of occupancy at the Scav binding site can be explained by high ion occupancy at S4, which would electrostatically repel ions approaching Scav. However, the lack of the S0 binding site is more

puzzling, also because it was preserved in AMBER simulations. The analysis of CA–CA distances of S0-forming G63 (Fig. 5 h) reveals that the vertical orientation of the E55 side chain makes S0 too narrow, and in consequence it likely cannot bind $K^+$ ions anymore. Only upon the E55 side chain rotation toward the horizontal orientation, S0 becomes as wide as in MthK WT, which correlates with the increased current (Fig. 5 h, black and light orange traces). Summarizing, CHARMM and AMBER force fields show marked differences in the exact behavior of the MthK V55E channel and in ion permeation through it. However, they are in a broad accord in three main points: (1) ion permeation in MthK V55E depends not only on the coupling between the activation gate and S4 (as is does in WT channels), but also

on the rotameric state of the E55 side chain; (2) the vertical E55 orientation, i.e., hydrogen-bonded with both D64 and Y51 (Fig. 1 e), reduces channel conductance, by imposing additional free energy barriers for ion permeation in the SF; and (3) the horizontal orientation of E55 increases currents in V55E to WT-like levels, by reducing these free energy barriers, especially in the upper part of the SF. The reduced outward conductance of MthK V55E, as compared to MthK WT, predicted by both force fields, is in very good agreement with our electrophysiological unitary conductance measurements (Fig. 3). The major difference between the force fields in this set of simulations is the fact that in AMBER both vertical and horizontal orientations of the E55 side chain are frequently visited (Fig. 4 g), whereas CHARMM has a strong preference for the vertical orientation (Fig. 5 c). This, in turn, results in very different simulated outward currents when no restraints on the E55 side chain are used, because, as shown above, the orientation of this side chain is a major factor regulating ion permeation. The origins of this distinct behavior, i.e., why the horizontal orientation is frequently visited in the AMBER force field, but not in CHARMM, are not obvious. It is unlikely, however, that this difference arises from variations in the non-bonded interactions. Even though the partial charges and Lennard-Jones parameters are different between AMBER and CHARMM, the resulting interaction energies of the D64–E55 hydrogen bond (E55 in the vertical orientation) are actually quite similar and favorable (Fig. S1, b and c). However, we suspected that the energetics of the two dihedrals, that involve the hydrogen atom from the protonated E55 side chain, might also play a role (Fig. S1 d). Indeed, these two dihedrals are low in energy (0–2 kJ/mol) in the vertical orientation in the CHARMM force field (Fig. S1 e). In contrast, however, they are energetically less favorable in the same orientation in the AMBER force field, as one of the dihedrals has an energy of ∼17–18 kJ/mol (Fig. S1 f). Only after the transition to the horizontal orientation, the energy drops to values close to 0 kJ/mol, likely explaining the frequent transition from vertical to horizontal orientation in the AMBER force field (and lack of thereof in CHARMM). Further research is needed to unravel which of these force field parametrizations describe the E55 dynamics more realistically. The orientational preference of E55 side chain in AMBER (but not in CHARMM) is also dependent on the activation gate opening (Fig. 4 g), which we attribute to the allosteric coupling between the gate and the SF, and the residues nearby (Kopec et al., 2019; Lewis et al., 2021): a variation in the activation gate width affects the SF width as well, which in turn might impact the delicate balance between non-bonded interactions and torsional energetics, dictating the overall E55 side chain dynamics.

## Longer in silico electrophysiology simulations

Apart from faster protein motions occurring on the tens and hundreds of nanoseconds timescale studied above, we also investigated the SF gating (inactivation), i.e., conformational transitions of the SF, with MD simulations. Even though in reality the timescales of such transitions are at least in the millisecond range (Cuello et al., 2010), way above the standard MD range, they have nevertheless been observed in simulations in a few cases (Li et al., 2018; Li et al., 2021). Structures of KcsA,

stably inactivated mutants of voltage-gated K+ channels (*Shaker* W434F and Kv1.2 W362F) and WT Kv1.3 channels revealed that C-type inactivation-related conformational changes at the SF can follow at least two distinct paths (Fig. 6; Reddi et al., 2022; Zhou, 2001; Tan et al., 2022; Selvakumar et al., 2022): in KcsA the SF pinches (constricts) at the level of the "first" glycine (G77, corresponding to G61 in MthK), which is accompanied by flipping of valine (V76 in KcsA, corresponding to V60 in MthK) carbonyl away from the SF axis. The SF also loses two K+ ions during inactivation, and has remaining ions bound at S1 and S4, whereas S3 is occupied by a water molecule, which presumably enhances the valine flipping. In voltage-gated channels, the SF dilates (widens) at the level of the "second" glycine (G446 in *Shaker*, G374 in Kv1.2, G448 in Kv1.3, corresponding to G63 in MthK). In this case, this SF dilation is accompanied by a breakage of hydrogen bonds between aspartate (D447 in *Shaker*, D375 in Kv1.2, D449 in Kv1.3, corresponding to D64 in MthK) and tyrosine/tryptophan (W434 in *Shaker*, W362 in Kv1.2, W436 in Kv1.3, corresponding to Y51 in MthK), which is replaced by a phenylalanine in stably inactivated mutants, and flipping of aspartate side chains toward the extracellular side. In contrast, these hydrogen bonds are retained during SF inactivation in KcsA and the aspartate (D80) flipping has not been observed (Fig. 6 b). In *Shaker* and Kv1.2, two K+ ions remain bound to the SF in its inactivated conformation, but they occupy S3 and S4 ion binding sites (in contrast to inactivated KcsA), and S2 is occupied by a water molecule. In Kv1.3, there is a third K+ ion bound at the S1 binding site. Consequently, the valine flipping has not been observed in structures of inactivated Kv channels, because S3 is occupied by a K+ ion. Previous works on KcsA revealed that the SF pinching occurs in unbiased MD simulations on the microsecond timescale with the CHARMM force field, and the SF adopts the crystal structure-like, inactivated conformation (PDB ID 1K4D; Li et al., 2018). Our previous work on the Kv1.2 channel had predicted aspartate flipping, SF dilation, and loss of K+ ions from S0 to S2, also using the CHARMM force field (Kopec et al., 2018), before recent cryo-EM structures of stably inactivated *Shaker* W434F, Kv1.2 W362F, and Kv1.3 channels became available (Fig. 6, c–f; Reddi et al., 2022; Tan et al., 2022; Selvakumar et al., 2022). In MthK, previous MD simulations done with the CHARMM force field (and reduced interaction strength between K+ ions and carbonyl oxygens) revealed that K+ ions from sites S1, S2, and S3 need to unbind before the SF collapse, which was then structurally similar to the one seen in KcsA, having a constriction at the level of the first glycine (G61), V60 and G61 carbonyl flipping and increased number of water molecules behind the SF (Boiteux et al., 2020). Here, we probed the stability of the SF of both MthK WT and the V55E in 20 independent, 5-μs long unbiased MD simulations per force field/system, under two applied voltages (300 and 150 mV), resulting in 800 μs of sampling, and at a large activation gate opening, which speeds up SF conformational transitions (Li et al., 2018; Li et al., 2021). We focused on whether the hydrogen bonds between D64 and Y51 are retained or broken (Figs. S2, S3, S4, S5, S6, S7, S8, and S9, the distance larger than ca. 0.7 nm indicates a broken interaction and a subsequent aspartate flip), and on the changes in the SF geometry, at the level of both first and second glycines

(G61 and G63; Figs. S10, S11, S12, S13, S14, S15, S16, S17, S18, S19, S20, S21, S22, S23, S24, and S25, respectively). In many of our simulations, the SF of both MthK WT and V55E undergoes dilation and/or constriction transitions at the level of G63 (and also G61), which is preceded by at least one broken D64–Y51 interaction (aspartate flip; Fig. 6, g–i; and Figs. S2, S3, S4, S5, S6, S7, S8, and S9). Specifically, this aspartate D64 flip has been shown as a central event underlying SF inactivation in voltage-gated channels, leading to the widening of the SF at the second glycine (Fig. 6, c–f). Therefore, we used G63 CA distance variations as a coordinate defining conducting and inactivated filter conformations (Figs. S18, S19, S20, S21, S22, S23, S24, and S25). Based on this coordinate, we counted the conducting to non-conducting (inactivated) transitions occurring in independent simulations (Fig. 6, j–m), at two different voltages (150 and 300 mV). On the MD simulations timescale used here, these transitions occur irreversibly, i.e., maximally once per simulation. In the AMBER force field, we observed 7/20 such "inactivation transitions" for V55E at 300 mV and 4/20 at 150 mV, whereas it did not occur at all for WT at either voltage. In the CHARMM force field, V55E transitioned to the inactivated state in 19/20 simulations at 300 mV, and 13/20 at 150 mV, whereas only 7/20 inactivation transitions at 300 mV, and 4/20 at 150 mV, were observed for WT. These simulations clearly show that MthK V55E has an increased propensity to enter the inactivated conformation as compared to WT, in both force fields. Moreover, the inactivation rate is increased at higher voltages, in agreement with electrophysiology (Fig. 2). Thus, in MD simulations, V55E reduces the stability of the conducting conformation of the SF and promotes inactivated conformations, which is in very good agreement with the decreased lifetimes of the open conformation seen in electrophysiological recordings (Fig. 2). These inactivated conformations of the SF, in both MthK WT and V55E, structurally resemble the conformations of stably inactivated mutants of Shaker and Kv1.2 channels, as well as WT Kv1.3 channels (Fig. 6, g–i). However, they also share some similarities with filters of inactivated KcsA and MthK at a low potassium concentration, namely, flipping of carbonyl groups of V60 and G61, due to water entry into the SF, as well as increased number of water molecules behind the SF (Fig. 6, g–i; and Fig. S26). Overall, the inactivated SFs in our simulations are quite dynamic and adopt a wide range of conformations unified however by the presence of D64 flips, widened upper parts, flipped V60 carbonyls, and a lower occupancy of the S0 ion binding site (Fig. S27). To gain more insights, we focused on molecular events happening in these MD simulations, in which D64 flipping and subsequent SF inactivation occurred, in both MthK WT and Mthk V55E (Fig. 7). We specifically looked at short MD trajectory fragments, just before the D64 flipping events, because, as argued before, it is an essential feature of the SF inactivation, both in our simulations and in the available structures of Kv channels. We observed that in MthK WT (for which inactivation occurred only in the CHARMM force field), there is an increased probability of water molecules entering the region behind the SF just before the aspartate flip (Fig. 7 b, red curve), which likely affects the stability of the Y51–D64 hydrogen bond. As explained above, this hydrogen bond is conserved in many K+ channels and

stabilizes the D64 side chain in the non-flipped orientation (Fig. 7 a). At the same time, there is a slight increase in the probability of water molecules entering the SF (yellow curve) and a slight decrease in the occupancy of the top ion binding sites by K+ ions (purple curve). A similar overall pattern is observed in simulations of MthK V55E with the same force field (Fig. 7, c and d), however, the probability of water molecules entering behind the SF is higher throughout the 50 ns period prior to the aspartate flip than in MthK WT (Fig. 7 b). We attribute this effect to the presence of a hydrophilic, protonated glutamate side chain, which has a higher propensity to attract water molecules than (hydrophobic) valine present in MthK WT. The E55 side chains adopt mostly the vertical orientation (Fig. 7 d, brown curve), however, the increased probability of ~0.2 (from ~0.1) of the E55 side chain to adopt the horizontal orientation means that it occurs quite often in one of the monomers (on average, see the caption of Fig. 7), which is most likely correlated with the entry of additional water molecules behind the SF, that breaks the Y51–D64 hydrogen bond and affects the E55 side chain dynamics (Fig. 7 c). There are no major differences in the probability of water molecules entering the SF between MthK WT and V55E, in the CHARMM force field. In simulations of MthK V55E with the AMBER force field (Fig. 7, e and f), there is a similar, high probability of water molecules entering behind the SF, and almost all four E55 side chains adopt the horizontal orientation (brown curve) at all times. Most strikingly however, there is a sharp drop in the probability of K+ occupying S0 and S1 ion binding sites (purple curve), just before the D64 flipping event. This suggests that K+ ions bound to the top of the SF stabilize the conducting conformation, in the AMBER force field, with the non-flipped D64 side chains. Accordingly, control calculations in simulations of MthK WT with the AMBER force field (Fig. S28), where the D64 flipping does not occur, show a consistent occupancy of the S0 and S1 by K+ ions. Summarizing, we discovered that the V55E mutation reduces the stability of the conducting conformation of the SF in the MthK channel, by promoting water entry behind the SF, leading to the breakage of the Y51–D64 hydrogen bond and subsequent flipping of the D64 side chains. This entry of additional water molecules likely correlates with the horizontal orientation of the E55 side chain. Although both force fields predicted an overall similar effect of the V55E on the SF stability, there are important differences. Firstly, as already described for shorter simulations, the AMBER force field strongly prefers the horizontal orientation of the E55 side chain, due to the differences in dihedral parametrization (Fig. S1). Secondly, the distribution of K+ ions in the SF and the effect of the mutation on the overall number of K+ ions in the SF vary between the force fields (Fig. S27). Importantly, the D64 flipping in MthK V55E simulated with the AMBER force field strongly correlates with the unbinding of K+ ions from the top of the SF. Thirdly, the same aspartate flipping and inactivation mechanism as in MthK V55E is seen in simulations of MthK WT, but only when the CHARMM force field is used. We further investigated the possible reasons for the last discrepancy. We evaluated two factors, possibly affecting the D64 side chain dynamics: (1) the strength of the Y51–D64 hydrogen bond (weaker interactions would

promote the aspartate flip; Fig. S29, a–c); and (2) the energetics of dihedrals defined by the heavy atoms of the D64 side chain (high energy orientations of the D64 side chains would contribute to a lower stability of a given state; Fig. S29, d–f). From this analysis, the differences between the two force fields become more clear—surprisingly, the Y51–D64 interactions are actually stronger in CHARMM than in AMBER (opposed to the inactivation rate trend). Strikingly however, one of the two dihedrals of the D64 side chain has a high energy (up to ca. 19 kJ/mol) in the CHARMM force field, when the D64 side chain is in the non-flipped orientation, and is able to adopt a lower energy conformation only after the aspartate flip (Fig. S29 e). In stark contrast, in AMBER simulations, both dihedrals are low in energy when the D64 side chain is in the non-flipped orientation (Fig. S29 f). These force field differences, primarily in the dihedral parametrization, resulting in different preferred orientations of the D64 side chain, and, to some degree, in the K$^+$ occupancy of the individual ion binding sites (high S0 occupancy in AMBER; Fig. S27), likely explain the much higher propensity of the MthK WT SF inactivation in MD simulations with the CHARMM force field. Interestingly, these inactivated conformations resemble experimentally solved structures of Kv channels (Fig. 6). We wondered how these two factors are affected by the V55E mutation. The energetics of the dihedrals is only slightly affected, in both force fields (Fig. S30), although the high energy dihedral in CHARMM is even higher in energy in MthK V55E. The energetics of the Y51–D64 and E55 (or water in MthK WT)–D64 hydrogen bonds is also affected, especially in the CHARMM force field: most of these interactions are stronger in MthK WT than in the V55E mutant (Fig. S30 c), which could also contribute to the lower stability of the conducting state in MthK V55E in this force field. In the AMBER force field, the trend is not so clear: some interactions are stronger in MthK WT and some in MthK V55E. Therefore, in MD simulations of MthK V55E with the AMBER force field, we assign the unbinding of K$^+$ ions from the top of the SF, and the entry of additional water molecules behind the SF as the primary cause of the aspartate flipping and SF inactivation.

### Free energy calculations—E55 pKa

In our MD simulations, we have repeatedly observed (mostly in the AMBER force field) transitions of the E55 side chain from the vertical orientation, where it closely interacts with D64, to the horizontal orientation, without a clear interaction partner, and instead interacting with a water molecule (or causing an entry of a larger number of water molecules). The question arises whether the E55 side chain would still stay protonated in the horizontal orientation. This putative deprotonation of the E55 side chain would presumably have a substantial effect on all structural observables in MD simulations. Therefore, to assess the likelihood of the E55 side chain to deprotonate, we used nonequilibrium free energy calculations to calculate the ΔG of E55 deprotonation and consequently its pKa. (Fig. S31, see Materials and methods; Awoonor-Williams and Rowley, 2016; Gapsys et al., 2015; Gapsys et al., 2019; Bastys et al., 2018). Free energy results in both force fields are in good agreement, showing the ΔG of E55 deprotonation of >40 kJ/mol (with respect to the

reference state in bulk), indicating that it is unlikely for E55 to be in the deprotonated form. In fact, such a value translates to a shift of the pKa (ΔpKa) of seven to eight units from the glutamate experimental value of 4 (Thurlkill et al., 2006). Even taking into account possible force field and simulation inaccuracies, these results strongly suggest that E55 stays protonated at all times.

## Discussion

In this work, we set out to better understand the molecular basis for SF gating in MthK WT and the KcsA-like mutant MthK V55E by combining electrophysiological measurements with large-scale MD simulations. Previously, it was established that MthK WT exhibits SF gating at depolarized voltages, similar to C-type inactivation described in KcsA and Kv channels (Thomson and Rothberg, 2010; Thomson et al., 2014). Here, both electrophysiology and computational methods showed that in MthK V55E, the open state of the channel is destabilized. Gating of V55E in experiments is characterized by markedly reduced Popen over a wide range of transmembrane voltage (–200 to +200 mV), and at high potassium concentration (+200 mM) at both sides of the membrane, as opposed to MthK WT, in which the open state is destabilized depolarized voltages and/or reduced external potassium concentration (Thomson and Rothberg, 2010; Thomson et al., 2014). Moreover, V55E shows markedly lower single channel unitary conductance in electrophysiological experiments than MthK WT. Two in silico electrophysiology simulation-based approaches revealed how the V55E mutation affects ion permeation and the SF stability in MthK on the atomistic scale. In contrast to the KcsA crystal structure, where the side chain of this protonated glutamate is thought to exist only in one rotameric state (vertical orientation), hydrogen bonding with a conserved aspartate and crystallographically resolved water molecule behind the SF, our MD simulations suggest that in MthK V55E, the E55 side chain is mobile and can exist in at least two rotameric states, namely, vertical and horizontal. The strong interaction between E55 and D64 in the vertical orientation leads to decreased structural plasticity of the SF in its upper part, which increases energetic barriers for ion permeation (in the AMBER force field) and compromises the geometry of the S0 binding site (in the CHARMM force field), resulting in much lower simulated outward currents through MthK V55E. This strain in the SF is relieved when the E55 side chain adopts the horizontal orientation, and then the mutant channel is able to permeate K$^+$ ions at rates approaching those recorded in MthK WT. Therefore, the ionic current in MthK V55E is regulated not only via activation gate–SF coupling (as previously described for MthK WT; Kopec et al., 2019), but also by the E55 side chain rotameric state. The preference for the vertical vs. horizontal orientation also depends on the level of the activation gate opening, revealing a complex activation gate–selectivity filter–pore helix coupling. Further, the two molecular force fields used in this work also predicted a very different propensity of the E55 side chain to adopt the horizontal orientation: in the AMBER force field, it was actually the preferred orientation in many cases, whereas in the CHARMM force field, it was hardly

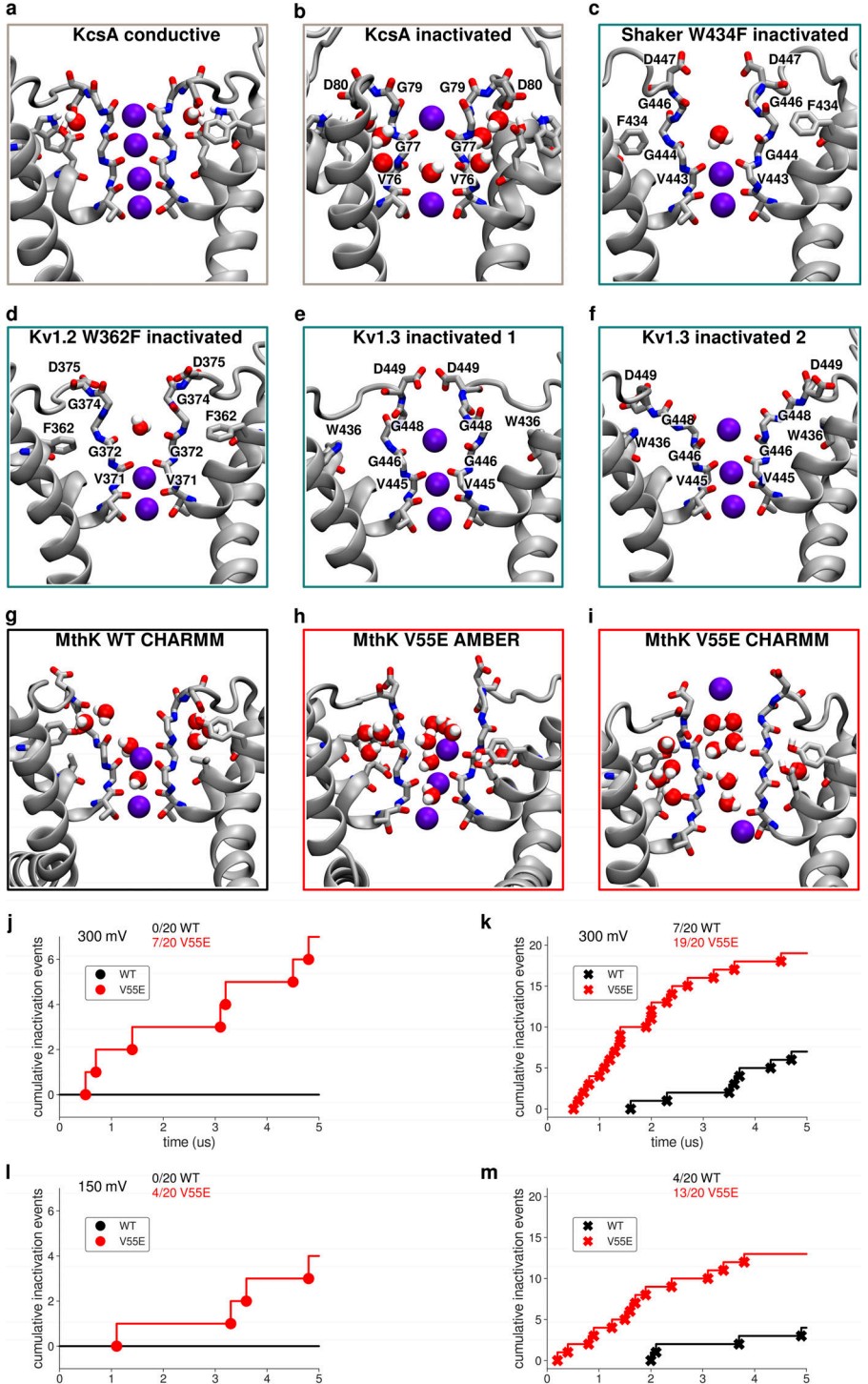

Figure 6. **Conformations of inactivated filters in different K⁺ channels. (a)** Typical conducting SF conformation shown for reference (PDB ID 1K4C). **(b)** Inactivated ("constricted" or "pinched") KcsA filter (PDB ID 1K4D). The filter narrows at the level of the first glycines (G77) and shows valine flipping at V76. **(c)** Inactivated ("dilated") filter of *Shaker* W434F. D447 side chains flip toward the extracellular side, and the SF is widened/dilated at the level of the second glycines (G446). **(d)** Inactivated (dilated) filter of Kv1.2 W362F. Similar to c D375 side chains are flipped, and the SF is widened. **(e and f)** Two (1 and 2) inactivated conformations of Kv1.3. Again, D449 side chains are flipped, and the SF is dilated. **(g–i)** Examples of SF conformations observed in this work in long simulations of either MthK WT (g) or MthK V55E (h and i) at 300 mV. In all cases, flipping of aspartates (D64) is observed, together with widening at the level of the second glycines (G63). Valine (V60) flipping is also observed as well. **(j–m)** Cumulative inactivation events recorded in simulations with AMBER (j and l) and CHARMM (k and m) force fields at 300 and 150 mV, respectively. The events were identified based on distances between G63 CA atoms in oppositely oriented monomers.

accessible, especially at the shorter timescales. Our longer simulations also allowed us to peek into larger conformational transitions of the SF, which we relate to the C-type inactivation process. Here, surprisingly, both MthK WT and MthK V55E show behavior more comparable with the stably inactivated Kv channels (*Shaker* W434F, Kv1.2 W362F, and WT Kv1.3), although some features seen with in KcsA, such as carbonyl flipping of V60 and G61 residues, as reported before (Boiteux et al., 2020), are present as well. These observations suggest that the

differences between MthK and KcsA channels are not only limited to the residue at the position 55 in MthK (valine in MthK WT, protonated glutamate in KcsA), and likely the variation in other residues in the pore helix play a major role as well in regulating the SF gating behavior of these channels. Indeed, the distribution of hydrophobic and hydrophilic residues varies in the pore region of both channels (Fig. S32). One major difference is the interaction partner of the aspartate behind the SF: in MthK, it is a tyrosine residue (Y51), whereas in KcsA it is a

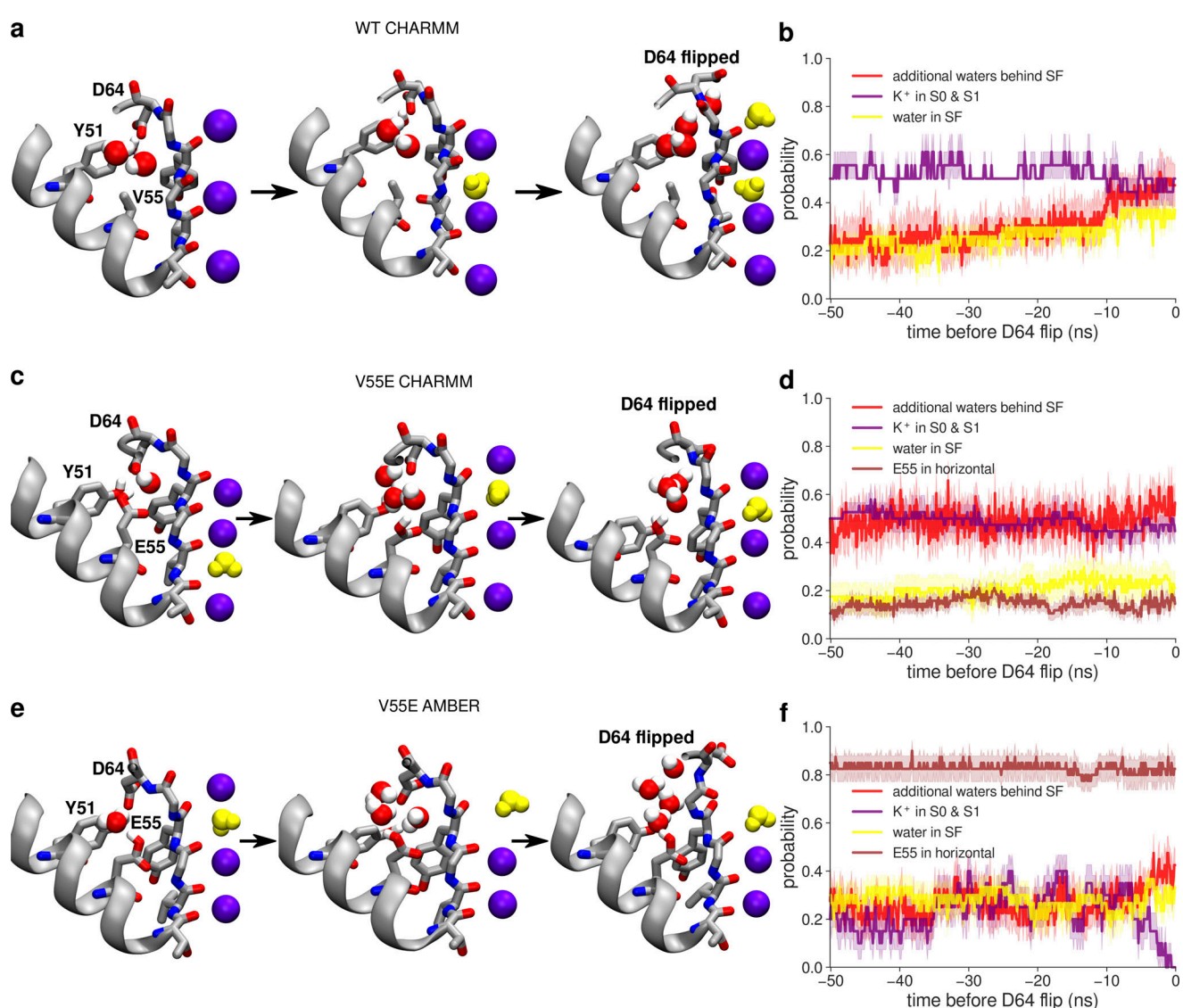

Figure 7. **Molecular events preceding the D64 flipping and filter inactivation in MD simulations of MthK WT and V55E channels. (a, c, and e)** Visualizations of the SF surroundings just before the D64 flip, in MthK WT simulated with the CHARMM force field, in MthK V55E simulated with the CHARMM force field, and in MthK V55E simulated with the AMBER force field, respectively. No D64 flips nor inactivations occurred in MthK WT simulated with the AMBER force field, therefore it is not included. Water molecules inside the SF are shown in yellow. **(b, d, and f)** Probabilities of specific events in 50-ns simulation time before the D64 flip: additional water molecules (to the crystallographic waters) entering the space between V/E55 and D64 behind the SF (red curve), K⁺ ions occupying ion binding sites S0 and S1 (purple curve), water occupying inner ion binding sites (S1–S3) in the SF (yellow curve), and E55 side chain adopting the horizontal orientation (brown curve). For calculating probabilities, all trajectories in which a D64 flip occurred were aligned at the time of the first D64 flip ($t = 0$) and the probability was at each frame over all of these trajectories. Note that for red and brown curves, the probability is calculated for all four monomers, i.e., the value 1 would mean an event occurring in all four monomers simultaneously. Error bars represent the standard error.

tryptophan residue (W67). These residues have different hydrophobicity and hydrogen bonding capabilities, therefore they might largely affect the behavior of the aspartate side chain, and subsequently its other interaction partner, namely, protonated glutamate (in KcsA and in MthK V55E). The double MthK Y51W, V55E mutant would be an interesting construct in the future work, to see if it shows a behavior even more related to KcsA. Another interesting difference is position 64 in KcsA (positively charged arginine), which in MthK is replaced by valine (V48). A positively charged residue near D80 might stabilize the aspartate in the non-flipped orientation. Residues that form the extracellular loop following the SF have also been postulated to play a

role in C-type inactivation in KcsA. For example, the entry of water molecules behind the SF in KcsA was proposed to occur through the Y82 "lid" residue (Cuello et al., 2017; Ostmeyer et al., 2013). MthK has a serine residue in this position, and tyrosine occurs one position earlier in the sequence. These variations in the extracellular loop might underlie different water penetration pathways in KcsA and MthK, again leading to the divergent dynamics of D80 and other residues behind the SF. Similar asymmetric SF conformations to ones reported in this work have been also observed in the structures of some two-pore domain (K2P) K⁺ channels: TREK-1 (Lolicato et al., 2020) and TASK-2 (Li et al., 2020). Specifically, we highlight that the conformational

transitions (inactivation) observed in our MD simulations of both MthK WT and V55E channels involve widening (dilation) of the SF at the level of second glycines (G63 in MthK), in agreement with available structural data for *Shaker* W434F, Kv1.2 W362F, and WT Kv1.3 channels (Reddi et al., 2022; Tan et al., 2022; Selvakumar et al., 2022). Our simulation data reveal that this dilation follows the breakage of the critical hydrogen bond behind the SF, between D64 and Y51. The strength of this hydrogen bond has been previously directly linked to the rate of inactivation in Kv channels in experiments employing non-natural amino acids (Pless et al., 2013), further highlighting the converging behavior of MthK and Kv channels with respect to SF gating. When this Y51–D64 hydrogen bond is broken (e.g., via the entry of additional water molecules behind the SF), the D64 side chain flips toward the extracellular side (aspartate flip) allowing the SF to widen. Interestingly, a combined experimental and computational study on the "MthK-like" KcsA E71V mutant showed a distinct mode of SF inactivation, without the filter constriction typical for KcsA WT, nor filter dilation, but rather with the overall rigidified filter (Rohaim et al., 2022). We note, however, that in the structures of KcsA E71V at large gate openings, the hydrogen bond between the aspartate and tryptophan (analogous to the Y51–D64 bond in MthK) was broken (Rohaim et al., 2022), although a full aspartate flip, with its side chain pointing toward the extracellular side, was not reported. On the other hand, the structures of (non-inactivating) KcsA E71A at low $K^+$ concentration show aspartate and valine flipping in the SF, akin to features reported in this work (Cheng et al., 2011). In our simulations, the widened SF displays compromised geometries of the ion binding sites, due to outward rotation of carbonyl oxygens as reported earlier (Thomson et al., 2014; Brennecke and de Groot, 2018; Cheng et al., 2011). Subsequently, the ion binding sites get filled with water molecules (Fig. 6, g–i; and Fig. S26), which disrupt the optimal ion permeation that normally occurs in a water-free fashion via direct ion–ion contacts (Mironenko et al., 2021). By directly simulating many inactivation events (160 individual simulations in total), we showed that this process occurs more frequently in MthK V55E than in the WT channel and is accelerated by more positive voltages, in full agreement with our electrophysiological recordings. Interestingly, our data suggest that the structural basis for reduced stability of the MthK V55E SF is, again, related to the orientation of the E55 side chain. As E55 participates in the intricate hydrogen bond network including D64 and Y55, the rotameric transition of its side chain from vertical to horizontal leads to the entry of additional water molecules behind the SF, further destabilizing D64–Y55 bond. In the horizontal orientation, the hydrophilic side chain of protonated E55 does not have any obvious hydrogen bond partners among protein residues to bond with, and it attracts additional water molecules from the extracellular side. Viral $K^+$ channels, which also exhibit exquisite SF gating, have hydrophilic threonine or serine residues in the position equivalent to E55 (Braun et al., 2014; Andersson et al., 2018), suggesting that a similar molecular behavior might be expected in those channels. We note major differences between the used force fields: CHARMM and AMBER, in their preferences regarding the vertical and horizontal orientations of the E55 side

chain, as well in the flipped and non-flipped orientations of the D64 side chain. We uncovered the origins of these differences being different parameterizations of dihedrals of these residues, and further work is needed to resolve this issue. It is interesting that the horizontal orientation of the E55 side chain (or E71 in KcsA) has not been seen before, neither in structural work nor in MD simulations. We suggest the following reasons for this situation. First, KcsA channels have been mostly simulated with the CHARMM force field (with some recent exceptions; Furini and Domene, 2020), in which the side chain of protonated E71 has a much lower propensity of adopting the horizontal orientation. Second, our analysis suggests that the horizontal orientation is favored for the larger openings of the activation gate. Most of the KcsA structures have been solved in the closed state, and in those solved in the open state, protonated glutamate is replaced by alanine (the E71A construct), to remove C-type inactivation. Third, as discussed above, MthK V55E and KcsA channels are quite different in their pore region (Fig. S32), thus, it is possible that the horizontal orientation is more common for MthK V55E than for KcsA. Our work is the first dealing with the MthK V55E channel, for which there is currently no structural data. We cannot, however, fully rule out a possibility that the high frequency of the horizontal orientation is an artifact of the AMBER force field, and we recommend further work in this direction. In our simulations, the inactivation process in the MthK channel, arising from SF gating, seems to be structurally related to the same process in Kv channels, but with some characteristic shared with KcsA. By combining electrophysiology and MD simulations, we showed that a single mutation in the pore helix (V55E) can simultaneously affect conducting properties and open state stability, enhancing the intrinsic propensity of WT channels to inactivate. We propose a mechanism of this effect based on a functionally relevant rotameric transition of a hydrophilic residue behind the SF, coupled with a water entry behind the SF and $K^+$ ion unbinding from the top of the filter. These results contribute to a conceptual framework for the effects of pore-helix residues on SF gating in various $K^+$ channels.

### Data availability
Exemplary molecular structures, input files and molecular dynamics trajectories, and the analysis jupyter notebook file with data underlying Figs. 4, 5, 6, and 7 are publicly available in Zenodo at https://doi.org/10.5281/zenodo.7986576. Electrophysiological recordings are available upon reasonable request from Brad S. Rothberg at brad.rothberg@temple.edu. All remaining MD trajectories and analysis scripts are available upon reasonable request from Wojciech Kopec at wkopec@gwdg.de.

### Acknowledgments
Crina M. Nimigean served as editor.

We thank Simon Bernèche and Florian Heer for many important insights and all three anonymous referees, whose comments greatly improved the manuscript, Vytas Gapsys for his help with the free energy calculations, and Lucie Delemotte for the critical reading of the revised manuscript.

This research was supported by the German Research Foundation (DFG) through FOR2518 "Dynion", Project P5 (to W. Kopec and B.L. de Groot) and National Institutes of Health grant R01 GM126581 (to B.S. Rothberg). Open Access provided by the Max Planck Society.

Author contributions: B.S. Rothberg, W. Kopec and B.L. de Groot conceived the project. A.S. Thomson and B.S. Rothberg performed and analyzed the electrophysiological recordings. W. Kopec performed and analyzed MD simulations. W. Kopec, B.S. Rothberg and B.L. de Groot wrote the manuscript.

Disclosures: The authors declare no competing interests exist.

Submitted: 29 March 2022

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

# Supplemental material

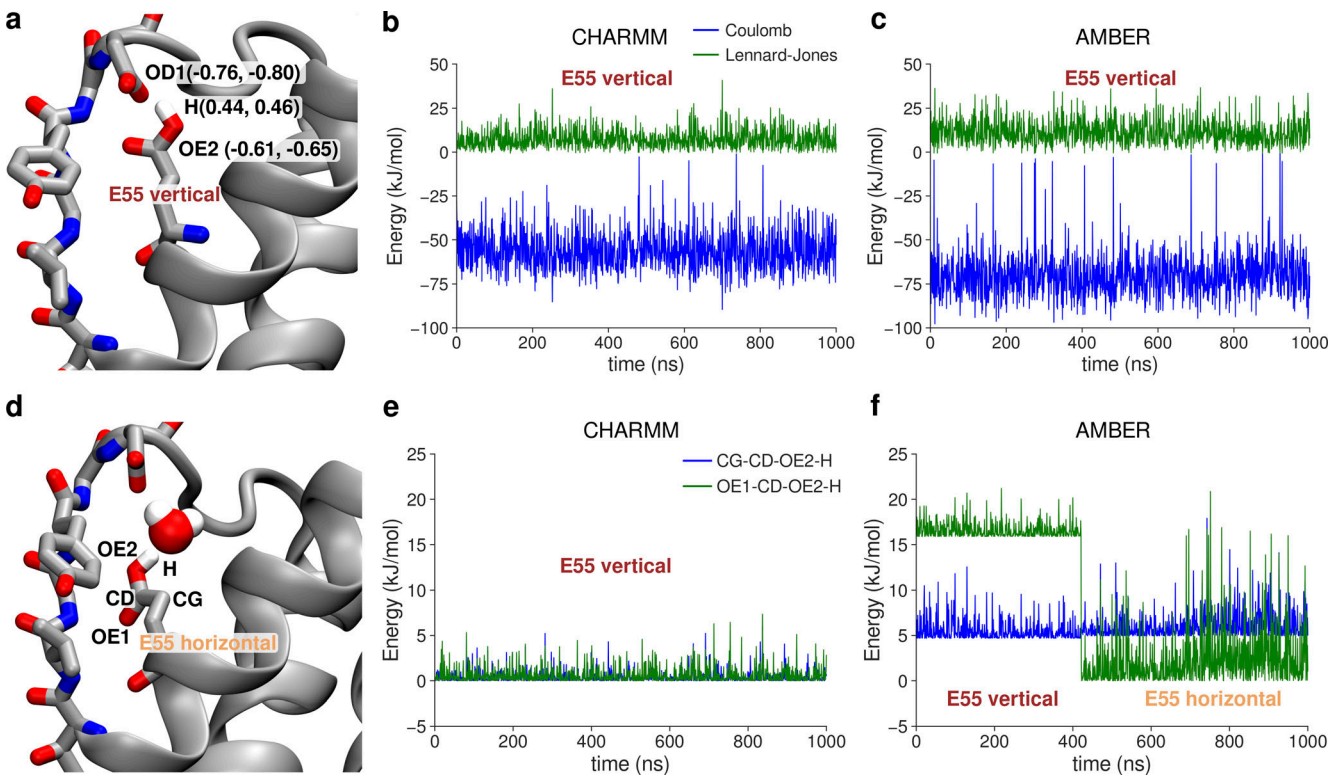

Figure S1. **Differences in energetics of the vertical vs. horizontal orientation of the E55 side chain in CHARMM and AMBER force fields. (a)** E55 side chain in the vertical orientation, making a hydrogen bond with D64. Partial charges of atoms participating in the hydrogen bond are given in parentheses, first the value for CHARMM and second for AMBER. **(b and c)** Exemplary interaction energies between hydrogen bond donors (H & OE2 atoms) and acceptor (OD1 atom), in both CHARMM and AMBER, respectively. **(d)** E55 side chain in one of the horizontal orientations visited in the AMBER force field. Atoms participating in two dihedral angles involving the H atom from the protonated E55 side chain are listed. **(e and f)** Exemplary torsional energy in a trajectory of two dihedrals involving the H from the E55 side chain, in both CHARMM and AMBER, respectively.

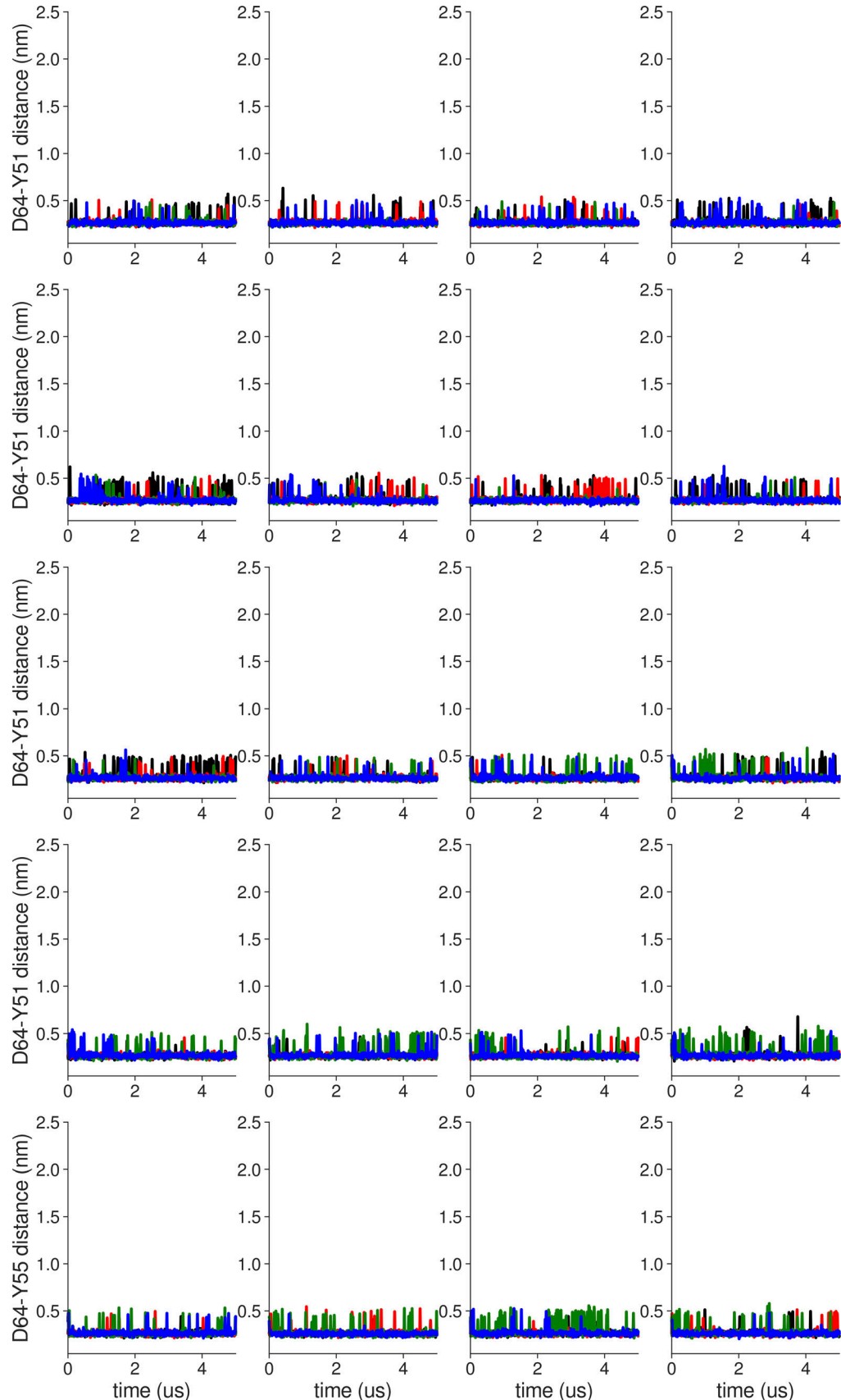

Figure S2.  **Individual distance traces between D64 CG atom and Y51 HH atom in long MthK WT AMBER simulations at 300 mV.** Each color refers to one of the channel monomers. Each panel shows traces from an independent, 5-μs long simulation. Distance values >1 nm indicate broken interactions.

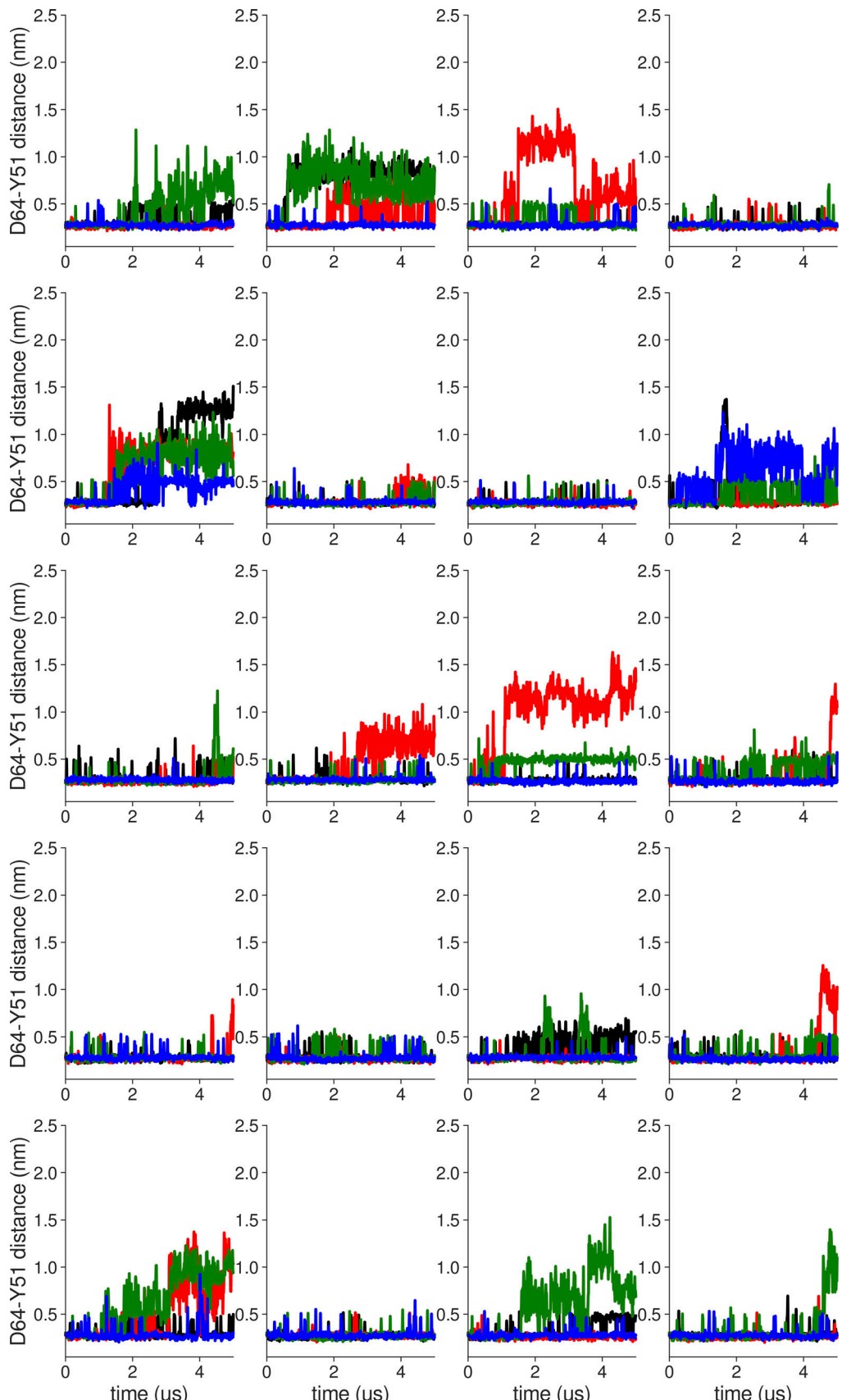

Figure S3.   **Individual distance traces between D64 CG atom and Y51 HH atom in long MthK V55E AMBER simulations at 300 mV.** Each color refers to one of the channel monomers. Each panel shows traces from an independent, 5-μs long simulation. Distance values >1 nm indicate broken interactions.

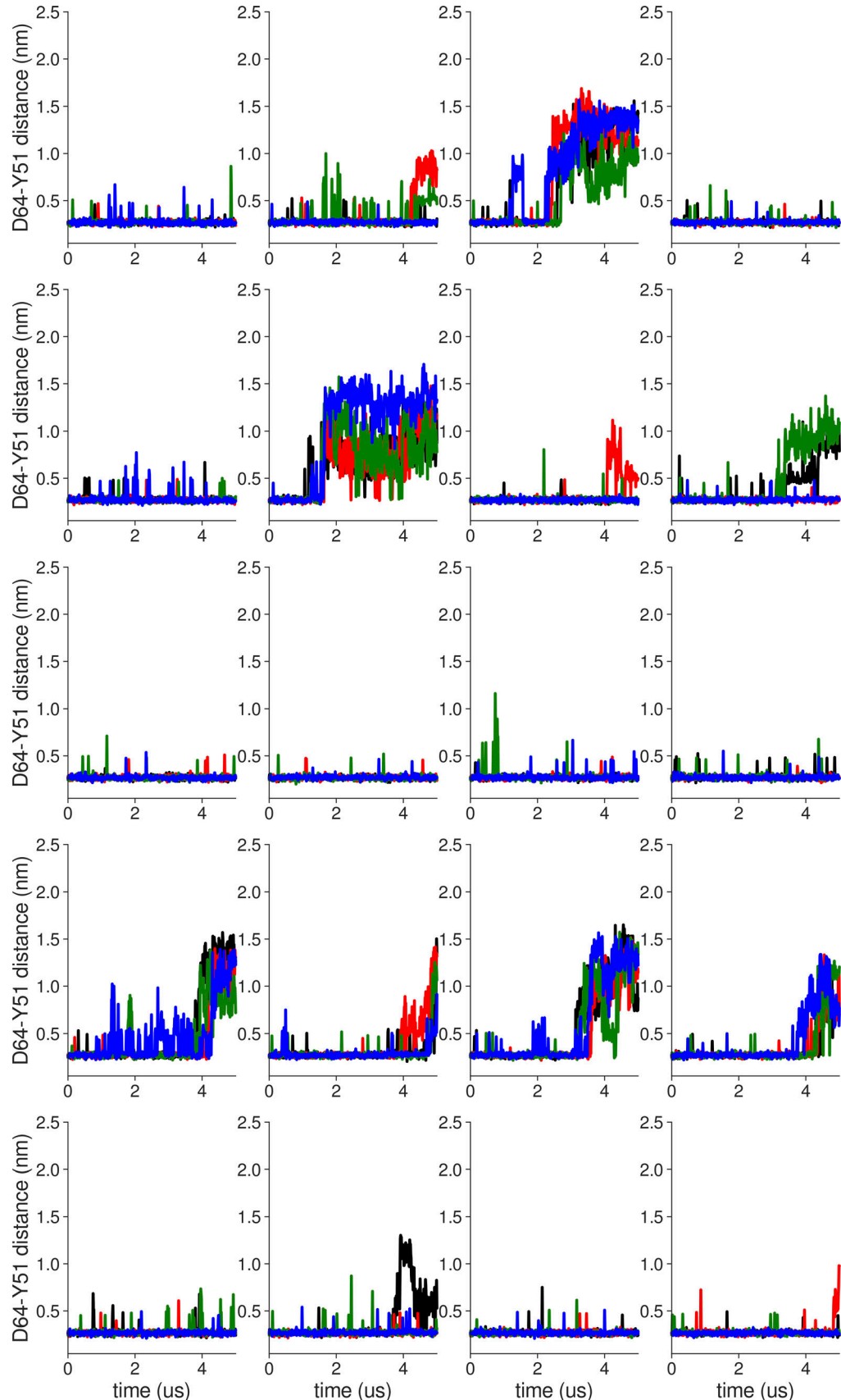

Figure S4. **Individual distance traces between D64 CG atom and Y51 HH atom in long MthK WT CHARMM simulations at 300 mV.** Each color refers to one of the channel monomers. Each panel shows traces from an independent, 5-μs long simulation. Distance values >1 nm indicate broken interactions.

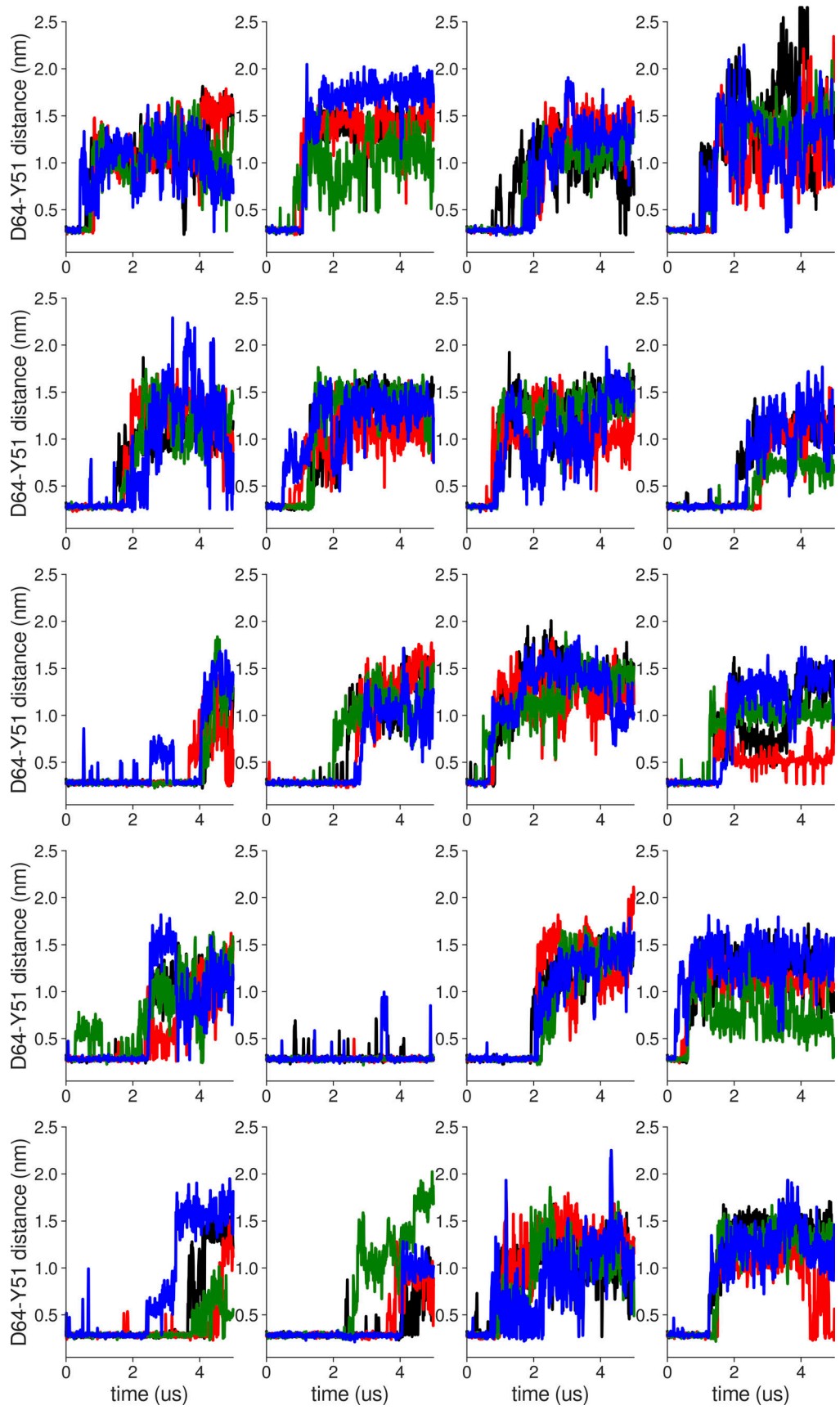

**Figure S5. Individual distance traces between D64 CG atom and Y51 HH atom in long MthK V55E CHARMM simulations at 300 mV.** Each color refers to one of the channel monomers. Each panel shows traces from an independent, 5-μs long simulation. Distance values >1 nm indicate broken interactions.

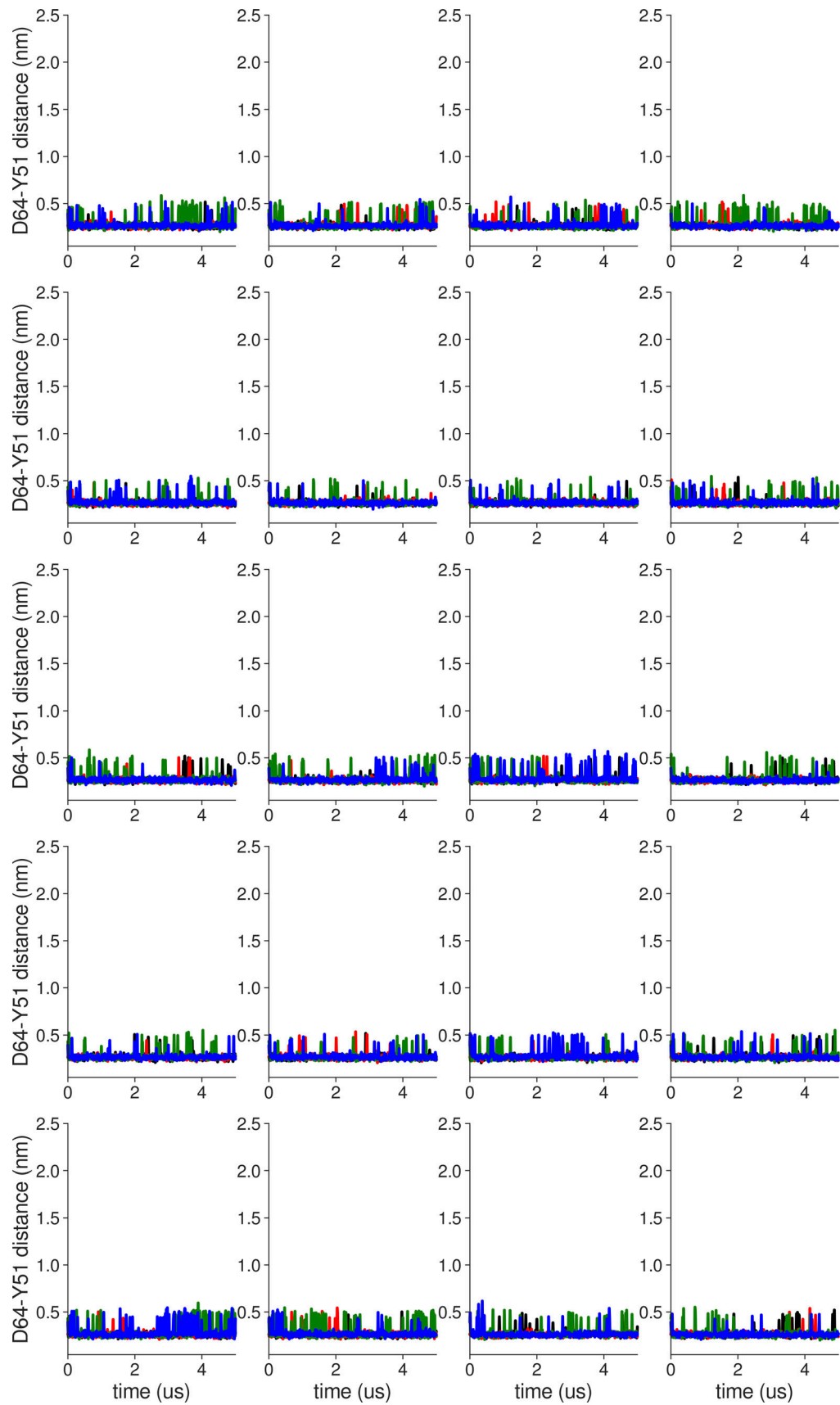

Figure S6.   **Individual distance traces between D64 CG atom and Y51 HH atom in long MthK WT AMBER simulations at 150 mV.** Each color refers to one of the channel monomers. Each panel shows traces from an independent, 5-µs long simulation. Distance values >1 nm indicate broken interactions.

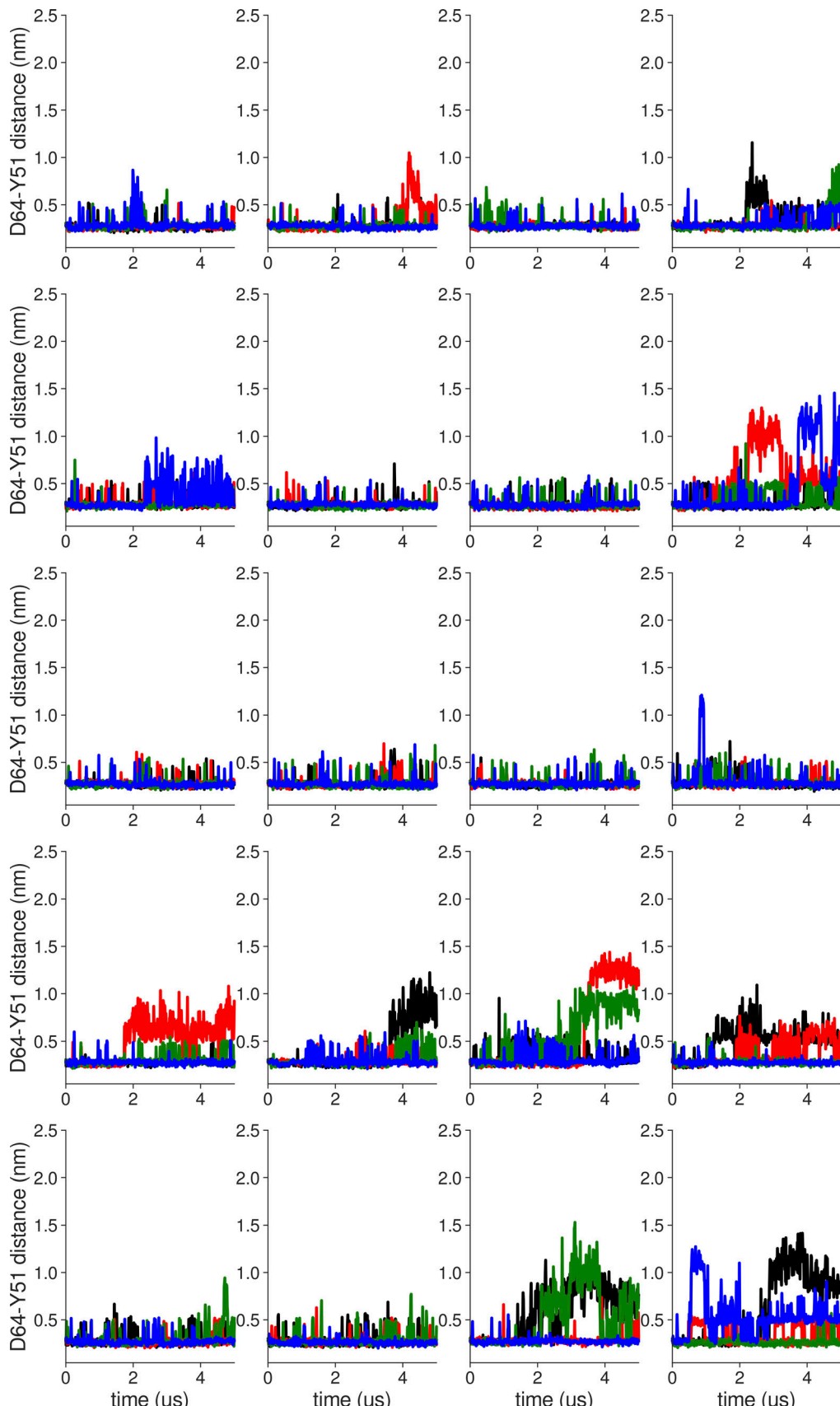

Figure S7. **Individual distance traces between D64 CG atom and Y51 HH atom in long MthK V55E AMBER simulations at 150 mV.** Each color refers to one of the channel monomers. Each panel shows traces from an independent, 5-µs long simulation. Distance values >1 nm indicate broken interactions.

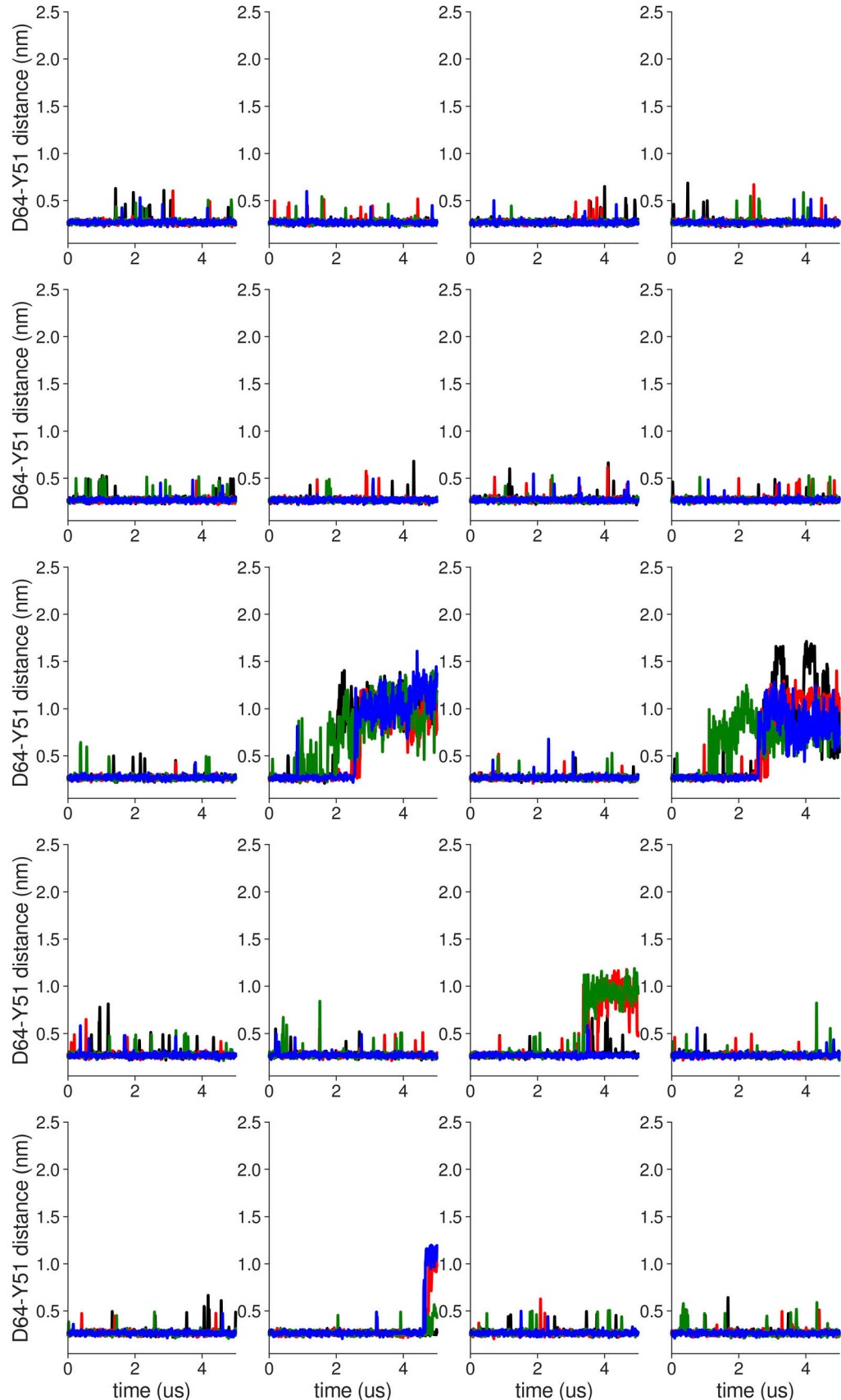

**Figure S8.   Individual distance traces between D64 CG atom and Y51 HH atom in long MthK WT CHARMM simulations at 150 mV**. Each color refers to one of the channel monomers. Each panel shows traces from an independent, 5-µs long simulation. Distance values >1 nm indicate broken interactions.

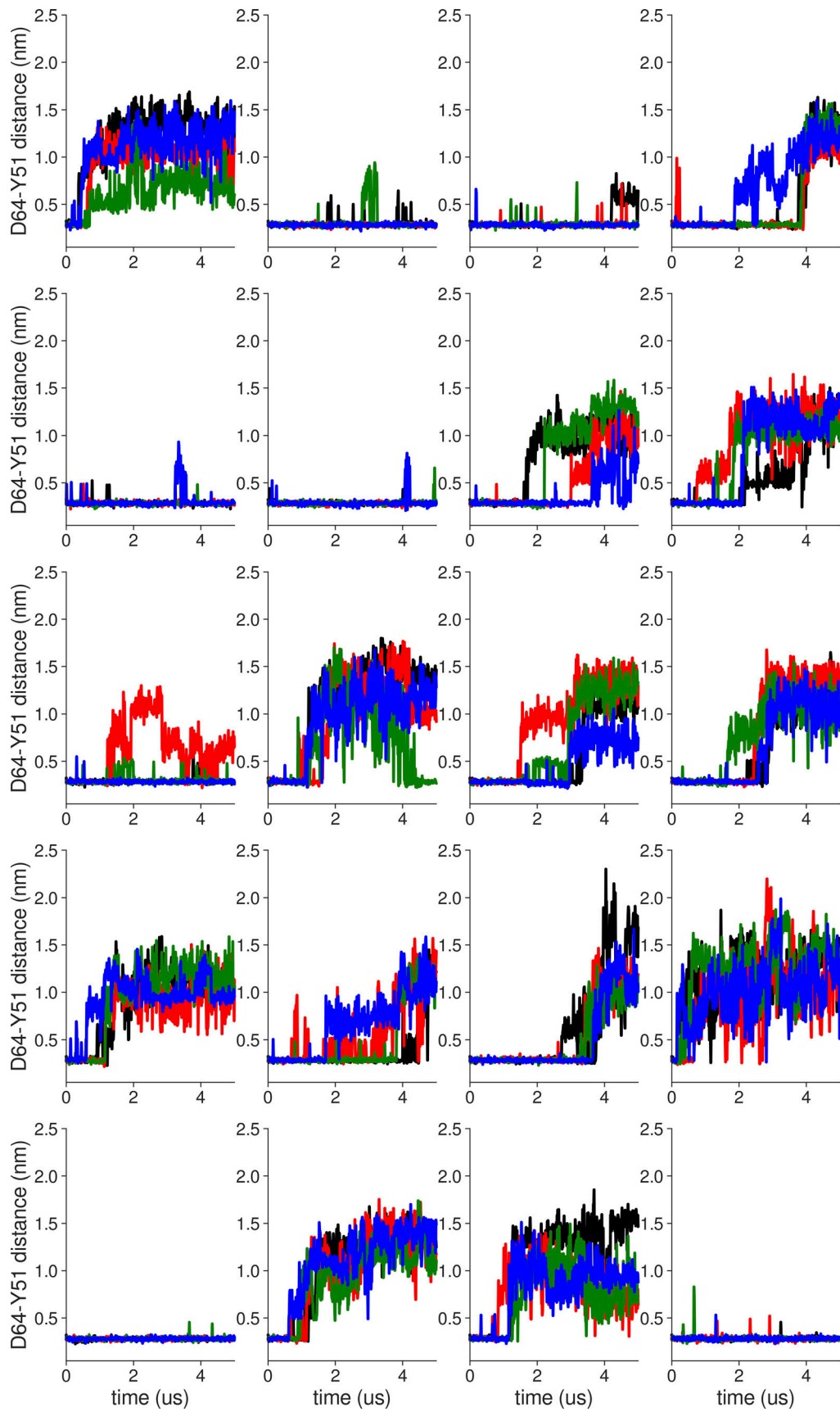

Figure S9. **Individual distance traces between D64 CG atom and Y51 HH atom in long MthK V55E CHARMM simulations at 150 mV.** Each color refers to one of the channel monomers. Each panel shows traces from an independent, 5-μs long simulation. Distance values >1 nm indicate broken interactions.

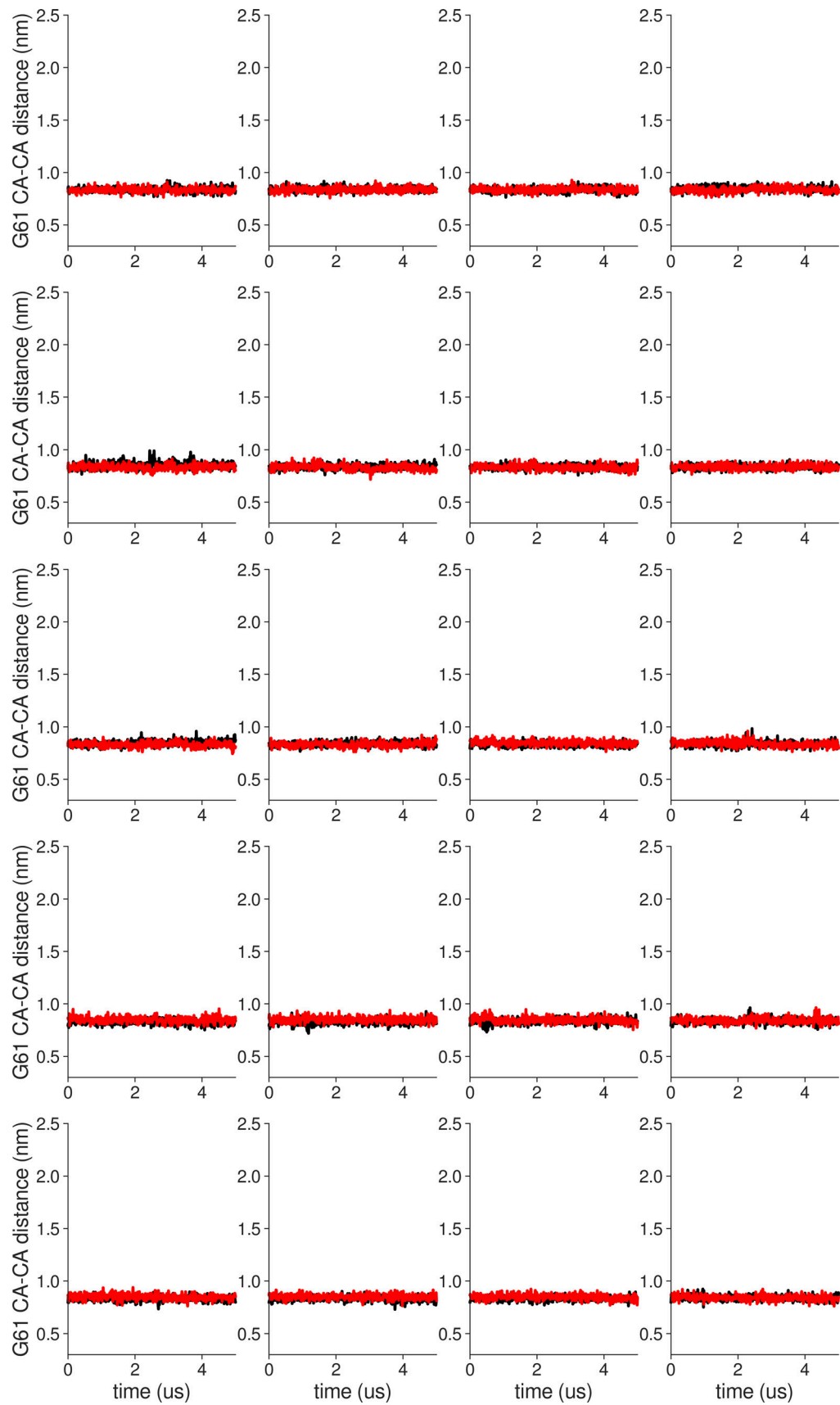

Figure S10. **Individual distance traces between G61 CA atoms between oppositely oriented monomers in long MthK WT AMBER simulations at 300 mV.** Each panel shows traces from an independent, 5-µs long simulation.

**Kopec et al.**
Selectivity filter gating in the MthK channel

**Journal of General Physiology**    S10

Figure S11. **Individual distance traces between G61 CA atoms between oppositely oriented monomers in long MthK V55E AMBER simulations at 300 mV.** Each panel shows traces from an independent, 5-μs long simulation.

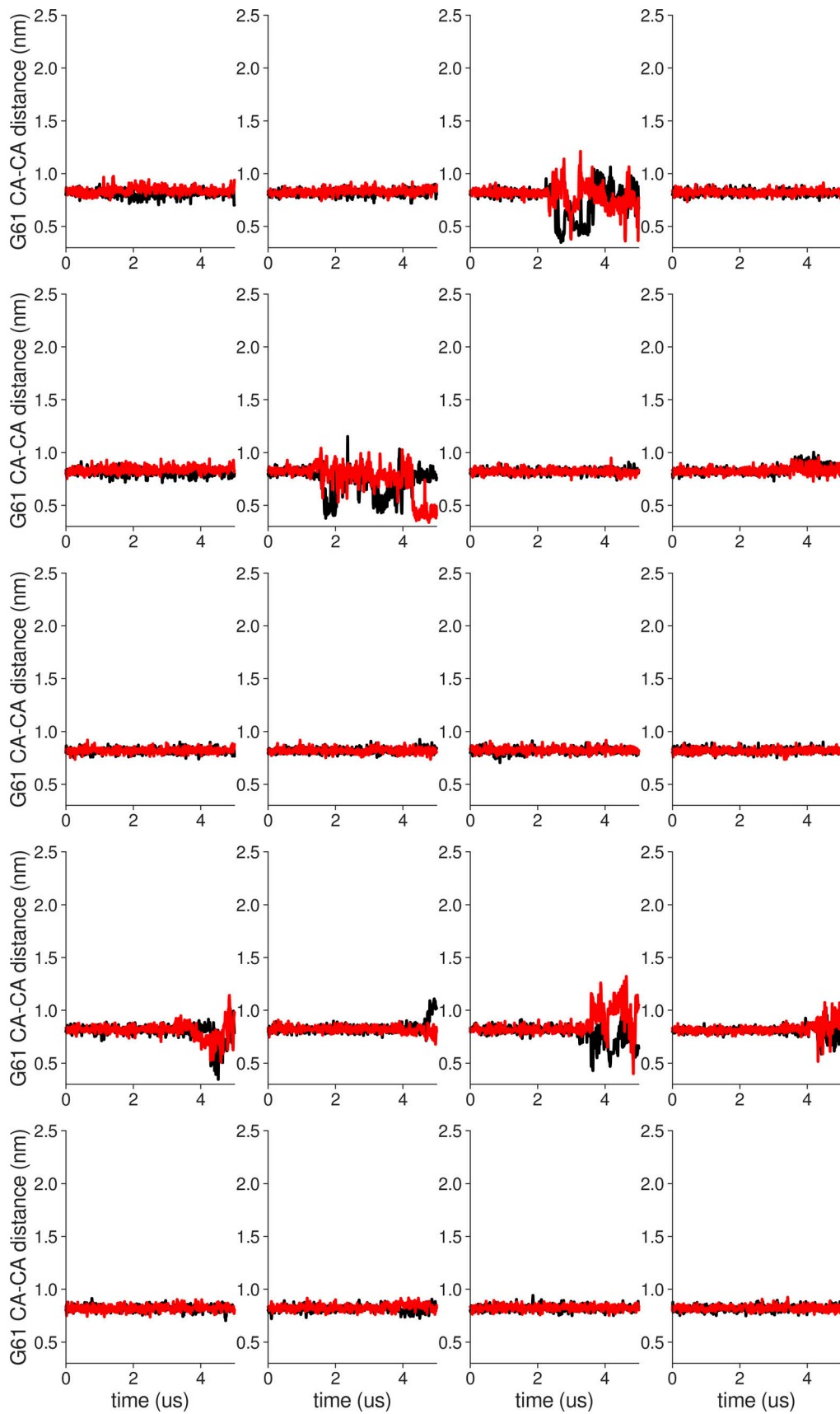

Figure S12.   **Individual distance traces between G61 CA atoms between oppositely oriented monomers in long MthK WT CHARMM simulations at 300 mV.** Each panel shows traces from an independent, 5-µs long simulation.

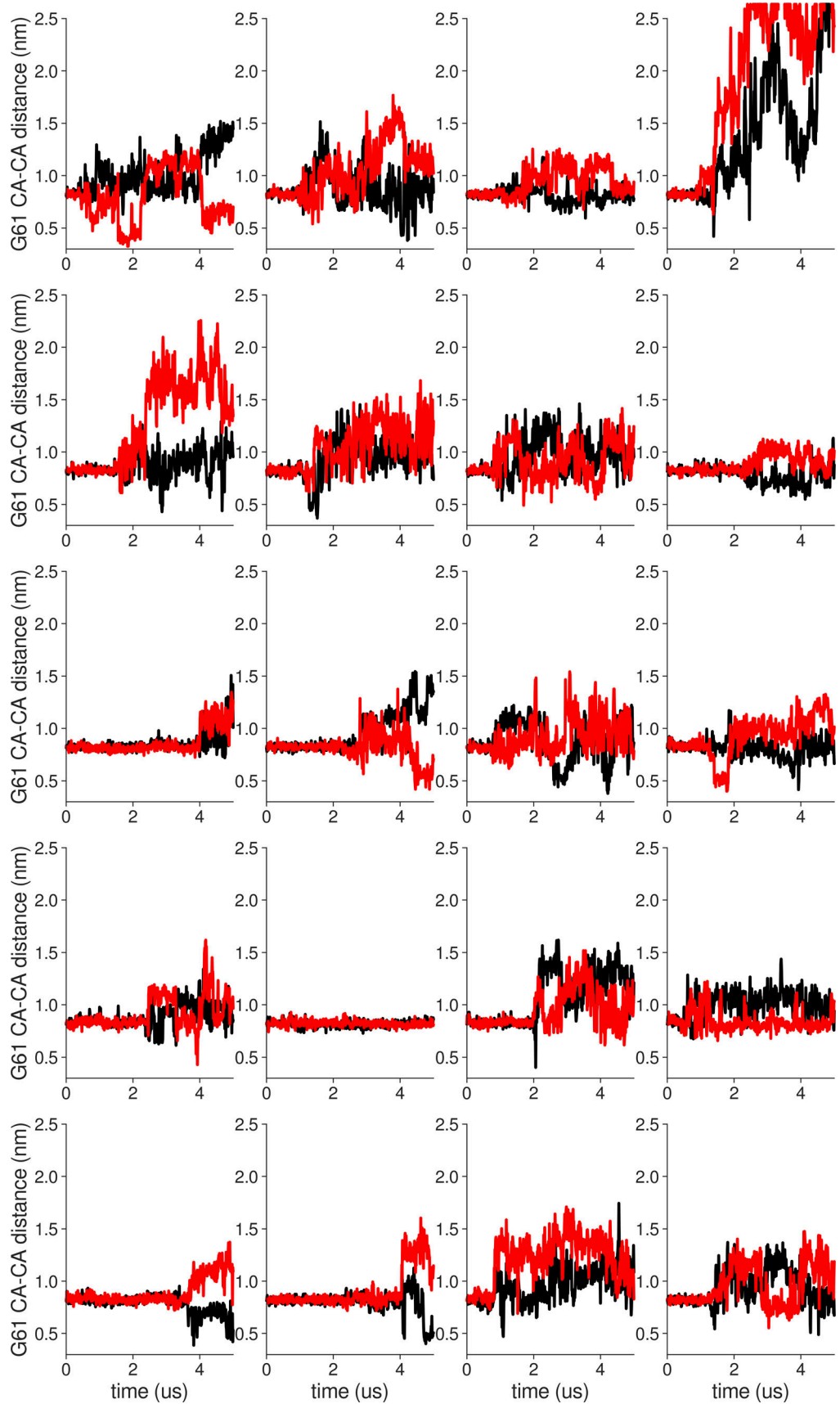

**Figure S13.** **Individual distance traces between G61 CA atoms between oppositely oriented monomers in long MthK V55E CHARMM simulations at 300 mV.** Each panel shows traces from an independent, 5-µs long simulation.

**Figure S14. Individual distance traces between G61 CA atoms between oppositely oriented monomers in long MthK WT AMBER simulations at 150 mV.** Each panel shows traces from an independent, 5-µs long simulation.

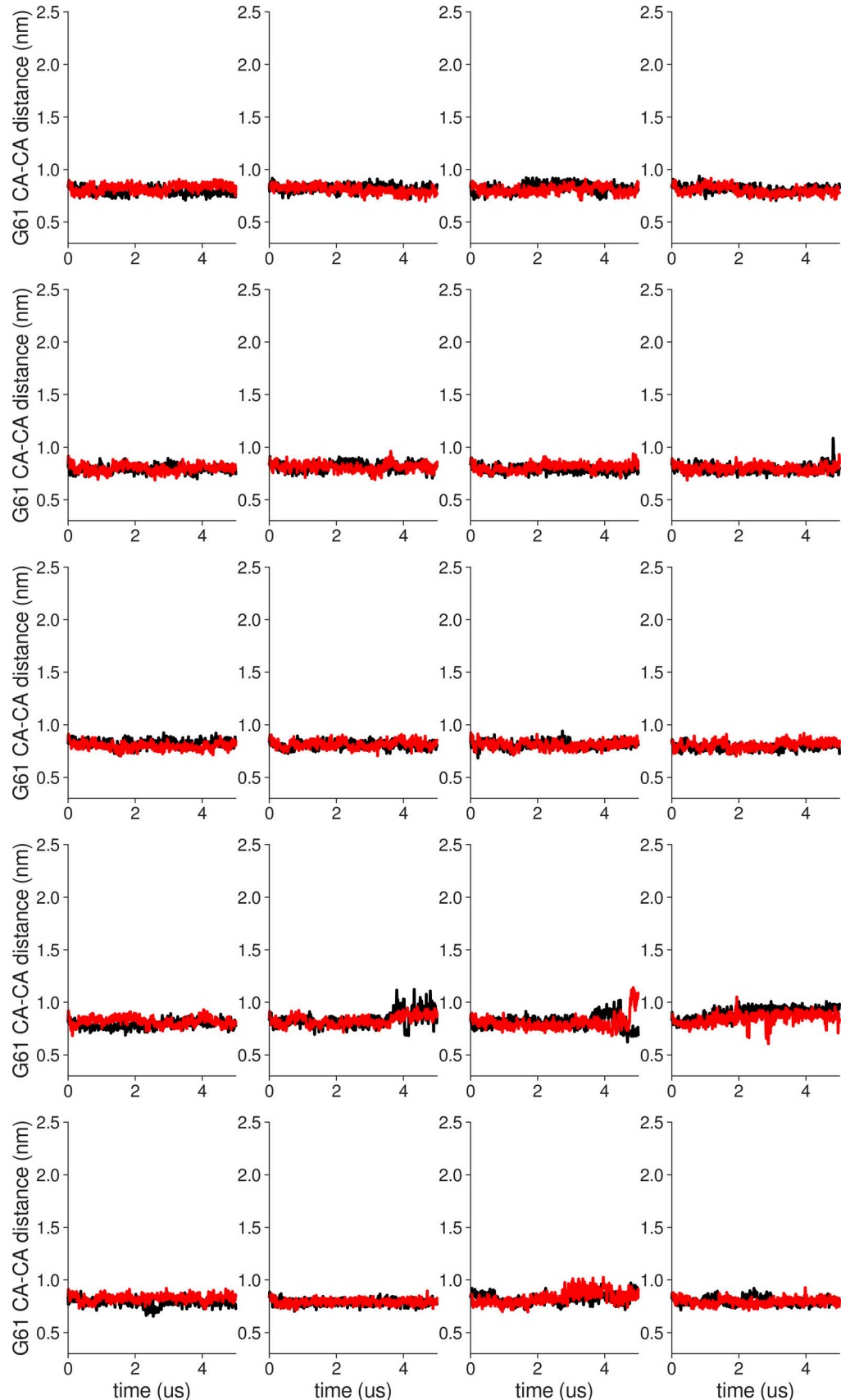

Figure S15. **Individual distance traces between G61 CA atoms between oppositely oriented monomers in long MthK V55E AMBER simulations at 150 mV.** Each panel shows traces from an independent, 5-μs long simulation.

**Kopec et al.**
Selectivity filter gating in the MthK channel

**Journal of General Physiology**    S15

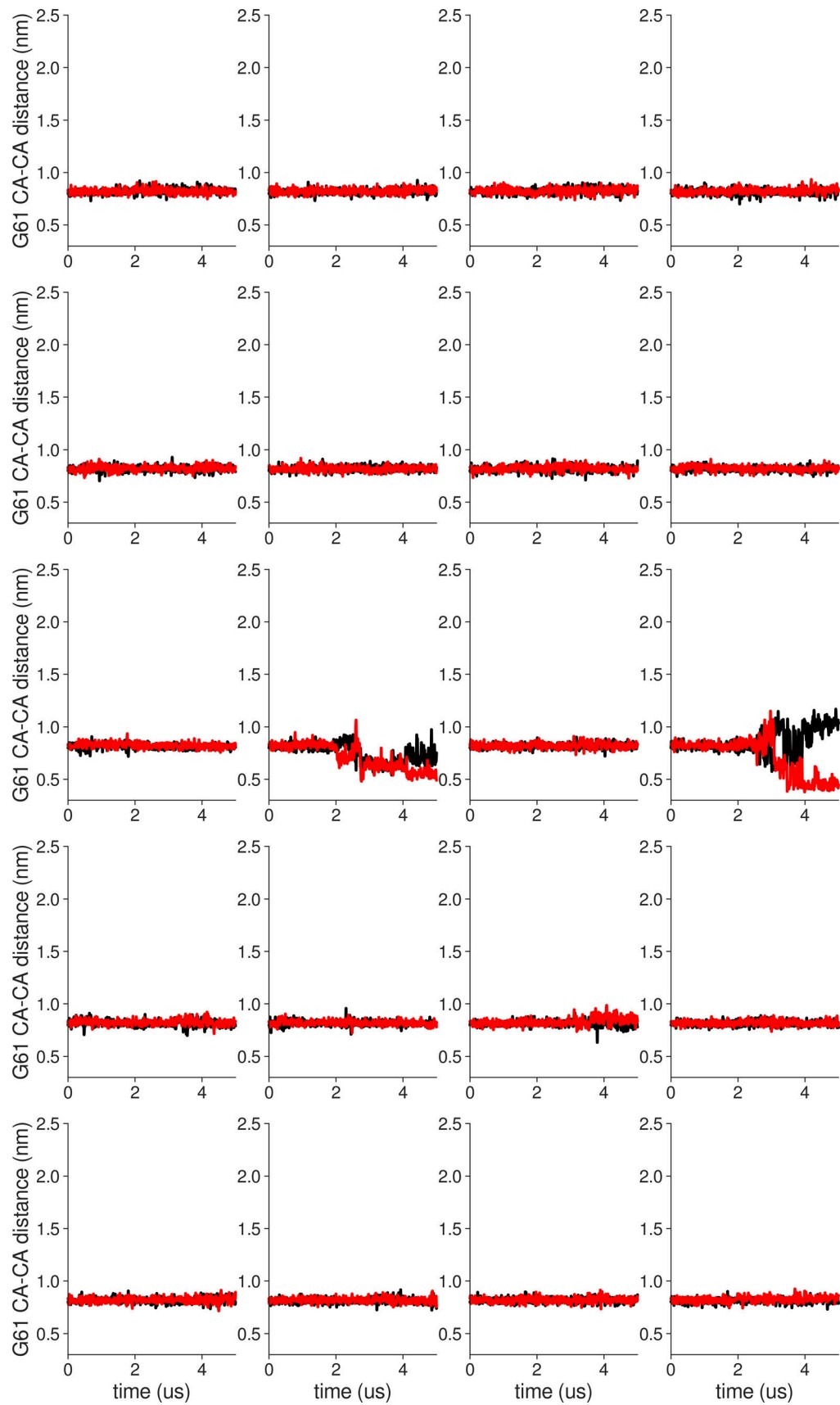

Figure S16. **Individual distance traces between G61 CA atoms between oppositely oriented monomers in long MthK WT CHARMM simulations at 150 mV.** Each panel shows traces from an independent, 5-μs long simulation.

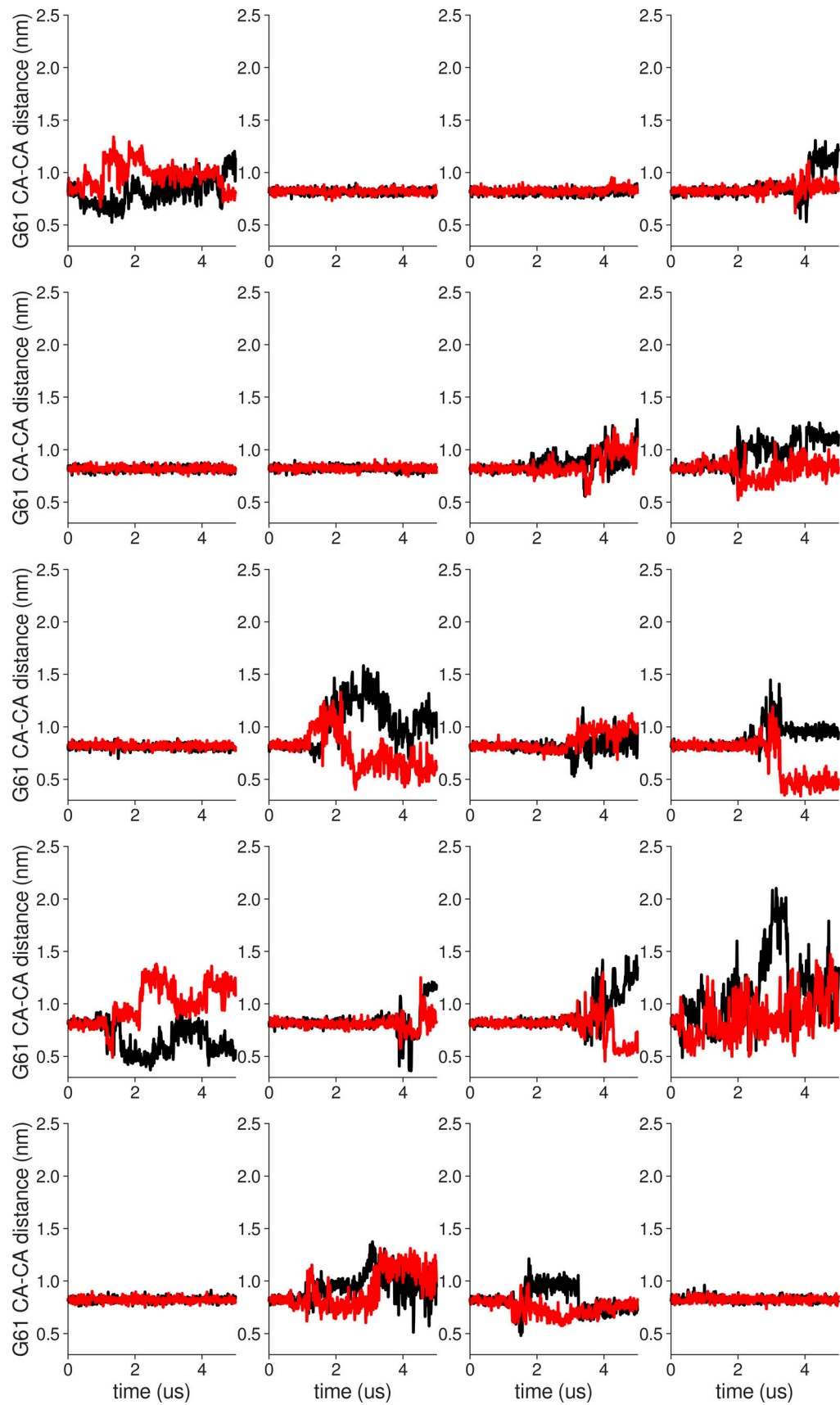

Figure S17.   **Individual distance traces between G61 CA atoms between oppositely oriented monomers in long MthK V55E CHARMM simulations at 300 mV.** Each panel shows traces from an independent, 5-μs long simulation.

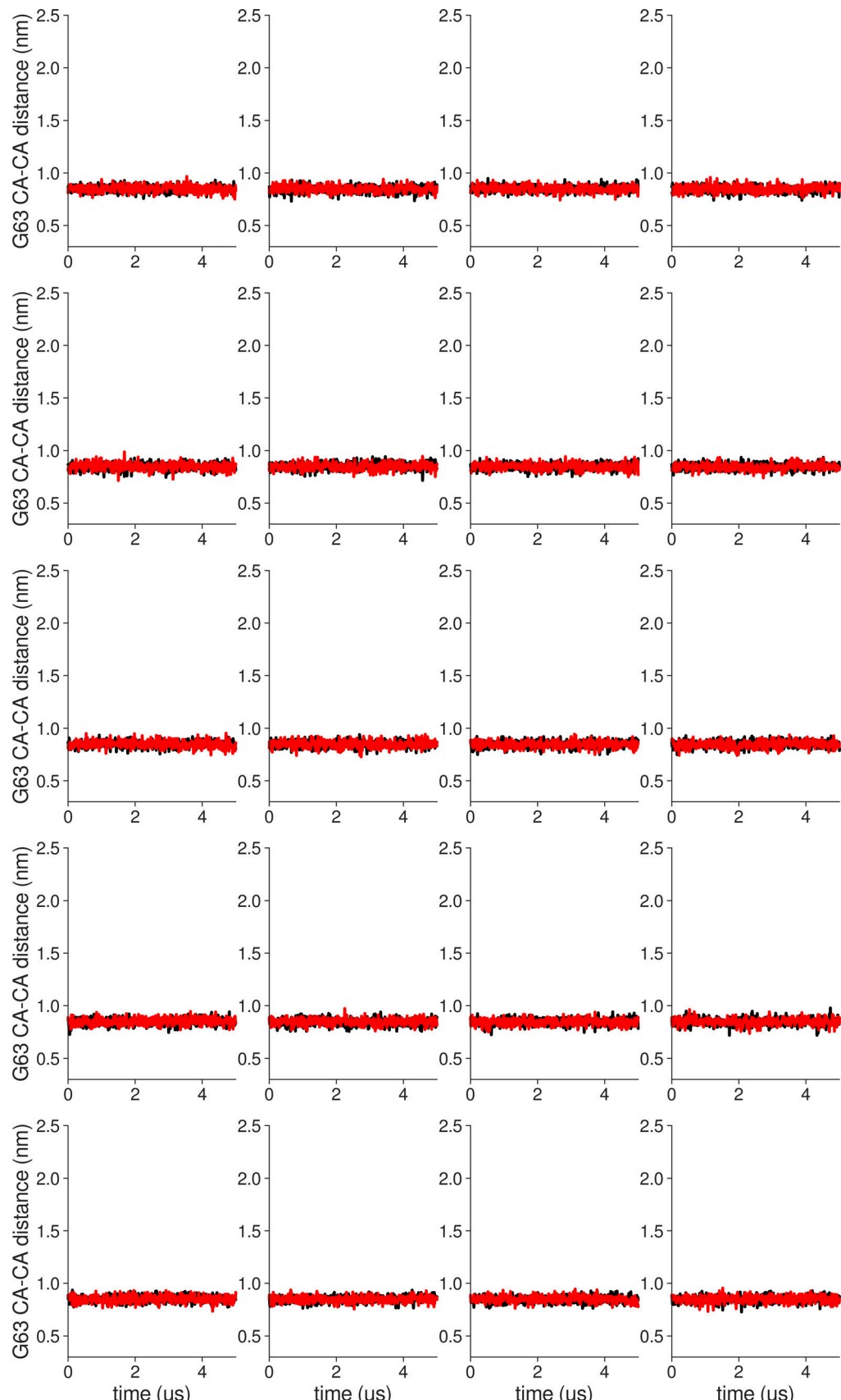

Figure S18. **Individual distance traces between G63 CA atoms between oppositely oriented monomers in long MthK WT AMBER simulations at 300 mV.** Each panel shows traces from an independent, 5-µs long simulation.

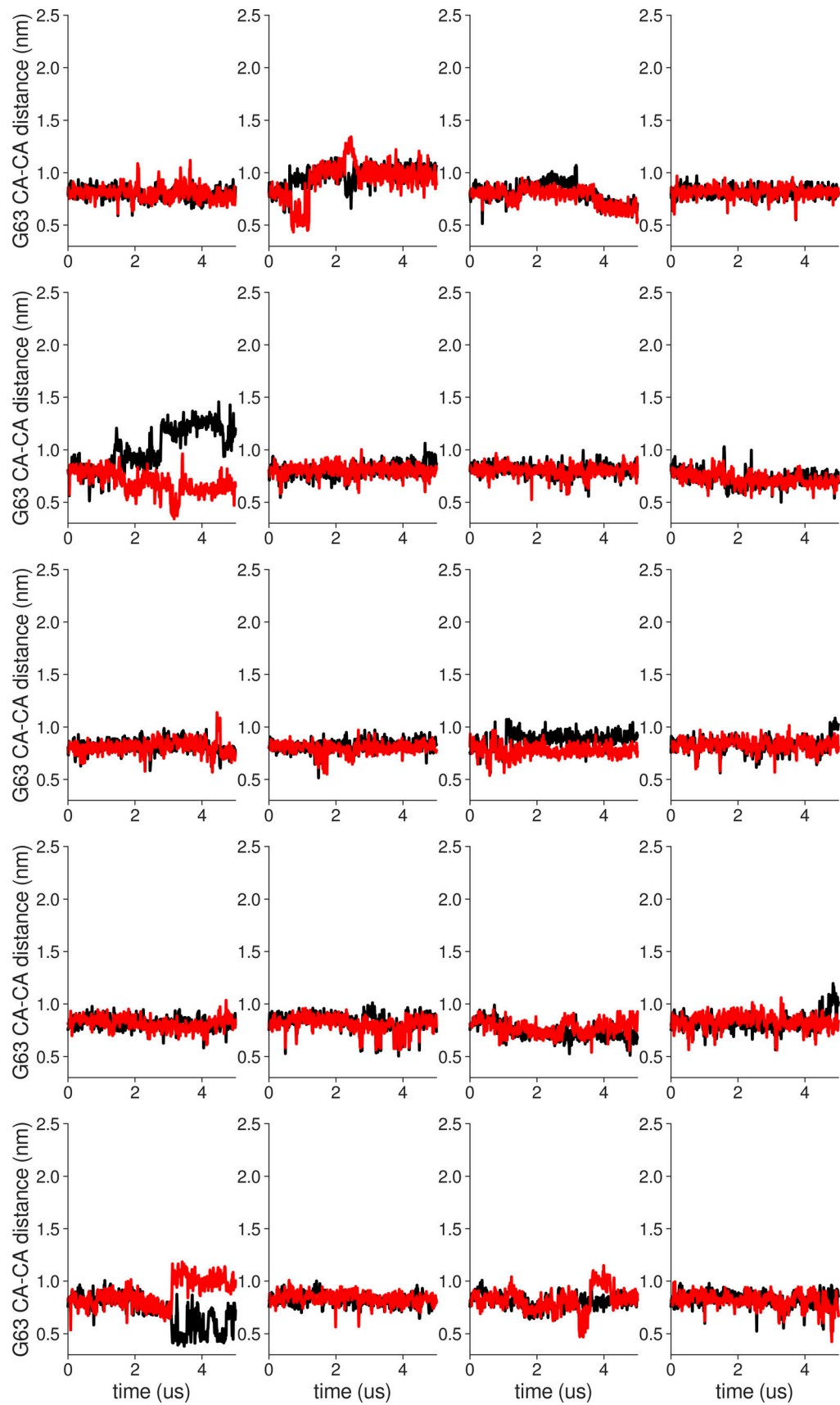

Figure S19.   **Individual distance traces between G63 CA atoms between oppositely oriented monomers in long MthK V55E AMBER simulations at 300 mV.** Each panel shows traces from an independent, 5-µs long simulation.

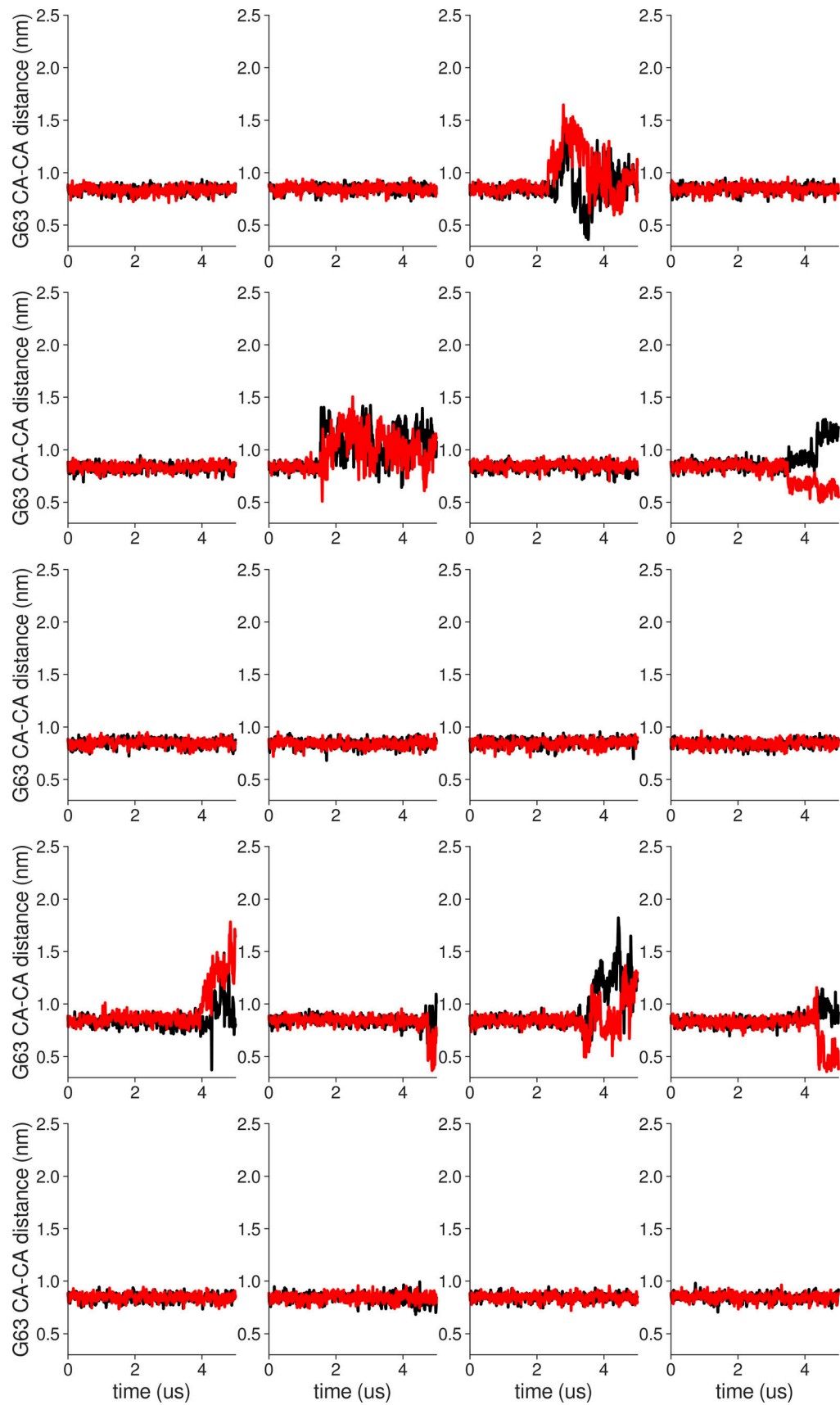

Figure S20.   **Individual distance traces between G63 CA atoms between oppositely oriented monomers in long MthK WT CHARMM simulations at 300 mV.** Each panel shows traces from an independent, 5-μs long simulation.

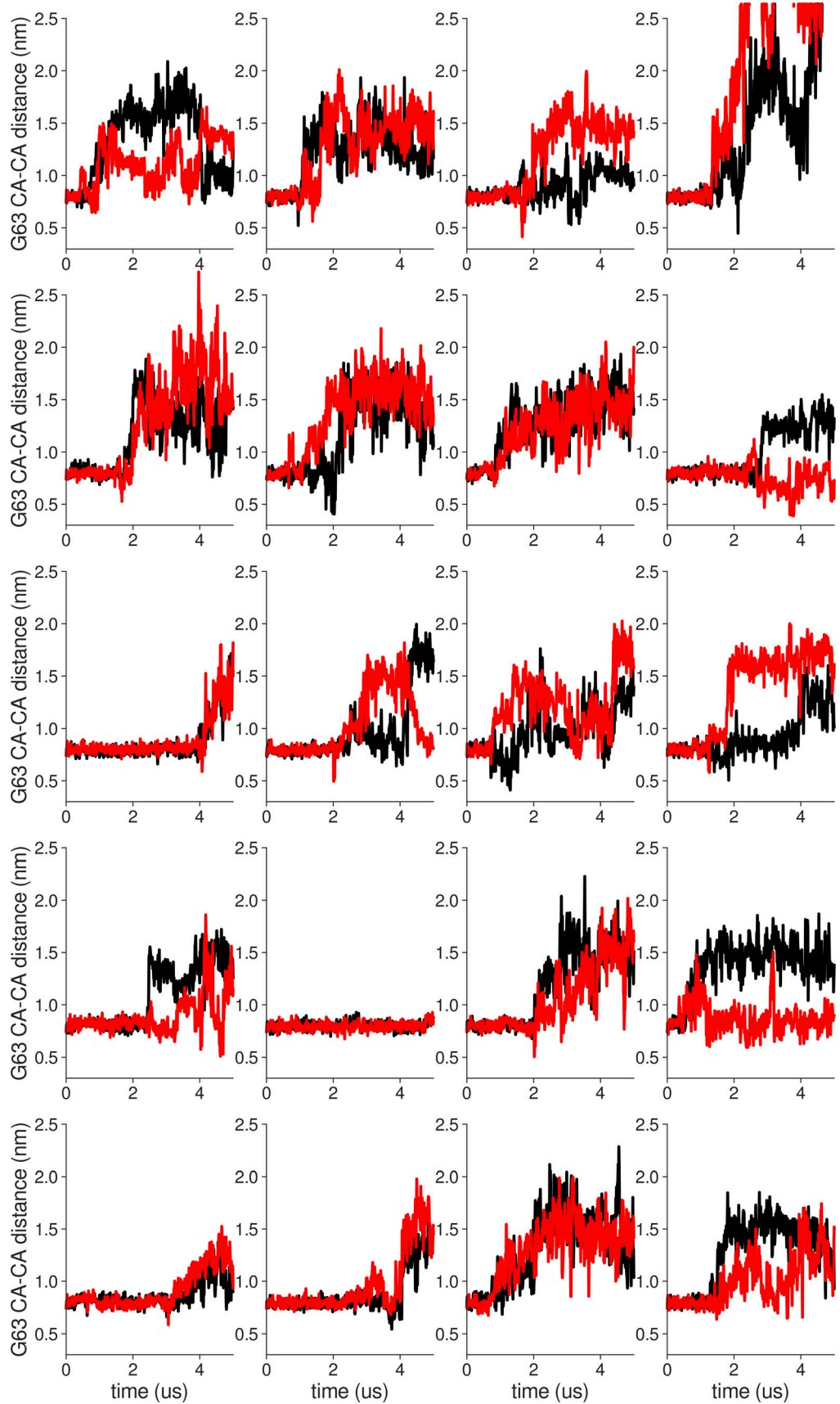

Figure S21. **Individual distance traces between G63 CA atoms between oppositely oriented monomers in long MthK V55E CHARMM simulations at 300 mV.** Each panel shows traces from an independent, 5-µs long simulation.

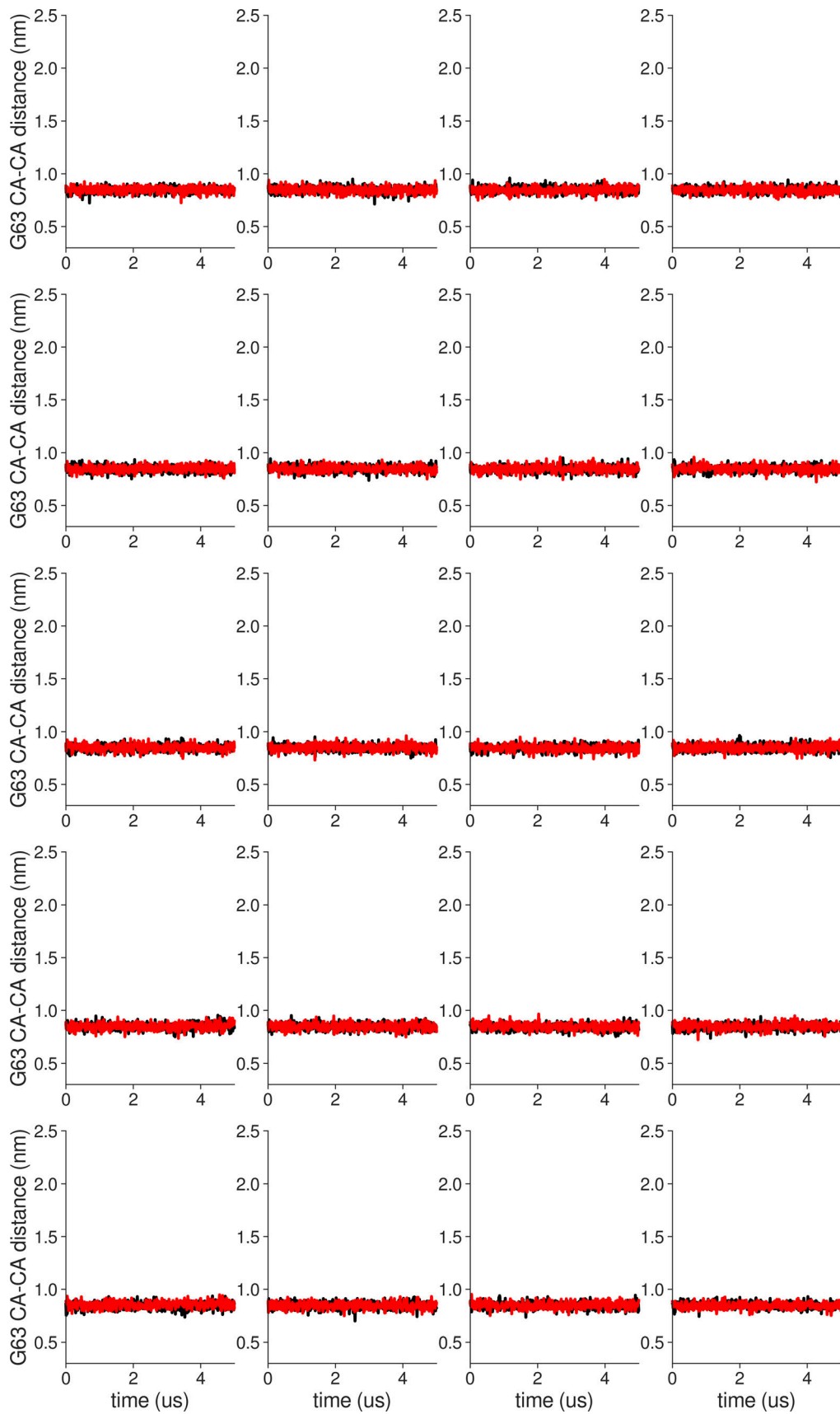

**Figure S22. Individual distance traces between G63 CA atoms between oppositely oriented monomers in long MthK WT AMBER simulations at 150 mV.** Each panel shows traces from an independent, 5-µs long simulation.

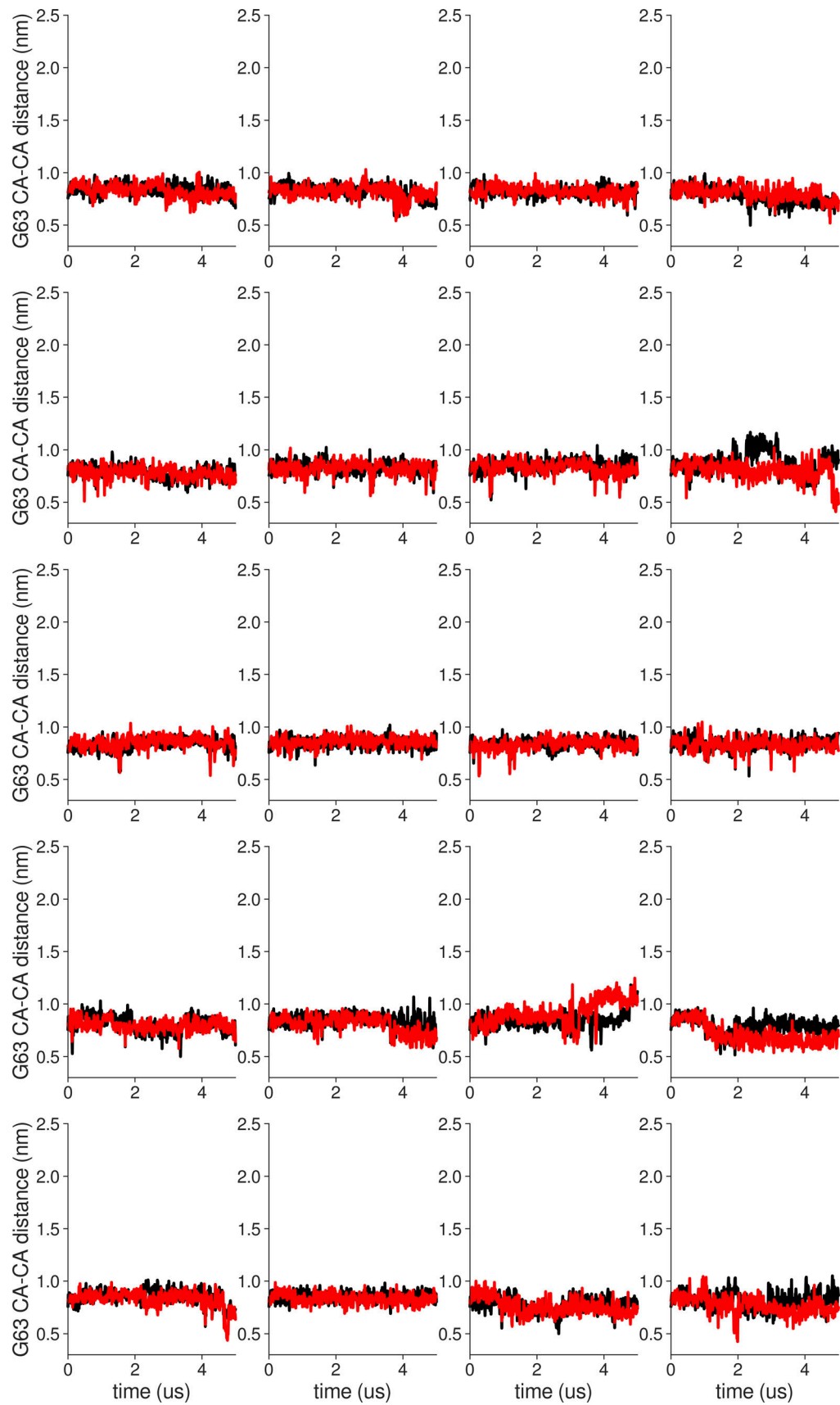

Figure S23. **Individual distance traces between G63 CA atoms between oppositely oriented monomers in long MthK V55E AMBER simulations at 150 mV.** Each panel shows traces from an independent, 5-µs long simulation.

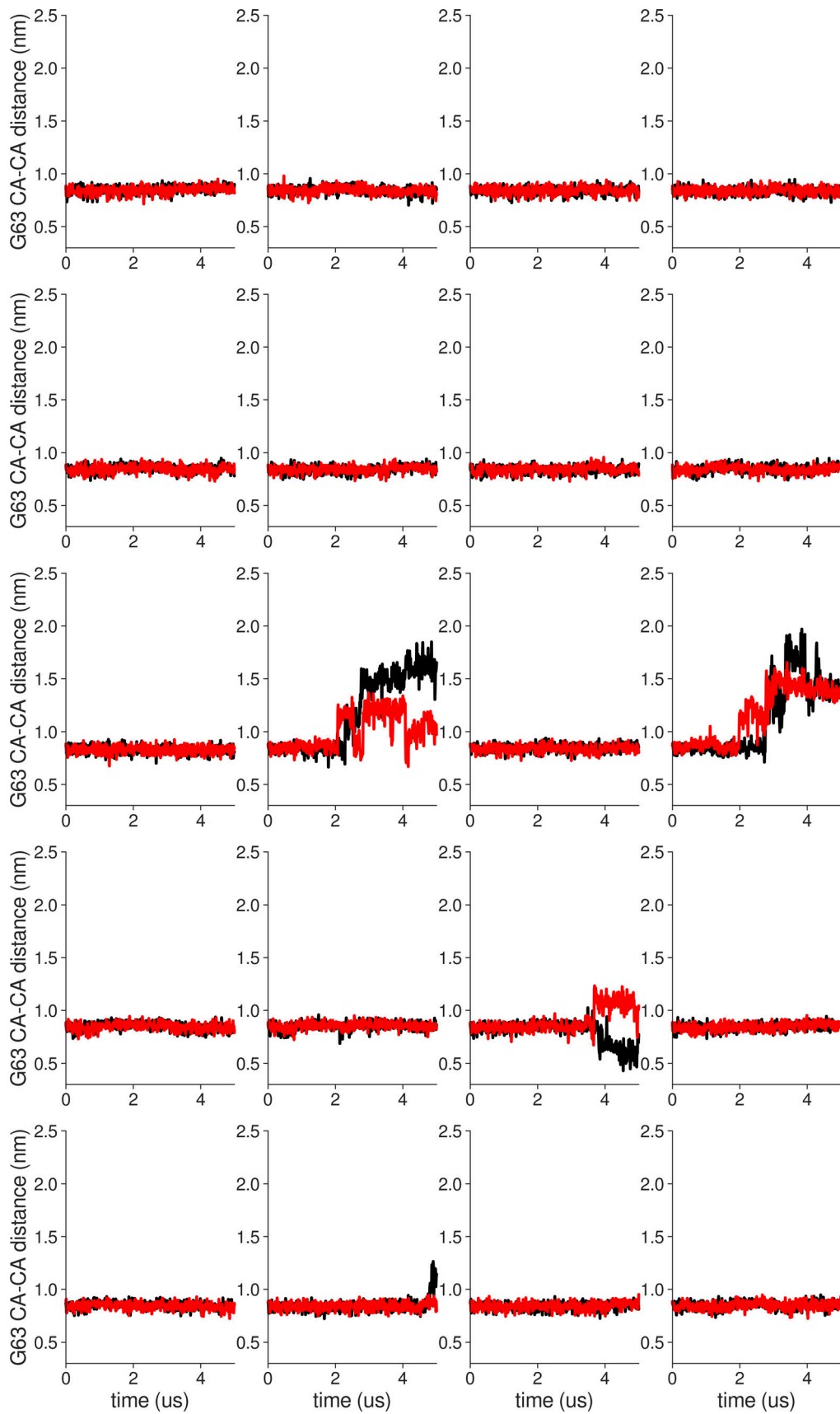

Figure S24. **Individual distance traces between G63 CA atoms between oppositely oriented monomers in long MthK WT CHARMM simulations at 150 mV.** Each panel shows traces from an independent, 5-µs long simulation.

Figure S25. **Individual distance traces between G63 CA atoms between oppositely oriented monomers in long MthK V55E CHARMM simulations at 150 mV.** Each panel shows traces from an independent, 5-µs long simulation.

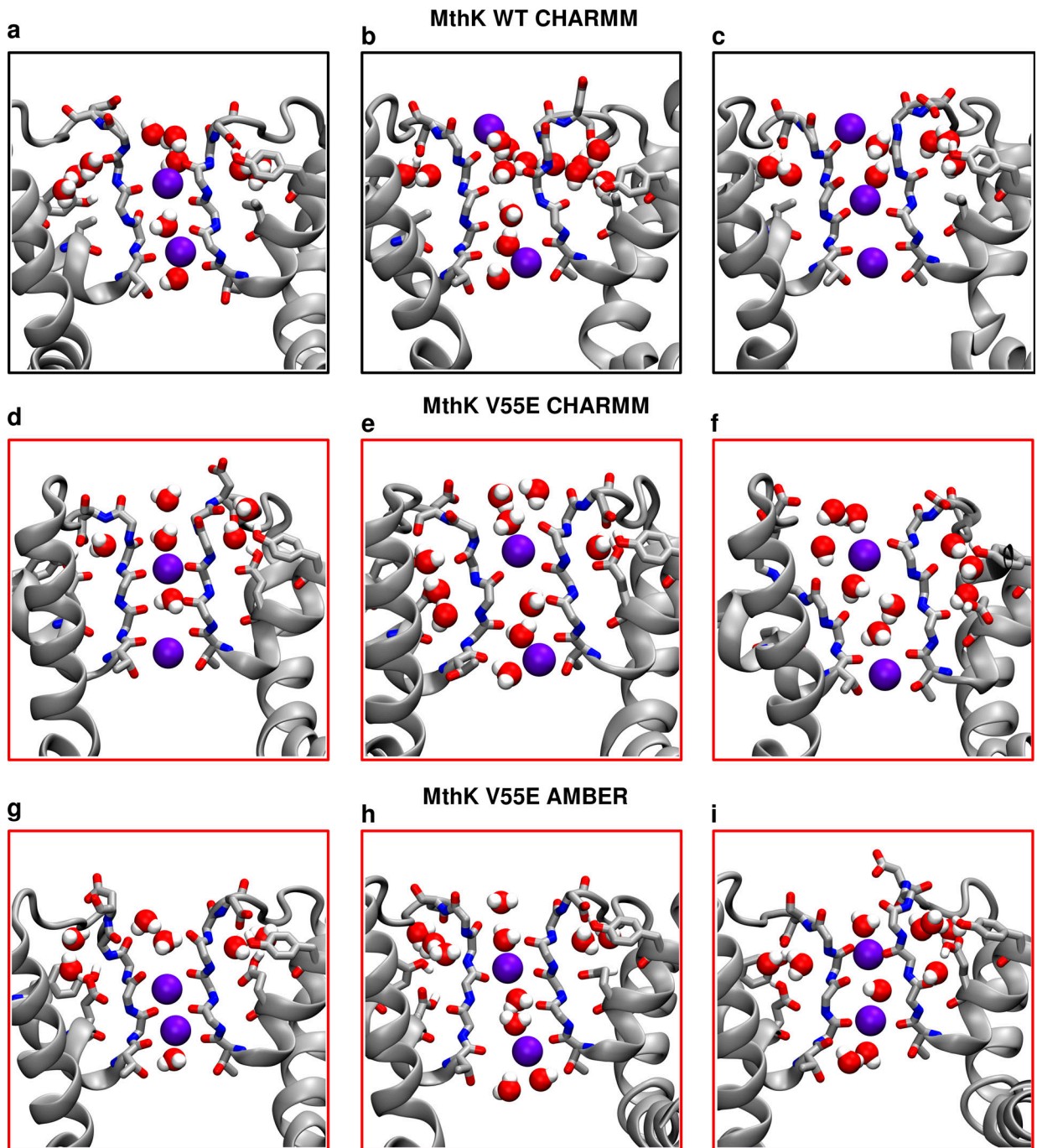

Figure S26. **Conformations of inactivated filters in MD simulations of MthK WT and MthK V55E at 150 mV. (a–c)** Snapshots from simulations of MthK WT with the CHARMM force field. **(d–f)** Snapshots from simulations of MthK V55E with the CHARMM force field. **(g–i)** Snapshots from simulations of MthK V55E with the AMBER force field. Both aspartate (D64) sidechain flipping as well as valine (V60) and glycine (G61) carbonyl flipping can be observed when SFs lose $K^+$ ions.

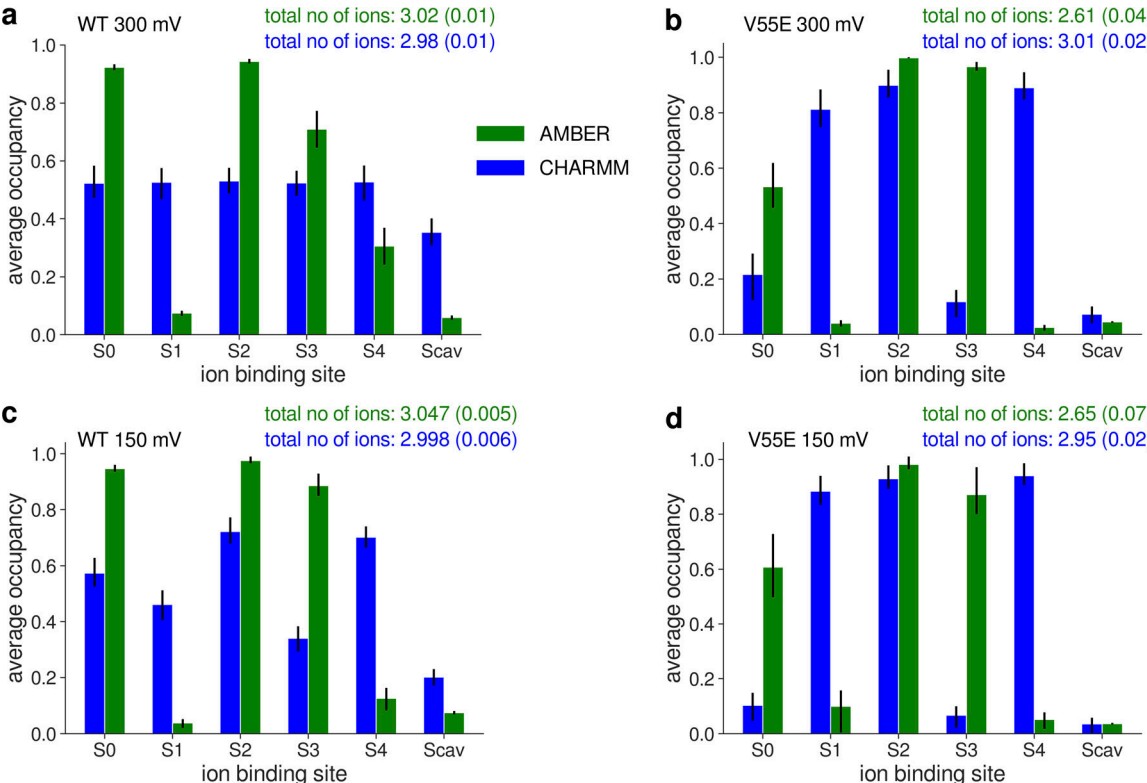

Figure S27.   **Average potassium occupancy of each ion binding site in the SF of MthK WT and V55E before the D64 flip. (a–d)** MthK WT at 300 mV (a), MthK V55E at 300 mV (b), MthK WT at 150 mV (c), and MthK V55E at 150 mV (d).

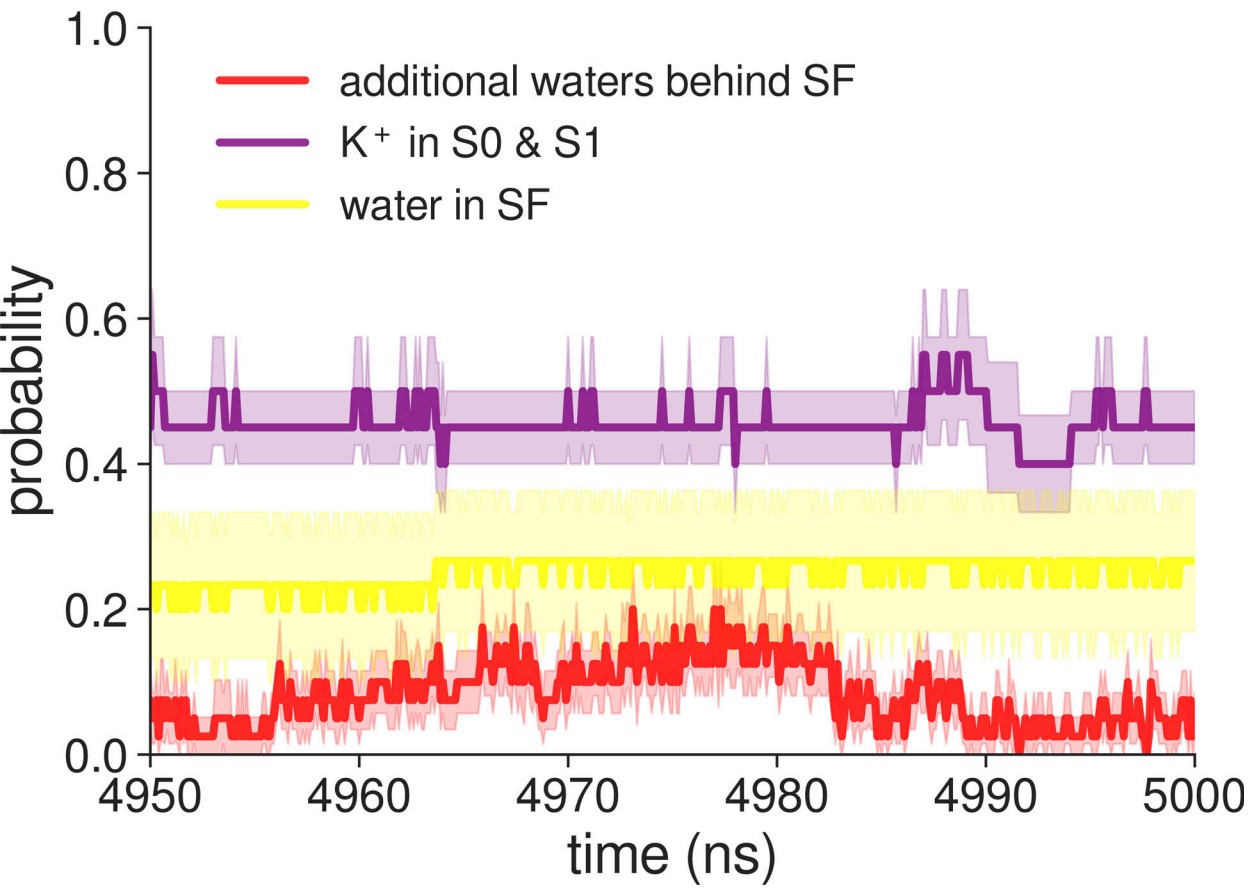

Figure S28. **Probabilities of molecular events in the last 50 ns of MD simulations of MthK WT with the AMBER force field.**

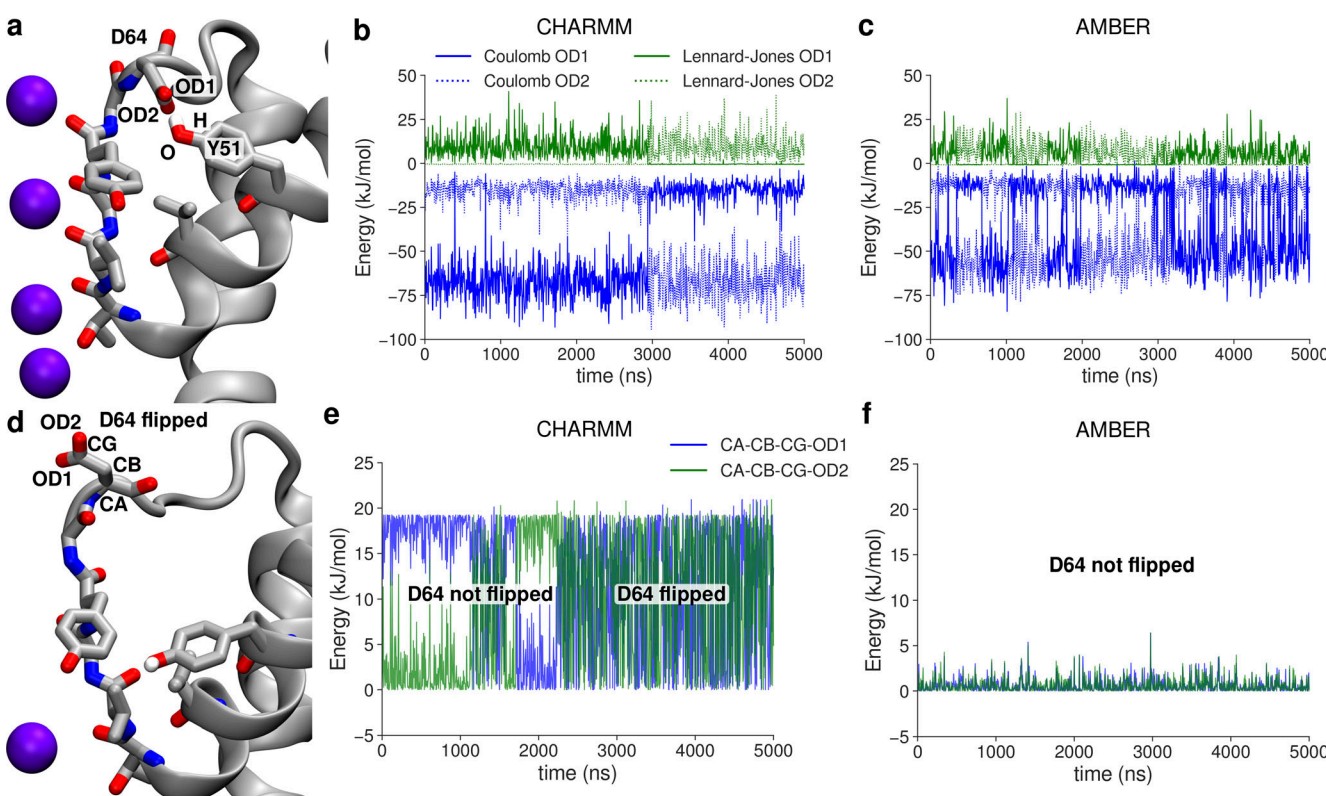

Figure S29. **Factors determining the dynamics of the D64 side chain in MthK WT in CHARMM and AMBER force fields. (a)** Typical position of side chains before the aspartate flip. The D64 side chain interacts with the Y51 side chain via hydrogen bond. **(b and c)** Interaction energies (Coulomb and Lennard-Jones) between atoms involved in D64–Y51 hydrogen bond, in CHARMM and AMBER, respectively. **(d)** Snapshot after the aspartate flip, showing the D64 side chain pointing away from the protein. **(e and f)** Energies of the two dihedrals angles defined by heavy atoms from the D64 side chain, in CHARMM and AMBER, respectively.

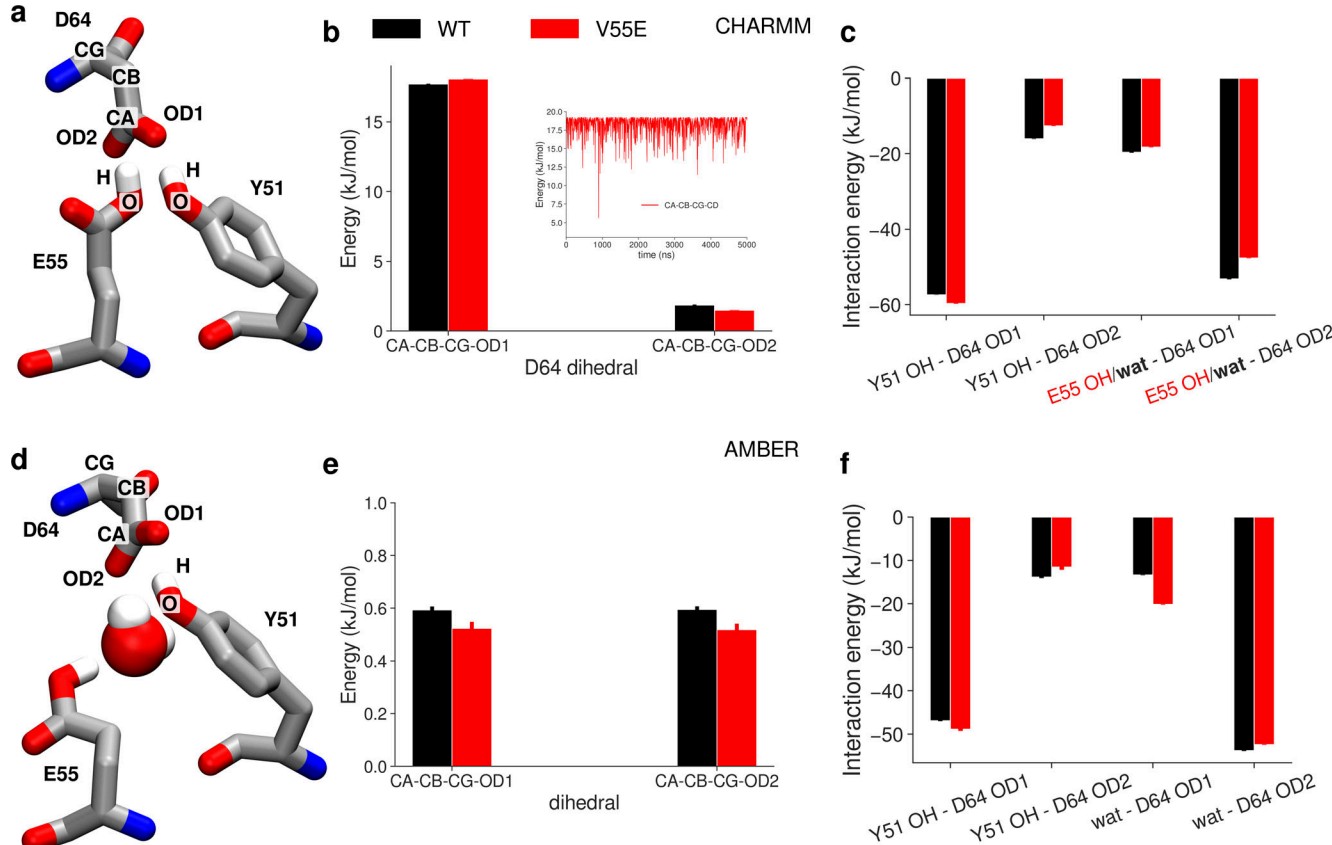

Figure S30. **The effects of the V55E mutation on factors determining the dynamics of the D64 side chain in CHARMM and AMBER force fields.**
**(a)** Typical position of D64, E55, and Y51 side chains in the CHARMM force field, before the aspartate flip. The E55 side chain is in the vertical orientation.
**(b)** Effects of the V55E mutation on the average energy of the two dihedrals angles of the D64 side chain. Inset shows a typical time evolution of a high energy dihedral. **(c)** Average interaction energies between atoms forming the Y51–D64 hydrogen bond and E55–D64 hydrogen bond (red, in MthK V55E) or D64–water hydrogen bond (black, in MthK WT). **(d)** Typical position of D64, E55, and Y51 side chain the AMBER force field, before the aspartate flip. The E55 side chain is in the horizontal orientation, and there is an additional water molecule between D64 and E55 side chains. **(e)** Same as in b, but in the AMBER force field. **(f)** Same as in c, but in the AMBER force field. As the E55 side chain is typically in the horizontal orientation, and thus does not form a hydrogen bond with the D64 side chain, only interaction energies with water are included.

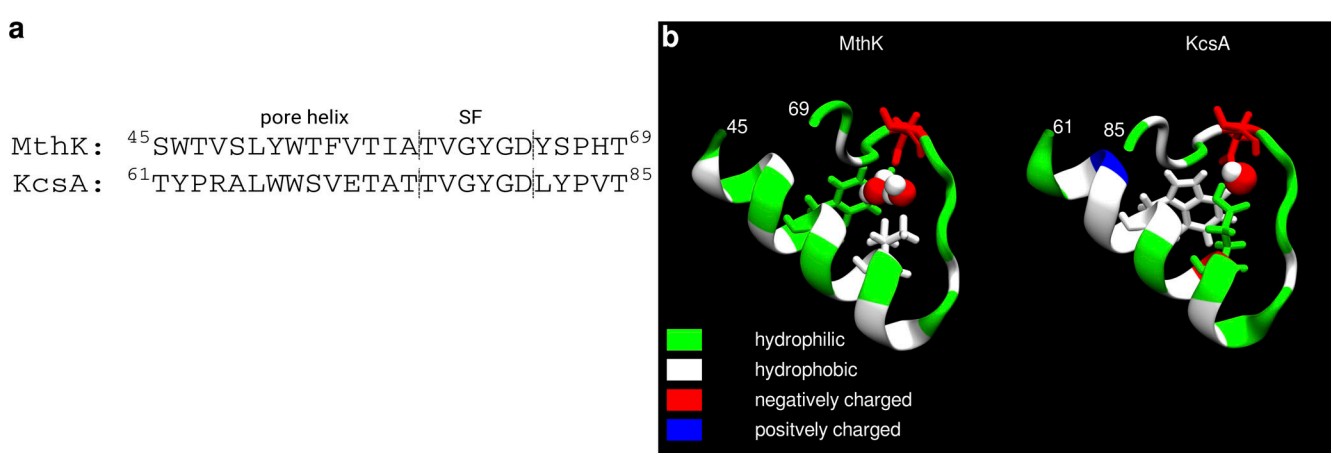

**Figure S31. Nonequilibrium free energy calculations of E55 deprotonation. (a)** System used in free energy calculations, showing a selected E55 glutamate (green box), a reference state tripeptide G-E2J-G and a lowered salt concentration of 150 mM. **(b and c)** End states for E55 in free energy simulations: in the State 0, E55 is protonated, and in the State 1 it is deprotonated. In both states, E55 is kept in its horizontal orientation. When E55 is deprotonated (State 1), the side chain of D64 can flip toward the extracellular space. **(d and e)** Distributions of work values in nonequilibrium transitions and the final estimates of ΔG obtained with pmx, in AMBER and CHARMM, respectively.

**Figure S32. Comparison of the pore region between MthK and KcsA channels. (a)** Sequence alignment for MthK and KcsA. **(b)** Visualization of the pore helix, selectivity filter and the loop following SF in MthK and KcsA. Residues are colored according to their chemical character: hydrophilic (green), hydrophobic (white), negatively charged (red), or positively charged (blue). The critical residues discussed in this work: Y51/W67, V55/E71, and D64/D80 (in MthK/KcsA) are shown as sticks. Water molecules are shown as red and white spheres.

