## [Peer Review File · The Journal of General Physiology]

Interactions between selectivity filter and pore helix control filter gating in the MthK channel

Wojciech Kopec, Andrew Thomson, Bert de Groot, and Brad Rothberg

Corresponding Author(s): Wojciech Kopec, Max Planck Institute for Multidisciplinary Sciences

Review Timeline:

Submission Date:	March 29, 2022
Editorial Decision:	May 10, 2022
Revision Received:	January 13, 2023
Editorial Decision:	March 2, 2023
Revision Received:	May 2, 2023
Editorial Decision:	May 23, 2023
Revision Received:	May 31, 2023

Editor: Crina Nimigean

Transaction Report:

DOI: <https://doi.org/10.1085/jgp.202213166>

May 10, 2022

Dr. Wojciech Kopec
Max Planck Institute for Multidisciplinary Sciences
Am Fassberg 11
Goettingen 37077
Germany

Re: 202213166

Dear Wojciech,

Thank you for submitting your manuscript, entitled "Driving Forces underlying Selectivity Filter Gating in the MthK Potassium Channel" to JGP. Your manuscript has now been seen by 3 reviewers, whose comments are appended below. You will see that although the reviewers found the manuscript timely and interesting, and appreciated the combination of electrophysiology and MD simulations as well as the total length of the simulations, they also have raised several concerns that should be addressed prior to further consideration of the manuscript at JGP. Importantly, to streamline the message and conclusions of the manuscript, a more effective integration between experimental results and MD simulations is needed, results from previous publications of the authors should be expanded upon, and special attention paid towards clarifying some confusion arising from using the terms open probability or permeability/single-channel conductance to compare the simulations of the V55E MthK mutant with the experimental data. In addition, the authors are encouraged to discuss findings from other researchers that appear in contrast with those presented in this manuscript.

Therefore, although we are unable to publish the current version of the manuscript, we would encourage you to submit a revised version that addresses all the concerns of the referees. We would be pleased to receive a suitably revised manuscript accompanied by a point-by-point response to the reviewers, which will be re-reviewed, most likely by some or all of the original referees. Based on the scope of the requested changes, we typically anticipate that the revision process will take no longer than 6 months, however, we understand you may need additional time to work on your resubmission to JGP. We therefore ask that you simply keep us informed as to a realistic submission timeline that is appropriate for your particular circumstances. In addition, please do not hesitate to contact me (via the editorial office) if you feel that a discussion of the reviewers' and editors' comments would be helpful.

Please submit your revised manuscript via the link below along with a point-by-point letter that details your responses to the editors' and reviewers' comments, as well as a copy of the text with alterations highlighted (boldfaced or underlined). If the article is eventually accepted, it would include a 'revised date' as well as submitted and accepted dates. If we do not receive the revised manuscript within one year, we will regard the article as having been withdrawn. We would be willing to receive a revision of the manuscript at a later time, but the manuscript will then be treated as a new submission, with a new manuscript number.

Please pay particular attention to recent changes to our instructions to authors in sections: Data presentation, Blinding and randomization and Statistical analysis, under Materials and Methods, as shown here: <https://rupress.org/jgp/pages/submission-guidelines#prepare>. Re-review will be contingent on inclusion of the required information (including for data added during revision) and demonstration of the experimental reproducibility of the results (i.e., all experimental data verified in at least 2 independent experiments).

Please note, for manuscripts with an original submission date received on or after Aug 2, 2021, JGP now requires authors to submit Source Data used to generate figures containing gels and Western blots with all revised manuscripts (when applicable). The original version of your paper was submitted on March 29, 2022.

This Source Data consists of fully uncropped and unprocessed images for each gel/blot displayed in the main and supplemental figures. If your paper includes cropped gel and/or blot images, please be sure to provide one Source Data file for each figure that contains gels and/or blots along with your revised manuscript files. File names for Source Data figures should be alphanumeric without any spaces or special characters (i.e., SourceDataF#, where F# refers to the associated main figure number or SourceDataFS# for those associated with Supplementary figures). The lanes of the gels/blots should be labeled as they are in the associated figure, the place where cropping was applied should be marked (with a box), and molecular weight/size standards should be labeled wherever possible.

Source Data files will be made available to reviewers during evaluation of revised manuscripts and, if your paper is eventually published in JGP, the files will be directly linked to specific figures in the published article.

Source Data Figures should be provided as individual PDF files (one file per figure). Authors should endeavor to retain a minimum resolution of 300 dpi or pixels per inch. Please review our instructions for export from Photoshop, Illustrator, and PowerPoint here: <https://rupress.org/jgp/pages/submission-guidelines#revised>

When revising your manuscript, please be sure it is a double-spaced MS Word file and that it includes editable tables, if appropriate.

Please submit your revised manuscript via this link:
Link Not Available

Thank you for the opportunity to consider your manuscript.

Sincerely,

Crina Nimigean, Ph.D.
On behalf of Journal of General Physiology

Journal of General Physiology's mission is to publish mechanistic and quantitative molecular and cellular physiology of the highest quality; to provide a best-in-class author experience; and to nurture future generations of independent researchers.

Reviewer #1 (Comments to the Authors):

In the manuscript by Kopec et al, the authors investigated the role of an electrostatic interaction network behind the selectivity filter region of MthK on K⁺ permeation and C-type inactivation. They use a combination of single-channel electrophysiology and molecular dynamics simulations (short and large scale) to study the effect of mutating V55 to a glutamate (to mimic the equivalent position in KcsA) on conductance and open-channel stability. They find that V55E displays lower single-channel open probability characterized by shorter mean open times, the effect is most notable at depolarizing potentials. In MD simulations, the glutamate side-chain adopts two orientations (vertical and horizontal), each revealing a distinct set of interactions with the residues lining the SF cavity and a different number of water molecules in the cavity. While in the vertical orientation, the conductance is lowered, in the horizontal orientation, the stability of the open state is impacted. In longer simulations, pore-widening transitions are observed to a greater frequency in the mutant compared to wt, and is proposed to underlie C-type inactivation in Mthk.

Overall this is an interesting and timely study that further sheds light on potentially divergent mechanisms underlying c-type inactivation in similar-looking K⁺ channel SF regions. Following are some areas that are unclear or need additional discussion. In general, the electrophysiology data and interpretation could be better integrated with MD simulation in the discussion.

1. It is mentioned that V55E may be affecting the extent of AG opening. If this were the case, is there an effect observed on Cd activation?
2. Do single-channel currents for V55E show a lower conductance compared to wt as predicted by the simulations (it appears to be the case visually, but it's hard to assess from the flickery behavior of the channel).
3. Page 7 "When compared to the WT channel, currents in V55E display a similar overall behavior, with an initial increase in magnitude upon the activation gate opening, followed by a decrease at larger openings" This is a follow up from a previous work by the authors, but please provide context here on the activation gate residues, the extent of opening, and current changes etc.
4. Page 16 "we suggest that the reduced Popen of MthK V55E might be partially caused by low permeability of the SF, due to the E55 side chain adopting the 'vertical' orientation." This is unclear to me. The vertical orientation of E55 sidechain appears to stabilize an ion occupancy in S2 and may therefore decrease permeability and single-channel conductance. But how does that decrease Popen? Moreover, in this orientation, larger AG openings do not decrease currents.
5. The authors may want to elaborate more on why the current amplitudes are quite different between CHARMM and AMBER forcefields for the mutants while in the previous reports for WT they appeared comparable. In addition, relative occupancy in the two orientations and subsequent pore dilation are also forcefield dependent.

Reviewer #2 (Comments to the Authors):

In this manuscript the authors undertake simulations and electrophysiology of MthK wildtype and a KcsA-like V55E mutant to describe conduction and selectivity filter gating/C-type inactivation. A total of 900 microseconds of simulation have been used to help describe the experimental phenomena. The authors find that MthK V55E has reduced open probability and that conductance depends on the orientation of the E55 side chain, either being directed upwards H-bonding to D64 (like in KcsA), or laying sideways towards the filter with no obvious H-bonding partner (a conformation not seen before) with wt-like conductance, but increased propensity to inactivate. Perhaps the most striking conclusion is that the authors observe inactivating like changes in MthK wt and mutant channels that they propose are not KcsA-like (with flipped valine carbonyl and pinched lower glycine), but more Kv-like, recently shown to possess a somewhat splayed outer region of the selectivity filter. This is a little surprising given a recent study (Nimigean and coworkers; ref.28) showed that MthK exhibits KcsA-like changes to

the filter due to the influence of ion occupancy, and that some evidence of KcsA-like changes are evident in this manuscript (discussed below), and this apparent disagreement could have been better discussed in this manuscript. The manuscript is otherwise well written, represents a significant amount of work and reveals interesting aspects of MthK structure and function, with some additional points below to address.

The authors find that V55E is more unstable than the wildtype, due to a compromised hydrogen bond network in the alternative horizontal orientation. This is surprising given KcsA with vertical-only E71, when mutated to E71A (eliminating vertical E71-D80) eliminates inactivation. It seems to me V55E is not simply an equivalent of KcsA, but a significant perturbation to MthK structure and activity that is not so obvious. In fact, the remaining differences between KcsA and MthK V55E would be interesting to discuss, especially in the pore helix region, and what effects they may be having on results.

The authors find that the horizontal orientation of E55 increases current in mutant MthK by reducing a free energy barrier in the upper filter. I think it is important to ask the question - how do we know that E55 should remain protonated when in that horizontal orientation? The reason E71 was protonated in KcsA was that it was in the vertical position in a proton sharing interaction with D80. Building the vertical E55 directed at D64 in a protonated state may make sense, but once that interaction has broken and the E55 directs without a proton-sharing partner, instead surrounded by one or a few water molecules, what would cause the large pKa shift to stabilise the protonated form? If instead it were deprotonated with standard state, it would have significant effects on all results (conduction, ion occupancy, water occupancy, stability of E55 conformations) and to the outcomes of this study.

To try to understand reduced currents in mutant E55 vertical models, despite an optimal width of S4, the authors examine a mean single ion density and an equivalent free energy via $-kT \ln(\rho)$, which they argue might be treated as "approximate, one dimensional free energy profiles for ion permeation". While I agree this is a useful calculation, it is important to point out that this is a mean single ion density that hides the detail of the multi-ion process of permeation, and as such cannot be expected to reveal the barriers between states in the actual multi-ion mechanism. However, barriers in $-kT \ln(\rho)$ may indicate a region where ions are disfavoured, such as a particular site. Related, when comparing such profiles, the authors write (p10) that for the horizontal E55 the S4 binding site is "compromised (a free energy minimum is replaced by a maximum), which likely gives rises to other very high barriers...". It is not clear what might be these "other" very high barriers.

Differences between Amber and Charmm simulations are obvious, but the causes for the differences are not discussed. e.g. Simulations of V55E with CHARMM show small currents, likely because Charmm stabilises the vertical E55 whereas Amber fluctuates between these configurations. Charmm simulations also show seemingly much higher levels of the proposed inactivation-like changes. Why would, for example, E55-D64 be more stable in Charmm than Amber? Related, when referring to the CHARMM force field, it is important to note that it may be simulated with or without modifications to ion-carbonyl Lennard-Jones terms, having major effects on permeation. It is presumed that this is the off-the-shelf CHARMM36 with strong ion-carbonyl interactions that raise the ion occupancy, similar to Amber. Fig.5g & h are interesting in this respect, showing so much more of the proposed inactivation widening change in Charmm than in Amber (which exhibits none at all for WT Mthk). Both of these FFs have strong ion-carbonyl terms, and widening happens regardless. To what extent are observations dependent on differences in FF, including those in ion parameters?

The authors note that previous work on KcsA revealed SF pinching (p13), yet this pinching at the central glycine is always associated with valine carbonyl flipping just below it, and for some reason this is not evident in Fig.5B for KcsA. Why is this so? Importantly, Fig.5f for MthK V55E does show this Valine flipping, which is KcsA-like, but I do not think this is discussed (even overlooked). This is important given the paper argues that MthK does not exhibit KcsA-like inactivation. The authors then argue (p17) that both MthK WT and V55E are more comparable with Kv rather than KcsA. Are the authors saying that the KcsA-like changes reported by Nimigean and co-workers (ref.28) are unrelated to inactivation?

Fig.6 is an interesting figure showing how water entry is associated with vertical - horizontal glutamate change, allowing more water to enter, apparently promoting D64 flip and widening of the upper filter. The authors state that similar changes to their filter widening were seen in ref.28, yet I am not sure this was what was concluded in that reference, which found KcsA-like carbonyl flipping and glycine pinching, albeit using simulations on shorter times. I believe disorder and some widening of the upper filter does occur when ions are moved from those sites. In fact, it would be interesting to see how the structural changes seen in this study correlate / are caused by vacating of ions from the upper filter. i.e. Breaking of H bonds behind the filter may be slow, but so may be loss of ions above S2. Perhaps analysis (like in fig6) could include ion occupancy in upper sites as well?

Reviewer #3 (Comments to the Authors):

The Paper addresses the molecular mechanisms of filter gating with the goal of atomistic explanations for this common phenomenon in the selectivity filter of K⁺ channels. The authors employ for this endeavor the well-studied MthK channel as a model system for the filter of K⁺ channels and they correlate the results of single channel recordings with MD simulations. Even though it is highly desirable for the field to see studies in which dynamic information from single channel recordings can be causally related to the results of MD simulations, the present work is not convincing. There are major inconsistencies between

experimental data and results from MD simulations, which prevent a mutual interpretation of the data. Hence the manuscript unfortunately provides, in spite of its interesting approach, no clear-cut information on the mechanisms of filter gating. The authors have to be more precise when they interpret experimental results and computational data with respect to what is related to unitary conductance or gating. I am sorry but with these general criticism and the detailed problems spelled out below I cannot be more positive at this point.

- The single channel data in Fig. 1a show that the mutation affects gating of the channel but at first glance not its unitary conductance. The MD simulations with the computational electrophysiology on the other hand conclude that the mutation reduces the unitary conductance. This discrepancy is not mentioned at all in the paper. Instead, the authors claim that the in silico determined currents for WT and V55E mutant are in very good agreement with experimental data. However, at the same time, the authors refrain from presenting I/V curves in Fig.2. Only the dashed lines in Fig. 2a suggest that the maximum amplitudes for WT and V55E mutant are identical. A closer look at the single channel data however implies that the fluctuations generated by the mutant are very brief. The very basic single channel analysis presented in the manuscript does not exclude that the full channel openings are not fully resolved. Hence the unitary conductance of the mutant might even be larger than that of the WT.

- Can the authors provide a plausible reason why the horizontal rotamer configuration of amino acid E55 (or E71 in KcsA) has not been observed previously in MD simulations or structural data?

- The authors have chosen for the MD simulation a very unfortunate voltage of +300 mV. At this voltage the WT channel and its mutant are in the experiments barely active (Fig. 2c). Also, a linear extrapolation of the data in Fig. 2d implies that the important parameter of mean open life time between wt and mutant is at +300 mV no longer relevant. Hence it is difficult to relate the MD data at +300 mV to the experimental findings. The authors probably argue that the simulation reflects the open structure at 0 mV and that the membrane voltage is only providing the driving force for ion permeation. But there are no data, which confirm that the structure of the channel remains insensitive to voltage. Hence the inactivation of the channel during long simulations could well be the result of a voltage dependency. At this point however it would be difficult to justify the difference in the behavior between wt and mutant; at +300 mV there is presumably no more difference between wt and mutant.

- From the text it is difficult to understand how in Fig.3g (orange circles) the E55 side group can take the vertical rotamer configuration, although it is supposed to be restrained in the simulation in the horizontal rotamer configuration. The same is true for 'E55 vertical' (brown circles), as values also deviate from 1 in some cases. The authors must explain much better in M&M but may be also in the results the nature of the constraints.

- In Fig. 6 the authors attempt to explain the mechanism of inactivation by conformational changes in the selectivity filter. This is potentially interesting, but I have great difficulties in understanding these data. My first problem is that figure 6a,b are so overloaded with information that I barely understand what is going on here. The same is true for Sup. Figs. 13&14. Important details, which the authors explicitly address in the text and which are of great importance for the inferred mechanism, are not clearly visible or have to be guessed by the reader. For example, the entry of another water molecule in the 'shear' subunit marked 1. in Fig.6a cannot be seen at all because the associated data points are completely overlaid by the green circles. Furthermore, it is not at all obvious from the graphs when the flipping of the D64 side group actually occurs. Immediately after the transition from E55 to the horizontal configuration? Or at the end of the trajectory? What is the meaning of the time axis then? Maybe it would make more sense instead of or in addition to the sample trajectories to show in a graph with which probability water enters the space behind the SF after the transition from the vertical to the horizontal configuration and/or with which probability flipping of the D64 side group then occurs.

But even more disturbing is that I do not understand the interpretation of the data in the context of the previous information on the V55E mutant. We have learned before that the vertical positioning of E55 gives low conductance while the horizontal one gives a high conductance. In Fig. 6a the authors now provide data from which they conclude that they might reflect the conformational changes in the filter responsible for inactivation. From what I learned before I would have anticipated to see E55 in the horizontal orientation e.g. the one which favors high conductance from which it transits to the vertical, low conductive orientation. But the data show the opposite: the major change in the structure prior to the flip of D64 (= inactivation) is a fluctuation of E55 from vertical to horizontal. I have been reading this part of the manuscript several times, but I don't find an explanation. My suspicion is that the authors mix up two different concepts here namely unitary conductance and gating. But there is no clear separation between the two concepts in the manuscript.

- The authors derive from the simulation data for the V55E mutant a sequence of events that precede or initiate the transition to the inactive state (vertical E55 configuration -> horizontal E55 configuration -> entry of water molecules into the space behind the SF -> breaking of the H-bridge between Y51 and D64 -> D64 flipping). However, the title of the paper talks about 'Driving forces underlying SF gating in MthK'. Since the configuration change of amino acid E55 plays a central role in the postulated sequence of events, but this amino acid does not occur at all in the WT channel, the reader is left with the question of which events in the WT channel initiate the transition to the inactive state. What do the 7/20 simulations performed using the CHARMM force field reveal in this regard, in which inactivation could also be observed for the WT channel?

We thank all the Reviewers for their comments, which greatly improved our manuscript. Our answers are marked in green.

Reviewer #1 (Comments to the Authors):

In the manuscript by Kopec et al, the authors investigated the role of an electrostatic interaction network behind the selectivity filter region of MthK on K⁺ permeation and C-type inactivation. They use a combination of single-channel electrophysiology and molecular dynamics simulations (short and large scale) to study the effect of mutating V55 to a glutamate (to mimic the equivalent position in KcsA) on conductance and open-channel stability. They find that V55E displays lower single-channel open probability characterized by shorter mean open times, the effect is most notable at depolarizing potentials. In MD simulations, the glutamate side-chain adopts two orientations (vertical and horizontal), each revealing a distinct set of interactions with the residues lining the SF cavity and a different number of water molecules in the cavity. While in the vertical orientation, the conductance is lowered, in the horizontal orientation, the stability of the open state is impacted. In longer simulations, pore-widening transitions are observed to a greater frequency in the mutant compared to wt, and is proposed to underlie C-type inactivation in Mthk.

Overall this is an interesting and timely study that further sheds light on potentially divergent mechanisms underlying c-type inactivation in similar-looking K⁺ channel SF regions. Following are some areas that are unclear or need additional discussion. In general, the electrophysiology data and interpretation could be better integrated with MD simulation in the discussion.

1. It is mentioned that V55E may be affecting the extent of AG opening. If this were the case, is there an effect observed on Cd activation?

We have deleted this statement. We observed in early experiments using V55E channels that P_o with 30 μM Cd²⁺ is nearly the same as with 100 μM Cd²⁺, similar to the Cd²⁺ sensitivity of WT channels (Thomson et al., PNAS 2014, PMID: 24733889), consistent with the idea that V55E channels are maximally activated with 100 μM Cd²⁺. Thus it is unlikely that V55E has a direct effect on the extent of AG opening.

2. Do single-channel currents for V55E show a lower conductance compared to wt as predicted by the simulations (it appears to be the case visually, but it's hard to assess from the flickery behavior of the channel).

We have now quantified unitary currents in V55E channels for comparison with WT channels, and incorporated a new figure (Figure 3) to address this point. V55E channels do exhibit decreased unitary conductance compared to WT, which cannot be entirely accounted for by the very brief nature of V55E openings alone. In each V55E recording, a fraction of longer-duration openings are predicted to reach their full amplitude, and these do not reach the same amplitude as WT openings under the same conditions.

3. Page 7 "When compared to the WT channel, currents in V55E display a similar overall behavior, with an initial increase in magnitude upon the activation gate opening, followed by a decrease at larger openings" This is a follow up from a previous work by the authors, but please provide context here on the activation gate residues, the extent of opening, and current changes etc.

We have added the requested details. This part now reads (page 9 in the new manuscript):

“To probe the effects of the V55E mutation in the MthK channel on an atomistic scale, we used our previous computational protocol to monitor changes in outward current, caused by motions of the inner helices (the activation gate, Fig. 4 A) [22]. Initially, we used the AMBER14sb force field (referred to as AMBER, see Methods for details). When compared to the WT channel, simulated currents at 300 mV in MthK V55E display a similar overall behavior, with an initial increase in magnitude upon the activation gate opening (measured as an average distance between CA of A88 residues from oppositely oriented monomers of MthK, Fig. 4 B), that is between 1.4 - 1.6 nm, from the initial value of 2-3 pA. Around the opening of 1.6 nm, the current reaches its maximum of ~16 pA in WT and ~9 pA in V55E. Further activation gate opening (above the value of 1.6 nm) leads to a current decrease, back to 2-3 pA in both channels. In WT, we have previously traced this behavior to the variations in the width of the S4 binding site, formed exclusively by T59 (Fig. 4 A). Indeed, when T59 CA-CA distance (i.e. the width of the S4 ion binding site) is used as an opening coordinate, currents in V55E show a similar trend upon S4 opening as in WT (Fig. 4 C).”

4. Page 16 "we suggest that the reduced Popen of MthK V55E might be partially caused by low permeability of the SF, due to the E55 side chain adopting the 'vertical' orientation." This is unclear to me. The vertical orientation of E55 sidechain appears to stabilize an ion occupancy in S2 and may therefore decrease permeability and single-channel conductance. But how does that decrease Popen? Moreover, in this orientation, larger AG openings do not decrease currents.

We thank the Reviewer for pointing out this inconsistency, and we have now deleted this sentence. The new version of the manuscript conveys our point more clearly: together with the new single channel recordings, we now propose a dual effect of the E55 side chain: i) reduced single channel conductance, as a consequence of the 'vertical' orientation of the E55 side chain (from shorter MD simulations), ii) reduced Popen as a consequence of the enhanced SF gating (inactivation) caused by the presence of the E55 side chain, affecting several key interactions behind the SF (from longer MD simulations).

5. The authors may want to elaborate more on why the current amplitudes are quite different between CHARMM and AMBER forcefields for the mutants while in the previous reports for WT they appeared comparable. In addition, relative occupancy in the two orientations and subsequent pore dilation are also forcefield dependent.

Indeed, we did not discuss in detail the differences between force fields in our previous version. We expanded on this topic, also taking into account the requests of the Reviewer #2. We have addressed the differences between the force fields in a few places in the new manuscript.

Page 14:

The reduced outward conductance of MthK V55E, as compared to MthK WT, predicted by both force fields, is in very good agreement with our electrophysiological unitary conductance measurements (Fig. 3). The major difference between the force fields in this set of simulations is the fact that in AMBER both 'vertical' and 'horizontal' orientations of the E55 side chain are frequently visited (Fig. 4 G), whereas CHARMM has a strong preference for the 'vertical' orientation (Fig. 5 C). This, in turn, results in very different simulated outward currents when no restraints on the E55 side chain are used, because, as shown above, the orientation of this side chain is a major factor regulating ion permeation. The origins of this distinct behavior, i.e. why the horizontal orientation is frequently visited in the AMBER force field, but not in CHARMM, are not obvious. It is unlikely however, that this difference arises from variations in the non-bonded interactions. Even though the partial charges and Lennard-Jones parameters are different between AMBER and CHARMM, the resulting interaction energies of the

D64-E55 hydrogen bond (E55 in the 'vertical' orientation) are actually quite similar and favorable (Supplementary Fig. 1 B, C). However, we suspected that the energetics of the two dihedrals, that involve the hydrogen atom from the protonated E55 side chain, might also play a role (Supplementary Fig. 1 D). Indeed, these two dihedrals are low in energy (0-2 kJ/mol) in the 'vertical' orientation in the CHARMM force field (Supplementary Fig. 1 E). In contrast however, they are energetically less favorable in the same orientation in the AMBER force field, as one of the dihedrals has an energy of ~17-18 kJ/mol (Supplementary Fig. 1 F). Only after the transition to the 'horizontal' orientation, the energy drops to values close to 0 kJ/mol, likely explaining the frequent transition from 'vertical' to 'horizontal' orientation in the AMBER force field (and lack of thereof in CHARMM). Further research is needed to unravel which of these force field parametrizations describe the E55 dynamics more realistically.

The orientational preference of E55 side chain in AMBER (but not in CHARMM) is also dependent on the activation gate opening (Fig. 4 G), which we attribute to the allosteric coupling between the gate and the SF, and the residues nearby [22,23]: a variation in the activation gate width affects the SF width as well, which in turn might impact the delicate balance between non-bonded interactions and torsional energetics, dictating the overall E55 side chain dynamics.

Page 21:

"Summarizing, we discovered that the V55E mutation reduces the stability of the conducting conformation of the SF in the MthK channel, by promoting water entry behind the SF, leading to the breakage of the Y51-D64 hydrogen bond and subsequent flipping of the D64 side chains. This entry of additional water molecules likely correlates with the 'horizontal' orientation of the E55 side chain. Even though both force fields predicted an overall similar effect of the V55E on the SF stability, there are important differences. Firstly, as already described for shorter simulations, the AMBER force field strongly prefers the 'horizontal' orientation of the E55 side chain, due to the differences in dihedral parametrization (Supplementary Fig. 1). Secondly, the distribution of K⁺ ions in the SF, and the effect of the mutation on the overall number of K⁺ ions in the SF, vary between the force fields (Supplementary Fig. 27). Importantly, the D64 flipping in MthK V55E simulated with the AMBER force field strongly correlates with the unbinding of K⁺ ions from the top of the SF. Thirdly, the same aspartate flipping and inactivation mechanism as in MthK V55E is seen in simulations of MthK WT, but only when the CHARMM force field is used.

We further investigated the possible reasons for the last discrepancy. We evaluated two factors, possibly affecting the D64 side chain dynamics: i) the strength of the Y51-D64 hydrogen bond (weaker interactions would promote the aspartate flip, Supplementary Fig. 29 A-C); and ii) the energetics of dihedrals defined by the heavy atoms of the D64 side chain (high energy orientations of the D64 side chains would contribute to a lower stability of a given state, Supplementary Fig. 29 D-F). From this analysis, the differences between the two force fields become more clear - surprisingly, the Y51-D64 interactions are actually stronger in CHARMM than in AMBER (opposed to the inactivation rate trend). Strikingly however, one of the two dihedrals of the D64 side chain has a high energy (up to ca. 19 kJ/mol) in the CHARMM force field, when the D64 side chain is in the non-flipped orientation, and is able to adopt a lower energy conformation only after the aspartate flip (Supplementary Fig. 29 E). In stark contrast, in AMBER simulations, both dihedrals are low in energy when the D64 side chain is in the non-flipped orientation (Supplementary Fig. 29 F). These force field differences, primarily in the dihedral parametrization, resulting in different preferred orientations of the D64 side chain, and, to some degree, in the K⁺ occupancy of the individual ion binding sites (high S0 occupancy in AMBER, Supplementary Fig. 27), likely explain the much higher propensity of the MthK WT SF inactivation in MD simulations with the CHARMM force field. Interestingly, these inactivated conformations resemble experimentally solved structures of Kv channels (Fig. 6).

We wondered how these two factors are affected by the V55E mutation. The energetics of the dihedrals is only slightly affected, in both force fields (Supplementary Fig. 30), although the high energy dihedral in

CHARMM is even higher in energy in MthK V55E. The energetics of the Y51-D64 and E55 (or water in MthK WT)-D64 hydrogen bonds is also affected, especially in the CHARMM force field: most of these interactions are stronger in MthK WT than in the V55E mutant (Supplementary Fig. 30 C), which could also contribute to the lower stability of the conducting state in MthK V55E in this force field. In the AMBER force field, the trend is not so clear: some interactions are stronger in MthK WT and some in MthK V55E. Therefore, in MD simulations of MthK V55E with the AMBER force field, we assign the unbinding of K⁺ ions from the top of the SF, and the entry of additional water molecules behind the SF as the primary cause of the aspartate flipping and SF inactivation.”

This analysis involves the new Figure 7, Supplementary Figure 1 and Supplementary Figures 27-30.

Reviewer #2 (Comments to the Authors):

In this manuscript the authors undertake simulations and electrophysiology of MthK wildtype and a KcsA-like V55E mutant to describe conduction and selectivity filter gating/C-type inactivation. A total of 900 microseconds of simulation have been used to help describe the experimental phenomena. The authors find that MthK V55E has reduced open probability and that conductance depends on the orientation of the E55 side chain, either being directed upwards H-bonding to D64 (like in KcsA), or laying sideways towards the filter with no obvious H-bonding partner (a conformation not seen before) with wt-like conductance, but increased propensity to inactivate. Perhaps the most striking conclusion is that the authors observe inactivating like changes in MthK wt and mutant channels that they propose are not KcsA-like (with flipped valine carbonyl and pinched lower glycine), but more Kv-like, recently shown to possess a somewhat splayed outer region of the selectivity filter. This is a little surprising given a recent study (Nimigeon and coworkers; ref.28) showed that MthK exhibits KcsA-like changes to the filter due to the influence of ion occupancy, and that some evidence of KcsA-like changes are evident in this manuscript (discussed below), and this apparent disagreement could have been better discussed in this manuscript. The manuscript is otherwise well written, represents a significant amount of work and reveals interesting aspects of MthK structure and function, with some additional points below to address.

The authors find that V55E is more unstable than the wildtype, due to a compromised hydrogen bond network in the alternative horizontal orientation. This is surprising given KcsA with vertical-only E71, when mutated to E71A (eliminating vertical E71-D80) eliminates inactivation. It seems to me V55E is not simply an equivalent of KcsA, but a significant perturbation to MthK structure and activity that is not so obvious. In fact, the remaining differences between KcsA and MthK V55E would be interesting to discuss, especially in the pore helix region, and what effects they may be having on results.

We are thankful for this very good suggestion. Now, we discuss the differences between KcsA and MthK in the Discussion section, and highlight them in the new Supplementary Figure 32.

The authors find that the horizontal orientation of E55 increases current in mutant MthK by reducing a free energy barrier in in the upper filter. I think it is important to ask the question - how do we know that E55 should remain protonated when in that horizontal orientation? The reason E71 was protonated in KcsA was that it was in the vertical position in a proton sharing interaction with D80. Building the vertical E55 directed at D64 in a protonated state may make sense, but once that interaction has broken and the E55 directs without a proton-sharing partner, instead surrounded by one or a few water molecules, what would cause the large pKa shift to stabilise the protonated form? If instead it were deprotonated with standard

state, it would have significant effects on all results (conduction, ion occupancy, water occupancy, stability of E55 conformations) and to the outcomes of this study.

We thank the Reviewer for this excellent point. Following the suggestions, we performed free energy calculations, using a nonequilibrium protocol, to assess the free energy difference associated with E55 deprotonation, when it is in the 'horizontal' orientation. The results (Supplementary Figure 31) clearly show a large free energy cost (more than 40 kJ/mol) of E55 deprotonation, even when it is surrounded with a few water molecules in its deprotonated form. This cost corresponds to a shift in pKa of a glutamate by ca. 7-8 units, therefore, assuming the reference pKa of a glutamate being 4, E55 would prefer its protonated form also in its horizontal orientation. Importantly, a similar free energy difference is obtained using both AMBER and CHARMM force fields. These additional calculations are now described in separate paragraphs in Results and Methods sections

To try to understand reduced currents in mutant E55 vertical models, despite an optimal width of S4, the authors examine a mean single ion density and an equivalent free energy via $-kT \ln(\rho)$, which they argue might be treated as "approximate, one dimensional free energy profiles for ion permeation". While I agree this is a useful calculation, it is important to point out that this is a mean single ion density that hides the detail of the multi-ion process of permeation, and as such cannot be expected to reveal the barriers between states in the actual multi-ion mechanism. However, barriers in $-kT \ln(\rho)$ may indicate a region where ions are disfavoured, such as a particular site.

We agree with the reviewer and we added these important points to the manuscript. This part now reads (pages 11-12):

"To gain additional insights, we calculated negative logarithms of potassium densities in the SF that might be treated as approximate, one dimensional free energy profiles for ion permeation (Fig. 4 I). It is important to note that such profiles do not reflect the underlying multi-ion permeation process, including permeation barriers. They indicate the regions in the SF where ion binding is favored and disfavored."

Related, when comparing such profiles, the authors write (p10) that for the horizontal E55 the S4 binding site is "compromised (a free energy minimum is replaced by a maximum), which likely gives rises to other very high barriers...". It is not clear what might be these "other" very high barriers.

We have now clarified this part (page 12, and Figure 4 i):

"which likely results in to two other high barriers, namely between the S3 and S2 as well as S2 and S1 ion binding sites, drastically decreasing the current at larger openings in 'E55 horizontal' simulations."

Differences between Amber and Charmm simulations are obvious, but the causes for the differences are not discussed. e.g. Simulations of V55E with CHARMM show small currents, likely because Charmm stabilises the vertical E55 whereas Amber fluctuates between these configurations. Charmm simulations also show seemingly much higher levels of the proposed inactivation-like changes. Why would, for example, E55-D64 be more stable in Charmm than Amber?

We have now looked at the differences between AMBER and CHARMM in more detail (please also see point 5 from Reviewer 1). On the page 14, we discuss the different propensity of the 'vertical' to 'horizontal' transition in the two force fields:

“The reduced outward conductance of MthK V55E, as compared to MthK WT, predicted by both force fields, is in very good agreement with our electrophysiological unitary conductance measurements (Fig. 3). The major difference between the force fields in this set of simulations is the fact that in AMBER both ‘vertical’ and ‘horizontal’ orientations of the E55 side chain are frequently visited (Fig. 4 G), whereas CHARMM has a strong preference for the ‘vertical’ orientation (Fig. 5 C). This, in turn, results in very different simulated outward currents when no restraints on the E55 side chain are used, because, as shown above, the orientation of this side chain is a major factor regulating ion permeation. The origins of this distinct behavior, i.e. why the horizontal orientation is frequently visited in the AMBER force field, but not in CHARMM, are not obvious. It is unlikely however, that this difference arises from variations in the non-bonded interactions. Even though the partial charges and Lennard-Jones parameters are different between AMBER and CHARMM, the resulting interaction energies of the D64-E55 hydrogen bond (E55 in the ‘vertical’ orientation) are actually quite similar and favorable (Supplementary Fig. 1 B, C). However, we suspected that the energetics of the two dihedrals, that involve the hydrogen atom from the protonated E55 side chain, might also play a role (Supplementary Fig. 1 D). Indeed, these two dihedrals are low in energy (0-2 kJ/mol) in the ‘vertical’ orientation in the CHARMM force field (Supplementary Fig. 1 E). In contrast however, they are energetically less favorable in the same orientation in the AMBER force field, as one of the dihedrals has an energy of ~17-18 kJ/mol (Supplementary Fig. 1 F). Only after the transition to the ‘horizontal’ orientation, the energy drops to values close to 0 kJ/mol, likely explaining the frequent transition from ‘vertical’ to ‘horizontal’ orientation in the AMBER force field (and lack of thereof in CHARMM). Further research is needed to unravel which of these force field parametrizations describe the E55 dynamics more realistically. The orientational preference of E55 side chain in AMBER (but not in CHARMM) is also dependent on the activation gate opening (Fig. 4 G), which we attribute to the allosteric coupling between the gate and the SF, and the residues nearby [22,23]: a variation in the activation gate width affects the SF width as well, which in turn might impact the delicate balance between non-bonded interactions and torsional energetics, dictating the overall E55 side chain dynamics.”

On the page 21 we discuss first why the aspartate flipping (and filter inactivation) occurs much more often in CHARMM than in AMBER for MthK WT:

“Summarizing, we discovered that the V55E mutation reduces the stability of the conducting conformation of the SF in the MthK channel, by promoting water entry behind the SF, leading to the breakage of the Y51-D64 hydrogen bond and subsequent flipping of the D64 side chains. This entry of additional water molecules likely correlates with the ‘horizontal’ orientation of the E55 side chain. Even though both force fields predicted an overall similar effect of the V55E on the SF stability, there are important differences. Firstly, as already described for shorter simulations, the AMBER force field strongly prefers the ‘horizontal’ orientation of the E55 side chain, due to the differences in dihedral parametrization (Supplementary Fig. 1). Secondly, the distribution of K⁺ ions in the SF, and the effect of the mutation on the overall number of K⁺ ions in the SF, vary between the force fields (Supplementary Fig. 27). Importantly, the D64 flipping in MthK V55E simulated with the AMBER force field strongly correlates with the unbinding of K⁺ ions from the top of the SF. Thirdly, the same aspartate flipping and inactivation mechanism as in MthK V55E is seen in simulations of MthK WT, but only when the CHARMM force field is used. We further investigated the possible reasons for the last discrepancy. We evaluated two factors, possibly affecting the D64 side chain dynamics: i) the strength of the Y51-D64 hydrogen bond (weaker interactions would promote the aspartate flip, Supplementary Fig. 29 A-C); and ii) the energetics of dihedrals defined by the heavy atoms of the D64 side chain (high energy orientations of the D64 side chains would contribute to a lower stability of a given state, Supplementary Fig. 29 D-F). From this analysis, the differences between the two force fields become more clear - surprisingly, the Y51-D64 interactions are

actually stronger in CHARMM than in AMBER (opposed to the inactivation rate trend). Strikingly however, one of the two dihedrals of the D64 side chain has a high energy (up to ca. 19 kJ/mol) in the CHARMM force field, when the D64 side chain is in the non-flipped orientation, and is able to adopt a lower energy conformation only after the aspartate flip (Supplementary Fig. 29 E). In stark contrast, in AMBER simulations, both dihedrals are low in energy when the D64 side chain is in the non-flipped orientation (Supplementary Fig. 29 F). These force field differences, primarily in the dihedral parametrization, resulting in different preferred orientations of the D64 side chain, and, to some degree, in the K⁺ occupancy of the individual ion binding sites (high S0 occupancy in AMBER, Supplementary Fig. 27), likely explain the much higher propensity of the MthK WT SF inactivation in MD simulations with the CHARMM force field. Interestingly, these inactivated conformations resemble experimentally solved structures of Kv channels (Fig. 6).

And also why it is the case for MthK V55E:

We wondered how these two factors are affected by the V55E mutation. The energetics of the dihedrals is only slightly affected, in both force fields (Supplementary Fig. 30), although the high energy dihedral in CHARMM is even higher in energy in MthK V55E. The energetics of the Y51-D64 and E55 (or water in MthK WT)-D64 hydrogen bonds is also affected, especially in the CHARMM force field: most of these interactions are stronger in MthK WT than in the V55E mutant (Supplementary Fig. 30 C), which could also contribute to the lower stability of the conducting state in MthK V55E in this force field. In the AMBER force field, the trend is not so clear: some interactions are stronger in MthK WT and some in MthK V55E. Therefore, in MD simulations of MthK V55E with the AMBER force field, we assign the unbinding of K⁺ ions from the top of the SF, and the entry of additional water molecules behind the SF as the primary cause of the aspartate flipping and SF inactivation.”

Related, when referring to the CHARMM force field, it is important to note that it may be simulated with or without modifications to ion-carbonyl Lennard-Jones terms, having major effects on permeation. It is presumed that this is the off-the-shelf CHARMM36 with strong ion-carbonyl interactions that raise the ion occupancy, similar to Amber.

It is true, we added these details to Method section:

We used the same setup as we used for MthK WT in our recent publication [22], where we thoroughly characterized ion permeation and selectivity filter-activation gate coupling, using the CHARMM36m (referred to as CHARMM) [49,50] and AMBER14sb (referred to as AMBER) [51] force fields. We did not use any further modifications of the carbonyl oxygen - potassium ion interactions, as non-modified force fields seem to better reproduce solvation free energies [52].

Indeed, the total number of ions is similar and high (~3 ions bound, Supplementary Figure 27), which suggest strong ion-carbonyl interactions.

Fig.5g & h are interesting in this respect, showing so much more of the proposed inactivation widening change in Charmm than in Amber (which exhibits none at all for WT Mthk). Both of these FFs have strong ion-carbonyl terms, and widening happens regardless. To what extent are observations dependent on differences in FF, including those in ion parameters?

We believe that our new analyses in Figure 7 and Supplementary Figure 27-30, and described in the Results section of the new manuscript cover this point. We found out that, apart from the ion-SF interactions, the dihedral energetics of D64 side chains plays a major role in the inactivation rate seen in MD simulations, together with the strength of hydrogen bonds behind the SF (Supplementary Figures 29 and 30).

The authors note that previous work on KcsA revealed SF pinching (p13), yet this pinching at the central glycine is always associated with valine carbonyl flipping just below it, and for some reason this is not evident in Fig.5B for KcsA. Why is this so? Importantly, Fig.5f for MthK V55E does show this Valine flipping, which is KcsA-like, but I do not think this is discussed (even overlooked). This is important given the paper argues that MthK does not exhibit KcsA-like inactivation. The authors then argue (p17) that both MthK WT and V55E are more comparable with Kv rather than KcsA. Are the authors saying that the KcsA-like changes reported by Nimigean and co-workers (ref.28) are unrelated to inactivation?

We apologize for this confusion in the previous version of the manuscript. Indeed, we focused mostly on the changes occurring in the top part of the SF, and we somewhat overlooked the valine carbonyl flipping. We have remade the b panel in the new Figure 6, which now shows the valine flipping more clearly. Also we discuss the fact that valine flipping occurs often in our simulations, as evident in the new Figure 6 h,i and Supplementary Figure 26 f, h, i. We mention now throughout the manuscript that the inactivation changes we observed in MthK channels do bear some similarities with KcsA inactivation, due to valine flipping occurring in both channels.

Fig.6 is an interesting figure showing how water entry is associated with vertical - horizontal glutamate change, allowing more water to enter, apparently promoting D64 flip and widening of the upper filter. The authors state that similar changes to their filter widening were seen in ref.28, yet I am not sure this was what was concluded in that reference, which found KcsA-like carbonyl flipping and glycine pinching, albeit using simulations on shorter times. I believe disorder and some widening of the upper filter does occur when ions are moved from those sites. In fact, it would be interesting to see how the structural changes seen in this study correlate / are caused by vacating of ions from the upper filter. i.e. Breaking of H bonds behind the filter may be slow, but so may be loss of ions above S2. Perhaps analysis (like in fig6) could include ion occupancy in upper sites as well?

We are thankful for this very important point, and we added the requested occupancy of the S0 and S1 ion binding sites just before the inactivation event in the new Figure 7. Indeed, it shows that, especially in MthK V55E simulated with the AMBER force field, there is a sharp decrease in potassium occupancy above S0.

Reviewer #3 (Comments to the Authors):

The Paper addresses the molecular mechanisms of filter gating with the goal of atomistic explanations for this common phenomenon in the selectivity filter of K⁺ channels. The authors employ for this endeavor the well-studied MthK channel as a model system for the filter of K⁺ channels and they correlate the results of single channel recordings with MD simulations. Even though it is highly desirable for the field to see studies in which dynamic information from single channel recordings can be causally related to the results of MD simulations, the present work is not convincing. There are major inconsistencies between experimental data and results from MD simulations, which prevent a mutual interpretation of the data. Hence the manuscript unfortunately provides, in spite of its interesting approach, no clear-cut information on the mechanisms of filter gating. The authors have to be more precise when they interpret experimental results and computational data with respect to what is related to unitary conductance or gating. I am sorry

but with these general criticism and the detailed problems spelled out below I cannot be more positive at this point.

- The single channel data in Fig. 1a show that the mutation affects gating of the channel but at first glance not its unitary conductance. The MD simulations with the computational electrophysiology on the other hand conclude that the mutation reduces the unitary conductance. This discrepancy is not mentioned at all in the paper. Instead, the authors claim that the in silico determined currents for WT and V55E mutant are in very good agreement with experimental data. However, at the same time, the authors refrain from presenting I/V curves in Fig.2. Only the dashed lines in Fig. 2a suggest that the maximum amplitudes for WT and V55E mutant are identical. A closer look at the single channel data however implies that the fluctuations generated by the mutant are very brief. The very basic single channel analysis presented in the manuscript does not exclude that the full channel openings are not fully resolved. Hence the unitary conductance of the mutant might even be larger than that of the WT.

We thank the reviewer for this comment. First (along with the comment of Reviewer 1), it brought to our attention that the traces in the initial version of Figure 2 had inadvertently been scaled to be uniform with the WT current amplitude. This has been rectified, so that both WT and V55E currents in Fig 2A correspond to the scale bar at the lower left of the panel. We noticed that the mean open and closed times presented in the initial version of Fig 2D-E were not analyzed using the full bandwidth of the recordings, and these have now been re-analyzed using low-pass filtering of 1 kHz. This resulted in slightly briefer mean open and closed times, but did not alter any conclusions drawn from these data. In addition, we expanded our electrophysiological analysis and we have now quantified unitary currents in V55E channels for comparison with WT channels, and incorporated a new figure (Figure 3). Here it can be seen more clearly that V55E channels exhibit decreased unitary conductance compared to WT, which cannot be accounted for by low-pass filtering alone. In each V55E recording, a fraction of longer-duration openings are predicted to reach their full amplitude, and these do not reach the same amplitude as WT openings under the same conditions. We incorporate additional text in the Results to present these data and our reasoning.

- Can the authors provide a plausible reason why the horizontal rotamer configuration of amino acid E55 (or E71 in KcsA) has not been observed previously in MD simulations or structural data?

We thank for this very good point, and we added a following paragraph to the Discussion section (page 25):

“It is interesting that the ‘horizontal’ orientation of the E55 side chain (or E71 in KcsA) has not been seen before, neither in structural work nor in MD simulations. We suggest the following reasons for this situation. First, KcsA channels have been mostly simulated with the CHARMM force field (with some recent exceptions [47]), in which the side chain of protonated E71 has a much lower propensity of adopting the ‘horizontal’ orientation. Second, our analysis suggests that the ‘horizontal’ orientation is favored for the larger openings of the activation gate. Most of the KcsA structures have been solved in the closed state, and in those solved in the open state, protonated glutamate is replaced by alanine (the E71A construct), to remove C-type inactivation. Third, as discussed above, MthK V55E and KcsA channels are quite different in their pore region (Supplementary Fig 32), thus it is possible that the ‘horizontal’ orientation is more common for MthK V55E than for KcsA. Our work is the first dealing with the MthK V55E channel, for which there is currently no structural data. We cannot however fully rule out a possibility that the high frequency of the ‘horizontal’ orientation is an artifact of the AMBER force field, and we recommend further work in this direction.”

- The authors have chosen for the MD simulation a very unfortunate voltage of +300 mV. At this voltage the WT channel and its mutant are in the experiments barely active (Fig. 2c). Also, a linear extrapolation of the data in Fig. 2d implies that the important parameter of mean open life time between wt and mutant is at +300 mV no longer relevant. Hence it is difficult to relate the MD data at +300 mV to the experimental findings.

The authors probably argue that the simulation reflects the open structure at 0 mV and that the membrane voltage is only providing the driving force for ion permeation. But there are no data, which confirm that the structure of the channel remains insensitive to voltage. Hence the inactivation of the channel during long simulations could well be the result of a voltage dependency. At this point however it would be difficult to justify the difference in the behavior between wt and mutant; at +300 mV there is presumably no more difference between wt and mutant.

We have repeated all our long simulations at a reduced voltage of 150 mV. These simulations show an overall behavior similar to ones at 300 mV, but the inactivation rate is lowered (Figure 6, l,m), in good agreement with electrophysiology. Similarly, the ion occupancy patterns and the effect of the mutation on them is comparable between the two voltages (Supplementary Fig. 27). We therefore believe that our simulations at 300 mV captured the essential features of both inactivation and ion permeation.

- From the text it is difficult to understand how in Fig.3g (orange circles) the E55 side group can take the vertical rotamer configuration, although it is supposed to be restrained in the simulation in the horizontal rotamer configuration. The same is true for 'E55 vertical' (brown circles), as values also deviate from 1 in some cases. The authors must explain much better in M&M but may be also in the results the nature of the constraints.

We added the detailed description of the restraints used in the Methods section. In the Amber simulations these are exclusively relatively weak distance restraints, therefore they are overcome from time to time.

- In Fig. 6 the authors attempt to explain the mechanism of inactivation by conformational changes in the selectivity filter. This is potentially interesting, but I have great difficulties in understanding these data. My first problem is that figure 6a,b are so overloaded with information that I barely understand what is going on here. The same is true for Sup. Figs. 13&14. Important details, which the authors explicitly address in the text and which are of great importance for the inferred mechanism, are not clearly visible or have to be guessed by the reader. For example, the entry of another water molecule in the 'shear' subunit marked 1. in Fig.6a cannot be seen at all because the associated data points are completely overlaid by the green circles. Furthermore, it is not at all obvious from the graphs when the flipping of the D64 side group actually occurs. Immediately after the transition from E55 to the horizontal configuration? Or at the end of the trajectory? What is the meaning of the time axis then? Maybe it would make more sense instead of or in addition to the sample trajectories to show in a graph with which probability water enters the space behind the SF after the transition from the vertical to the horizontal configuration and/or with which probability flipping of the D64 side group then occurs.

We apologize for the confusing Figure and we have redone it from scratch following the reviewer's suggestions. In the new Figure 7, we show 50ns fragments before the D64 flip (so the 0 point at the time axis is aligned with the D64 flipping event). The probabilities of various molecular events, i.e. water entry behind the SF, water entering the SF, potassium ion unbinding from the top of the SF and E55 adopting the 'horizontal' orientation are then plotted. We believe that the new figure and the new text in the Result section adequately explains the proposed mechanism of inactivation.

But even more disturbing is that I do not understand the interpretation of the data in the context of the previous information on the V55E mutant. We have learned before that the vertical positioning of E55 gives low conductance while the horizontal one gives a high conductance. In Fig. 6a the authors now provide data from which they conclude that they might reflect the conformational changes in the filter responsible for inactivation. From what I learned before I would have anticipated to see E55 in the horizontal orientation e.g. the one which favors high conductance from which it transits to the vertical, low conductive orientation. But the data show the opposite: the major change in the structure prior to the flip of D64 (= inactivation) is a fluctuation of E55 from vertical to horizontal. I have been reading this part of the manuscript several times, but I don't find an explanation. My suspicion is that the authors mix up two different concepts here namely unitary conductance and gating. But there is no clear separation between the two concepts in the manuscript.

We thank the reviewer for this comment and we agree that the separation between the conductance and gating concepts was not very clear in the previous version. We have now rewritten several parts of the manuscript and added additional experimental and computational data to make it clearer:

The 'shorter' simulations (Figures 4 and 5 in the new version) probe the effect of the mutation (and, subsequently, E55 side chain orientation) on the channel conductance, and these data are directly compared to the new experimental conductance measurements (Figure 3).

The 'longer' simulations (Figures 6 and 7) probe the effect of the mutation on the SF stability, that we link to gating, and we compare with the channel gating measurements (Figure 2).

Of particular interest, both simulation parts are now in good agreement with the experimental data.

- The authors derive from the simulation data for the V55E mutant a sequence of events that precede or initiate the transition to the inactive state (vertical E55 configuration -> horizontal E55 configuration -> entry of water molecules into the space behind the SF -> breaking of the H-bridge between Y51 and D64 -> D64 flipping). However, the title of the paper talks about 'Driving forces underlying SF gating in MthK'. Since the configuration change of amino acid E55 plays a central role in the postulated sequence of events, but this amino acid does not occur at all in the WT channel, the reader is left with the question of which events in the WT channel initiate the transition to the inactive state. What do the 7/20 simulations performed using the CHARMM force field reveal in this regard, in which inactivation could also be observed for the WT channel?

We apologize for the confusion and agree that we did not spell it out clearly in the previous version of the manuscript.

We show now in the new Figure 7 also the inactivation process in MthK WT simulated with the CHARMM force field. We see that in both MthK WT and MthK V55E the inactivation mechanism is shared and depends on flipping of the D64 side chain and widening of the SF. This mechanism is enhanced in the V55E mutant, by increased probabilities of the water flux behind the SF and/or reduced number of potassium ions bound to the top of the SF, which is a consequence of the rotameric transition of the E55 side chain.

March 2, 2023

Dr. Wojciech Kopec
Max Planck Institute for Multidisciplinary Sciences
Theoretical and Computational Biophysics
Am Fassberg 11
Goettingen 37077
Germany

Re: 202213166R1

Dear Dr. Kopec,

Thank you for submitting your manuscript, entitled "Driving Forces underlying Selectivity Filter Gating in the MthK Potassium Channel" to JGP. Your manuscript has now been seen by the same 3 reviewers, whose comments are appended below. You will see that the reviewers were in general very happy with your revisions and enthusiastic about the potential impact of your manuscript. However, as reviewer 2 points out, there are still some concerns that need to be addressed before the manuscript is acceptable for publication, in particular about the validity of the existing analysis of the single-channel currents in the mutant channels and how this may impact your conclusions about the lower mutant channel conductance predicted by the computational data.

We hope that you will be able to submit a revised manuscript that addresses these points, which we believe will pose no problems, and which may be re-reviewed. Based on the scope of the requested changes, we typically anticipate that the revision process will take no longer than 2 months, however, we understand you may need additional time to work on your resubmission to JGP. We therefore ask that you simply keep us informed as to a realistic submission timeline that is appropriate for your particular circumstances. In addition, please do not hesitate to contact me (via the editorial office) if you feel that a discussion of the reviewers' and editors' comments would be helpful.

Please submit your revised manuscript via the link below, along with a point-by-point letter that details your response to the reviewers' and editors' comments, as well as a copy of the text with alterations highlighted (boldfaced or underlined). If the article is eventually accepted, it would include a 'revised date' as well as submitted and accepted dates. If we do not receive the revised manuscript within one year, we will regard the article as having been withdrawn. We would be willing to receive a revision of the manuscript at a later time, but the manuscript will then be treated as a new submission, with a new manuscript number.

Please pay particular attention to recent changes to our instructions to authors in the following sections: Data presentation, Blinding and randomization and Statistical analysis, under Materials and Methods, as shown here: <https://rupress.org/jgp/pages/submission-guidelines#prepare>. Re-review will be contingent on inclusion of the required information (including for data added during revision) and demonstration of the experimental reproducibility of the results (i.e., all experimental data verified in at least 2 independent experiments).

Please note, JGP now requires authors to submit Source Data used to generate figures containing gels and Western blots with all revised manuscripts (when applicable). This Source Data consists of fully uncropped and unprocessed images for each gel/blot displayed in the main and supplemental figures. If your paper includes cropped gel and/or blot images, please be sure to provide one Source Data file for each figure that contains gels and/or blots along with your revised manuscript files. File names for Source Data figures should be alphanumeric without any spaces or special characters (i.e., SourceDataF#, where F# refers to the associated main figure number or SourceDataFS# for those associated with Supplementary figures). The lanes of the gels/blots should be labeled as they are in the associated figure, the place where cropping was applied should be marked (with a box), and molecular weight/size standards should be labeled wherever possible. Source Data files will be made available to reviewers during evaluation of revised manuscripts and, if your paper is eventually published in JGP, the files will be directly linked to specific figures in the published article.

Source Data Figures should be provided as individual PDF files (one file per figure). Authors should endeavor to retain a minimum resolution of 300 dpi or pixels per inch. Please review our instructions for export from Photoshop, Illustrator, and PowerPoint here: <https://rupress.org/jgp/pages/submission-guidelines#revised>

Whilst you are revising your manuscript, we ask that you consider whether you have any artwork that might be suitable for the cover of JGP. Microscopy images are particularly good for cover artwork, but other types of image can be very effective, so we encourage you to be creative. Please don't restrict yourself to images from the paper; an image that is relevant to the work described would be just as suitable. Images should be a minimum resolution of 300 dpi. To see recent examples, visit the following page and click on 'Show covers? Yes': <https://jgp.rupress.org/content/by/year>

Thank you for submitting your interesting research to JGP.

Please submit your revised manuscript, and any associated files, via this link:
Link Not Available

Sincerely,

Crina Nimigean, Ph.D.
On behalf of Journal of General Physiology

Journal of General Physiology's mission is to publish mechanistic and quantitative molecular and cellular physiology of the highest quality; to provide a best-in-class author experience; and to nurture future generations of independent researchers.

Reviewer #1 (Comments to the Authors):

The authors have addressed my comments appropriately.
I have nothing more to add.

Reviewer #2 (Comments to the Authors):

The authors have addressed my main queries and this has improved the manuscript. I appreciate the resolution of the issue where the paper seemed to report that MthK was not acting in a KcsA-like way in terms of carbonyl flipping, when the simulations seemed to be doing just that. They have also discussed the involvement of ions, relating to past studies of MthK inactivation. Moreover, they have addressed the force field dependence to some extent. They have performed analysis and discussed differences in Amber and Charmm, clarifying an aspect that was previously left open. The energy differences, particularly in dihedral terms, are surprising and worthwhile pointing out to the readers, needed further studies in future. The authors have importantly checked the protonation state of E55, to be sure it does not play a role. While this KcsA-like Mthk mutant may behave differently to KcsA, and some results may depend on force field, I believe the study is interesting in that it sheds light on how interactions behind the filter can control K channel conduction.

Reviewer #3 (Comments to the Authors):

Report on manuscript „Driving force underlying selectivity filter gating...." by Kopec and coworkers.

The revision of a previous version of this work has greatly improved the manuscript. In principle I keep on supporting publication of the work but before being accepted the authors need to rectify some crucial issues.

- 1) Page 5, ref 19. This reference addresses filter gating in a K2P channel. This type of channels and the information on its filter gating is not considered in the present paper. Hence the reference should be deleted, or the authors have to include the mechanism of filter gating in K2P channels throughout the manuscript.
- 2) Page 7, analysis of unitary current of V55E: This analysis is not adequate. First, the assumption that the amplitude of unitary events with a duration > 0.716 ms should be fully resolved is based on a mean open time. If the channel has more than one short lived open state this value is meaningless. Second, the calculation of this threshold value for resolving full openings is only valid if the individual opening events are sufficiently separated. In bursts of channel activity, which seems to be the case here, the smearing effect of the filter is adding up and the threshold value becomes much larger. This effect is clearly seen in Fig. 3d. The small number of openings of the mutant at +200 mV with the same amplitude of the wt channel in Fig. 3e even suggest that the unitary current of the mutant reaches the same amplitude; the large cloud of smaller amplitudes clearly shows that the > 0.716 ms criterium is not sufficient for resolving full channel openings. Since these data are important for the main message of the computational part of the manuscript in that the open mutant channel has a lower conductance than the wt, the authors need to address this problem. One way of doing this could be by using the information of access noise. Yellen G (1984) Ionic permeation and blockade in Ca^{2+} -activated K^{+} channels of bovine chromaffin cells. *J Gen Physiol* 84, 157-186.
- 3) I wonder why the authors show in Fig. 4 c no data point for the mutant in the T59 distance between 0.84 and 0.86 nm. This distance is so critical for the analysis and the data in Fig. 4f show how much the current can increase by a very small change in distance.
- 4) Page 19, water dynamics: I find these data very interesting, but I have a few questions for understanding the results. The data in Figs 7 b, d, f show mean values {plus minus} SE of probabilities. I don't find a description on what the statistical representation means. Are these mean values from many events in different simulations which were normalized to a common time point in which D64 flips? Or are the data showing the mean behavior of the same filter monomer which undergoes the

same flipping many times? The authors need to tell the reader if these are selected episodes or if this is a main type of behavior.

5) On page 19 the authors mention that: ...the probability of water molecules entering behind the SF is higher throughout the 50ns period... This description is a bit confusion since it seems that the water in Fig. 7d is permanently present in this position.

We thank all the Reviewers for their positive assessment of our work. We answer the remaining comments in green.

Reviewer #3 (Comments to the Authors):

The revision of a previous version of this work has greatly improved the manuscript. In principle I keep on supporting publication of the work but before being accepted the authors need to rectify some crucial issues.

1) Page 5, ref 19. This reference addresses filter gating in a K2P channel. This type of channels and the information on its filter gating is not considered in the present paper. Hence the reference should be deleted, or the authors have to include the mechanism of filter gating in K2P channels throughout the manuscript.

We have removed ref 19.

2) Page 7, analysis of unitary current of V55E: This analysis is not adequate. First, the assumption that the amplitude of unitary events with a duration > 0.716 ms should be fully resolved is based on a mean open time. If the channel has more than one short lived open state this value is meaningless.

The assumption concerning the 0.716 ms threshold actually is *not* based on the mean open time. It is based on the amplitude of a square pulse after low-pass filtering at 1 kHz (see Figure 8F in McManus et al. 1987).

Perhaps the reviewer is referring to our estimates of the *fraction* of openings *brief*er than 0.716 ms. These were based on mean open times, as noted on p.7 of the previous version. No conclusions about the amplitudes were based on those estimates; these were only noted to make the point that V55E gating comprises many brief openings whose amplitudes were likely truncated by filtering. In any case, the section on analysis of unitary currents has been substantially revised and these estimates are no longer included.

Second, the calculation of this threshold value for resolving full openings is only valid if the individual opening events are sufficiently separated. In bursts of channel activity, which seems to be the case here, the smearing effect of the filter is adding up and the threshold value becomes much larger. This effect is clearly seen in Fig. 3d.

We believe that the “smearing” effect” that the reviewer is referring to in the all-points histograms in Fig 3D (Fig 3B in the newly-revised version) is arising from the large fraction of points falling in transitions between the open and closed levels, as we now note in the text. These all-points histograms are presented on both a linear scale (top half of panel) and logarithmic scale (bottom half of the panel).

The small number of openings of the mutant at +200 mV with the same amplitude of the wt channel in Fig. 3e even suggest that the unitary current of the mutant reaches the same amplitude; the large cloud of smaller amplitudes clearly shows that the > 0.716 ms criterium is not sufficient for resolving full channel openings.

We agree that the 0.716 ms criterion may be insufficient to resolve full openings very well, because of errors arising from the presence of open channel noise. We therefore re-analyzed these data using a longer threshold of 1.074 ms (6 x the system deadtime). Our new analysis did not substantially affect the

mean unitary current estimates, but it did eliminate outliers for which the measured amplitudes were likely affected by noise (including the few V55E openings with apparently high amplitudes). We thought that these data would be represented more clearly by constructing histograms of the unitary current amplitudes (revised Fig 3E, F).

Since these data are important for the main message of the computational part of the manuscript in that the open mutant channel has a lower conductance than the wt, the authors need to address this problem. One way of doing this could be by using the information of access noise. Yellen G (1984) Ionic permeation and blockade in Ca²⁺-activated K⁺ channels of bovine chromaffin cells. *J Gen Physiol* 84, 157-186.

As the reviewer knows, the analysis performed in Yellen (1984) was aimed at estimating opening and closing rates for a fast blocker. This was not the goal of our analysis in this manuscript.

In response to the reviewer's points on the caveats of our analysis, we have substantially rewritten the section on analysis of the unitary currents, and the accompanying Fig 3. We have also added a paragraph in the Methods to better explain the analysis protocols. We hope that this newly revised section is now written more clearly than the previous version to support our conclusions.

3) I wonder why the authors show in Fig. 4 c no data point for the mutant in the T59 distance between 0.84 and 0.86 nm. This distance is so critical for the analysis and the data in Fig. 4f show how much the current can increase by a very small change in distance.

We conducted additional 2 sets of simulations (10us each) aiming at the T59 distance between 0.84 and 0.86 and updated Figure 4. We present the same figure below, with green arrows pointing at the new data points. Indeed, we see slightly higher currents (ca. 10 pA) in these simulations, as compared to all other simulations from this set. We therefore modified the sentence on page 9 (new value in bold): "Around the opening of 1.6 nm, the current reaches its maximum of ~16 pA in WT and ~**10** pA in V55E." Given only a minimal increase in current in comparison to the previous version of the manuscript, we left the remaining text unaltered.

4) Page 19, water dynamics: I find these data very interesting, but I have a few questions for understanding the results. The data in Figs 7 b, d, f show mean values {plus minus} SE of probabilities. I don't find a description on what the statistical representation means. Are these mean values from many events in different simulations which were normalized to a common time point in which D64 flips? Or are

the data showing the mean behavior of the same filter monomer which undergoes the same flipping many times? The authors need to tell the reader if these are selected episodes or if this is a main type of behavior.

We apologize for the unclear description of this part. It is indeed the former representation. We added the following sentence to the caption of Figure 7:

“For calculating probabilities, all trajectories in which a D64 flip occurred were aligned at the time of the first D64 flip ($t=0$) and the probability was at each frame over all of these trajectories.”

5) On page 19 the authors mention that:the probability of water molecules entering behind the SF is higher throughout the 50ns period... This description is a bit confusion since it seems that the water in Fig. 7d is permanently present in this position.

We apologize for this confusion and agree with the reviewer - we meant that this probability is higher in MthK V55E than in MthK WT (Fig 7d vs Fig 7b). We modified the sentence accordingly:

“A similar overall pattern is observed in simulations of MthK V55E with the same force field (Fig. 7 C, D), however the probability of water molecules entering behind the SF is higher throughout the 50ns period prior to the aspartate flip than in MthK WT (Fig. 7 B). “

May 25, 2023

Dr. Wojciech Kopec
Max Planck Institute for Multidisciplinary Sciences
Theoretical and Computational Biophysics
Am Fassberg 11
Goettingen 37077
Germany

Re: 202213166R2

Dear Dr. Kopec,

I am pleased to let you know that your manuscript, entitled "Driving Forces underlying Selectivity Filter Gating in the MthK Potassium Channel" is scientifically acceptable for publication in Journal of General Physiology. Formal acceptance will follow when it is modified in accordance with our editorial policies.

Please note items that need attention are listed at the bottom of this email (under 'manuscript formatting checklist') and on the attached marked-up pdf file. Please also be sure to include a copy of the text with alterations highlighted (boldfaced or underlined). Your manuscript should be a double-spaced MS Word file and include editable tables, if appropriate.

Also, JGP now requires a data availability statement for all research article submissions. These statements will be published in the article directly above the Acknowledgments. The statement should address all data underlying the research presented in the manuscript. Please visit the JGP instructions for authors for guidelines and examples of statements at <https://rupress.org/jgp/pages/editorial-policies#data-availability-statement>.

Please submit your final files via this link:
Link Not Available

Thank you for choosing to publish your research in JGP and please feel free to contact me with any questions.

Sincerely,

Crina Nimigean, Ph.D.
On behalf of Journal of General Physiology

Journal of General Physiology's mission is to publish mechanistic and quantitative molecular and cellular physiology of the highest quality; to provide a best in class author experience; and to nurture future generations of independent researchers.

Manuscript formatting checklist:

- MS Word document of text needed (including editable tables)
- MS Word document of supplemental text needed, if applicable (including figure legends and editable tables)
- Brief Statement describing supplementary information needed, if applicable (in subsection at end of Materials & Methods)
- Please include a data availability statement preceding the Acknowledgments section. Please see <https://rupress.org/jgp/pages/editorial-policies#data-availability-statement>
- References need to follow JGP style (This article has numbered citations in it, which is not the style we use). Please refer to our guidelines here: <https://rupress.org/jgp/pages/reference-guidelines>
- Figures created at sufficient resolution and in acceptable format (including supplemental if applicable). If working in Illustrator, we prefer .ai or .eps file format. If working in Photoshop please use 600dpi/1000dpi .tiff or .psd file format. Minimum resolution at estimated print size: Minimum resolution for all figures is 600 dpi. For figures that contain both photographs and line art or text, 600 dpi is highly recommended. Figures containing only black and white elements (line art, no color, and no gray) should be 1,000 dpi. Maximum figure size is 7 in wide x 9 in high (17.5 x 22.8 cm) at the correct resolution. <https://jgp.rupress.org/fig-vid-guidelines>
- Supplemental figures, if any, conforming to same guidelines as manuscript figures (noted above)
- If images resemble one from a prior publications, the author must seek permissions (to reproduce or adapt) from the original publisher. [You can resubmit your paper while waiting to hear back from the original publisher but please keep us updated]
- All authors must complete a disclosure form prior to acceptance. A link to complete the form has been sent to all coauthors. Please provide the editorial office with updated email addresses if necessary

Reviewer #3 (Comments to the Authors):

I am satisfied with the latest revision of the manuscript